# CLIPPING IMPROVES ADAM AND ADAGRAD WHEN THE NOISE IS HEAVY-TAILED

## ABSTRACT

Methods with adaptive stepsizes, such as AdaGrad and Adam, are essential for training modern Deep Learning models, especially Large Language Models. Typically, the noise in the stochastic gradients is heavy-tailed for the later ones. Gradient clipping provably helps to achieve good high-probability convergence for such noises. However, despite the similarity between AdaGrad/Adam and Clip-SGD, the current understanding of the high-probability convergence of AdaGrad/Adam-type methods is limited in this case. In this work, we prove that AdaGrad/Adam (and their delayed version) can have provably bad high-probability convergence if the noise is heavy-tailed. We also show that gradient clipping fixes this issue, i.e., we derive new high-probability convergence bounds with polylogarithmic dependence on the confidence level for AdaGrad and Adam with clipping and with/without delay for smooth convex/non-convex stochastic optimization with heavy-tailed noise. Our empirical evaluations highlight the superiority of clipped versions of AdaGrad/Adam in handling the heavy-tailed noise.

## 1 INTRODUCTION

Stochastic first-order optimization methods such as Stochastic Gradient Descent (SGD) (Robbins & Monro, 1951) are the methods of choice in training modern Machine Learning (ML) and Deep Learning (DL) models (Shalev-Shwartz & Ben-David, 2014; Goodfellow et al., 2016). There are multiple reasons for that, including but not limited to their simplicity, computation cost, memory usage, and generalization. However, standard SGD is rarely used due to its sensitivity to the choice of stepsize. Therefore, methods such as AdaGrad (Streeter & McMahan, 2010; Duchi et al., 2011) and Adam (Kingma & Ba, 2014), which use adaptive[1] stepsizes, are much more popular in the DL community (Vaswani et al., 2017; You et al., 2019; Nikishina et al., 2022; Moskvoretskii et al., 2024). In particular, Adam-type methods are not just easier to tune but they also typically achieve much better results in terms of the model performance than SGD in the training of Large Language Models (LLMs) (Devlin et al., 2019; Zhang et al., 2020).

In the attempt to explain the later phenomenon, Zhang et al. (2020) consider the noise distribution in the stochastic gradients appearing in the pre-training of the BERT model (Devlin et al., 2019) and show that (i) the gradient noise is heavy-tailed in this case, (ii) Adam significantly outperforms SGD (with momentum), (iii) Clip-SGD (Pascanu et al., 2013) also converges better than SGD for such problems, and (iv) Clip-SGD is provably convergent (in-expectation) when the noise has bounded $\alpha$-th moment for some $\alpha \in (1, 2]$ while SGD can diverge for $\alpha < 2$. Moreover, gradient clipping also plays a central role in the recent advances on the *high-probability convergence* of stochastic methods under the heavy-tailed noise (Gorbunov et al., 2020; Cutkosky & Mehta, 2021; Sadiev et al., 2023; Nguyen et al., 2023). Taking into account the similarities between Adam and Clip-SGD (the former one can be seen as Clip-SGD with momentum and iteration-dependent clipping level), one can conjecture that Adam enjoys good theoretical high-probability convergence when the gradient noise is heavy-tailed. If this was true, it would be perfectly aligned with the observations from (Zhang et al., 2020) about the connection between the noise in the gradients and Adam's performance. Moreover, some recent works show that AdaGrad/Adam have provable convergence

---

[1]Throughout the paper, we use the word "adaptivity" in its general meaning: stepsizes are adaptive if they depend on the (stochastic) gradients or function values. We emphasize that, in this sense, an adaptive method can still have parameters affecting its convergence.

under generalized smoothness assumptions (Faw et al., 2023; Wang et al., 2023; Li et al., 2023; Wang et al., 2024). Since Clip-SGD has similar convergence properties and since some authors explicitly mention that in this regard Adam and Clip-SGD are similar[2], it is natural to conjecture that clipping is not needed in Adam/AdaGrad.

However, there are no theoretical results showing the high-probability convergence with *polylogarithmic dependence on the confidence level* of Adam under the heavy-tailed noise and even in the case of the bounded variance. Even for simpler "twin"[3] such as AdaGrad there exists a similar gap in the literature. Moreover, Mosbach et al. (2020) apply gradient clipping even for Adam in the fine-tuning of BERT and ALBERT (Lan et al., 2019) models. However, Mosbach et al. (2020) do not report the results that can be achieved by Adam without clipping. Therefore, it remains unclear whether and when the gradient clipping is needed for AdaGrad/Adam and whether Ada-Grad/Adam enjoy desirable high-probability convergence under the heavy-tailed noise.

In this work, we address this gap in the literature, i.e., we consider the following questions:

*Does the high-probability complexity of Adam/AdaGrad without clipping has*

*polylogarithmic dependence on the confidence level under the heavy-tailed noise?*

*Does clipping improve the convergence of AdaGrad/Adam under the heavy-tailed noise?*

We provide a negative answer to the first question and a positive answer to the second one.

## 1.1 OUR CONTRIBUTIONS

The main contributions of this work are summarized below.

- **Negative results for Adam and AdaGrad.** We show that the high-probability complexities of Adam and AdaGrad and their variants with delay by Li & Orabona (2020) do not have polylogarithmic dependence on the confidence level in the worst case when the noise is heavy-tailed. In particular, we design an example of a convex stochastic optimization problem such that the noise is heavy-tailed and the high-probability convergence complexity of Adam/AdaGrad has the inverse-power dependence on the target accuracy and confidence level.

- **Clipping fixes Adam and AdaGrad.** We prove that the above issue can be addressed via gradient clipping. That is, we derive high-probability complexity results for Clip-Adam and Clip-AdaGrad (with and without momentum) in the case of smooth convex (for the methods with delay) and non-convex (for the methods with and without delay) optimization with the heavy-tailed noise having bounded $\alpha$-th moment with $\alpha \in (1, 2]$. The obtained results have the desired polylogarithmic dependence on the confidence level. Moreover, in the non-convex case, the derived complexities are optimal up to logarithmic factors, and match the complexity of Clip-SGD in the convex case up to logarithmic factors.

- **Numerical experiments.** We conducted numerical experiments for synthetic and real-world problems. More precisely, we illustrate the superiority of different versions of Adam/AdaGrad with clipping to the non-clipped versions of Adam/AdaGrad on a simple quadratic problem with additive heavy-tailed noise in the gradients. Next, we also test Adam with and without clipping on the fine-tuning of ALBERT Base model (Lan et al., 2019) on CoLa and RTE datasets (Wang et al., 2018) and observe that Adam with clipping significantly outperforms Adam without clipping when the noise is heavy-tailed.

## 1.2 PRELIMINARIES

In this section, we formalize the setup. We focus on unconstrained minimization problems

$$\min_{x \in \mathbb{R}^d} f(x), \tag{1}$$

---

[2]Pan & Li (2023) write in the abstract: *"We conclude that the sharpness reduction effect of adaptive coordinate-wise scaling is the reason for Adam's success in practice."* In addition, Zhou et al. (2020) mention in the discussion of the related work: *"... adaptation in ADAM provides a clipping effect."*

[3]The existing convergence results for Adam often require the choice of parameters that make Adam very similar to AdaGrad with momentum (Défossez et al., 2022); see more details in Section 1.3.

where the differentiable function $f(x)$ is accessible through the calls of stochastic first-order oracle returning an approximation $\nabla f_\xi(x)$ of $\nabla f(x)$. Here $\xi$ is a random variable following some distribution $\mathcal{D}$ that may be dependent on $x$ and time. In the simplest case, $f_\xi(x)$ is a loss function on the data sample $\xi$ and $f(x) = \mathbb{E}_{\xi \sim \mathcal{D}}[f_\xi(x)]$ is a population risk (Shalev-Shwartz & Ben-David, 2014).

**Notation.** The notation is quite standard in this work. We use $\mathbb{E}_\xi[\cdot]$ to denote an expectation w.r.t. random variable $\xi$. All norms are standard Euclidean ones: $\|x\| = \sqrt{\langle x, x \rangle}$. The ball centered at $x$ with a radius $R$ is defined as $B_R(x) := \{y \in \mathbb{R}^d \mid \|y - x\| \leq R\}$. We also use $x^*$ to denote (any) solution of (1) and $f_* := \inf_{x \in \mathbb{R}^d} f(x)$. Clipping operator with clipping level $\lambda > 0$ is defined as $\texttt{clip}(x, \lambda) := \min\{1, \lambda/\|x\|\}x$ for $x \neq 0$ and $\texttt{clip}(x, \lambda) := 0$ for $x = 0$.

**Assumptions.** We start with the assumption[4] about the noise.

**Assumption 1.** *There exists set $Q \subseteq \mathbb{R}^d$ and $\sigma \geq 0, \alpha \in (1, 2]$ such that the oracle satisfies*

$$\mathbb{E}[\nabla f_\xi(x)] = \nabla f(x), \quad \mathbb{E}[\|\nabla f_\xi(x) - f(x)\|^\alpha] \leq \sigma^\alpha. \tag{2}$$

The above assumption is used in many recent works (Zhang et al., 2020; Cutkosky & Mehta, 2021; Sadiev et al., 2023; Nguyen et al., 2023). When $\alpha < 2$, it allows the stochastic gradients to have unbounded variance, e.g., Lévy $\alpha$-stable noise. When $\alpha = 2$, it reduces to the standard bounded variance assumption (Nemirovski et al., 2009; Ghadimi & Lan, 2012; 2013; Takáč et al., 2013).

Next, we make a standard assumption about the smoothness of the objective function.

**Assumption 2.** *There exists set $Q \subseteq \mathbb{R}^d$ and $L > 0$ such that for all $x, y \in Q$*

$$\|\nabla f(y) - \nabla f(x)\| \leq L \|y - x\|, \quad \|\nabla f(x)\|^2 \leq 2L(f(x) - f_*). \tag{3}$$

We emphasize that the second part of (3) follows from the first part if $Q = \mathbb{R}^d$. However, in more general situations, this is not always the case; see (Sadiev et al., 2023, Appendix B) for further details. Interestingly, when $Q$ is a compact set, function $f$ can have non-Lipschitz gradients (e.g., polynomially growing with $x$) on $\mathbb{R}^d$, see also (Patel et al., 2022; Patel & Berahas, 2022).

In addition, for some of our results, we assume that the objective is convex.

**Assumption 3** (Optional)**.** *There exists set $Q \subseteq \mathbb{R}^d$ such that for all $x, y \in Q$*

$$f(y) \geq f(x) + \langle \nabla f(x), y - x \rangle. \tag{4}$$

Finally, for the methods without the delay, we assume that function $f$ is bounded.

**Assumption 4** (Optional)**.** *There exists constant $M > 0$ such that for all $x \in \mathbb{R}^d$*

$$f(x) - f_* \leq M. \tag{5}$$

A stronger version of the above assumption (boundedness of the empirical risk) is used in (Li & Liu, 2023), which is the only existing work analyzing AdaGrad with gradient clipping.

**Why high-probability convergence?** The vast majority of the existing literature on stochastic optimization focuses on the in-expectation convergence guarantees only. In particular, for some metric $\mathcal{P}(x)$ quantifying the output's quality, e.g., $\mathcal{P}(x) = f(x) - f(x^*)$, $\|\nabla f(x)\|^2$, $\|x - x^*\|^2$, such guarantees provide upper bounds on the number of iterations/oracle calls required for a method to find $x$ such that $\mathbb{E}[\mathcal{P}(x)] \leq \varepsilon$. However, during recent years, *high-probability convergence* guarantees have been gaining a lot of attention as well. Such guarantees give upper bounds on the number of iterations/oracle calls required for a method to find $x$ such that $\mathbb{P}\{\mathcal{P}(x) \leq \varepsilon\} \geq 1 - \delta$, where $\delta$ is usually called confidence level or failure probability. One can argue that using Markov's inequality, one can easily deduce a high-probability guarantee from an in-expectation one: if

---

[4]Similarly to (Sadiev et al., 2023), for our results, it is sufficient to make all the assumptions only on some set $Q$. This set is typically bounded and depends on some metric of sub-optimality of the starting point, e.g., the distance from the starting point to the optimum. We emphasize that our assumptions are strictly weaker than corresponding ones for $Q = \mathbb{R}^d$. To achieve this kind of generality, we prove that the proposed method does not leave some set $Q$ with high probability.

$\mathbb{E}[\mathcal{P}(x_{K(\varepsilon\delta)})] \leq \varepsilon\delta$, where $x_{K(\varepsilon\delta)}$ is an output of the method after $K(\varepsilon\delta)$ iterations/oracle calls, then $\mathbb{P}\{\mathcal{P}(x_{K(\varepsilon\delta)}) > \varepsilon\} < \mathbb{E}[\mathcal{P}(x_{K(\varepsilon\delta)})]/\varepsilon \leq \delta$. Unfortunately, for many methods such as SGD (Ghadimi & Lan, 2013) $K(\varepsilon)$ has inverse-power dependence on $\varepsilon$ implying that $K(\varepsilon\delta)$ has inverse-power dependence on $\varepsilon\delta$, leading to a noticeable deterioration when $\delta$ is small. Therefore, deriving high-probability complexities with *polylogarithmic dependence on $\delta$* requires a separate and thorough consideration and analysis. Moreover, such bounds more accurately reflect the methods' behavior than in-expectation ones (Gorbunov et al., 2020).

## 1.3 RELATED WORK

**High-probability convergence.** The first results showing the high-probability convergence of SGD and its variants are derived under the sub-Gaussian noise assumption for convex and strongly convex problems by Nemirovski et al. (2009); Ghadimi & Lan (2012); Harvey et al. (2019) for non-convex problems by Li & Orabona (2020). Although the distribution of the noise is near-sub-Gaussian in some cases, like in the training of ResNet50 (He et al., 2016) on ImageNet (Russakovsky et al., 2015) as shown by Zhang et al. (2020), this assumption does not cover even the distributions with bounded variance. To relax the sub-Gaussian noise assumption, Nazin et al. (2019) consider a truncated version of Stochastic Mirror Descent, which is closely related to Clip-SGD, and prove its high-probability complexity with polylogarithmic dependence on $\delta$ under bounded variance assumption for convex smooth problems on the bounded domain. In the strongly convex case, Davis et al. (2021) propose a general approach for obtaining high-probability convergence based on the robust distance estimation and show accelerated high-probability rates in the strongly convex case. Next, for the unconstrained problems, Gorbunov et al. (2020) prove the first high-probability convergence results for Clip-SGD and the first accelerated high-probability rates in the convex case for a version of Clip-SGD with Nesterov's momentum (Nesterov, 1983). This result is generalized to the problems with Hölder-continuous gradients by Gorbunov et al. (2021). Cutkosky & Mehta (2021) derive the first high-probability convergence results under Assumption 1 with $\alpha < 2$ for a version of Clip-SGD with normalization and Polyak's momentum (Polyak, 1964) in the case of non-convex problems with bounded gradient. Sadiev et al. (2023) remove the bounded gradient assumption in the non-convex case and also prove the first high-probability convergence results under Assumption 1 for Clip-SGD and its accelerated version in the convex and strongly convex cases. Nguyen et al. (2023) provide improved results in the non-convex case under Assumption 1 and also improved the dependency on the logarithmic factors in the convergence bounds. The generalization to the composite and distributed optimization problems is developed by Gorbunov et al. (2024). It is also worth mentioning (Jakovetić et al., 2023; Puchkin et al., 2024) who consider potentially heavier noise than in Assumption 1 through utilizing the additional structure of the noise such as (near-)symmetry. This direction is further explored by Kornilov et al. (2024) and adjusted to the case of the zeroth-order stochastic oracle.

**AdaGrad and Adam.** AdaGrad[5] (Streeter & McMahan, 2010; Duchi et al., 2011) has the following update-rule

$$x_{t+1} = x_t - \frac{\gamma}{b_t}\nabla f_{\xi_t}(x_t), \quad \text{where} \quad b_t = \sqrt{b_{t-1}^2 + (\nabla f_{\xi_t}(x_t))^2} \qquad \text{(AdaGrad-CW)}$$

where all operations (taking a square and taking a square root of a vector, division by a vector) are performed coordinate-wise. The method is analyzed in many works, including (Streeter & McMahan, 2010; Duchi et al., 2011; Zou et al., 2018; Chen et al., 2018; Ward et al., 2020; Défossez et al., 2022; Faw et al., 2022) to name a few. However, the high-probability convergence of AdaGrad is studied under restrictive assumptions such as almost surely sub-Gaussian noise (Li & Orabona, 2020; Liu et al., 2023) or without such an assumption but with inverse-power dependence on the confidence level $\delta$ (Wang et al., 2023) or boundedness of the empirical risk and (non-central) $\alpha$-th moment (Li & Liu, 2023), which in the worst case implies boundedness of the stochastic gradient (see the discussion after Theorem 4). In contrast, our results for Clip-Adam(D)/Clip-M-AdaGrad(D) hold under Assumption 1 (and under additional Assumption 4 for the methods without delay) and have polylogarithmic dependence on the confidence level $\delta$.

---

[5]The original AdaGrad is described in formula (AdaGrad-CW). However, for the sake of simplicity, we use the name AdaGrad to describe a "scalar" version of AdaGrad also known as AdaGrad-Norm (Ward et al., 2020), see Algortihm 1 for the pseudocode. A similar remark holds for Adam.

Adam (Kingma & Ba, 2014) can be seen as a modification of AdaGrad with an exponential moving average $b_t^2$ of the squared stochastic gradients and with Polyak's momentum (Polyak, 1964):

$$x_{t+1} = x_t - \frac{\gamma}{b_t} m_t, \tag{Adam-CW}$$

$$m_t = \beta_1 m_{t-1} + (1 - \beta_1)\nabla f_{\xi_t}(x_t), \quad b_t = \sqrt{\beta_2 b_{t-1}^2 + (1 - \beta_2)(\nabla f_{\xi_t}(x_t))^2}, \tag{6}$$

where all operations (taking a square and taking a square root of a vector, division by a vector) are performed coordinate-wise. Although the original proof by Kingma & Ba (2014) has a flaw spotted by Reddi et al. (2019), one can still show the convergence of Adam when $\beta_2$ goes to 1 (Défossez et al., 2022; Zhang et al., 2022; Wang et al., 2024). Moreover, for any fixed $\beta_1$ and $\beta_2$ such that $\beta_1 < \sqrt{\beta_2}$, e.g., for the default values $\beta_1 = 0.9$ and $\beta_2 = 0.999$, Adam is not guaranteed to converge (Reddi et al., 2019, Theorem 3). Therefore, the standard choice of $\beta_2$ in theory is $\beta_2 = 1 - 1/K$, where $K$ is the total number of steps, and that is why, as noticed by Défossez et al. (2022), AdaGrad and Adam are "twins". Indeed, taking $\beta_1 = 0$ (no momentum) and $\beta_2 = 1 - 1/K$ in (6) we get that $b_t^2 = (1 - 1/K)^{t+1}b_{-1}^2 + \frac{1}{K}\sum_{k=0}^t (1 - 1/K)^{t-k}(\nabla f_{\xi_k}(x_k))^2 = \Theta\left(b_{-1}^2 + \frac{1}{K}\sum_{k=0}^t (\nabla f_{\xi_k}(x_k))^2\right)$ since $1/4 = (1 - 1/2)^2 \leq (1 - 1/K)^{t-k} \leq 1$ for $0 \leq k \leq t \leq K$. Thus, up to the rescaling of $\gamma$ and $b_{-1}^2$ the effective stepsize of Adam-CW is $\Theta(\cdot)$ of the effective stepsize of AdaGrad-CW (though the points where the gradients are calculated can be quite different for these two methods). This aspect explains why AdaGrad and Adam have similar proofs and convergence guarantees. The high-probability convergence of Adam is studied by Li et al. (2023) under bounded noise and sub-Gaussian noise assumptions, while our results for Clip-Adam(D) do not require such assumptions.

## 2 FAILURE OF Adam/AdamD AND AdaGrad/AdaGradD WITH MOMENTUM

---

**Algorithm 1** Adam/AdamD and M-AdaGrad/M-AdaGradD

---

**Input:** Stepsize $\gamma > 0$, starting point $x_0 \in \mathbb{R}^d$, initial constant $b_{-1} > 0$ (for Adam and M-AdaGrad) or $b_0 > 0$ (for AdamD and M-AdaGradD), momentum parameters $\beta_1, \beta_2 \in [0, 1]$
1: Set $m_{-1} = 0$
2: **for** $t = 0, 1, \dots$ **do**
3: $\quad m_t = \beta_1 m_{t-1} + (1 - \beta_1)\nabla f_{\xi_t}(x_t)$
4: $\quad$ **if** no delay **then**
5: $\quad\quad b_t = \begin{cases} \sqrt{\beta_2 b_{t-1}^2 + (1 - \beta_2)\|\nabla f_{\xi_t}(x_t)\|^2} & \text{for Adam} \\ \sqrt{b_{t-1}^2 + \|\nabla f_{\xi_t}(x_t)\|^2} & \text{for M-AdaGrad} \end{cases}$
6: $\quad$ **else**
7: $\quad\quad b_{t+1} = \begin{cases} \sqrt{\beta_2 b_t^2 + (1 - \beta_2)\|\nabla f_{\xi_t}(x_t)\|^2} & \text{for AdamD} \\ \sqrt{b_t^2 + \|\nabla f_{\xi_t}(x_t)\|^2} & \text{for M-AdaGradD} \end{cases}$
8: $\quad$ **end if**
9: $\quad x_{t+1} = x_t - \frac{\gamma}{b_t}m_t$
10: **end for**

---

In this section, we present the negative result on the convergence of Adam, AdaGrad with Momentum (M-AdaGrad), and their delayed versions – AdamD/M-AdaGradD (Li & Orabona, 2020).

**Theorem 1.** *For any $\sigma > 0$ and sufficiently small $\varepsilon, \delta \in (0, 1)$, there exist problems (1) such that Assumptions 1, 2, 3, hold with with $L = 1$, $\alpha = 2$, and the iterates produced by Adam(D)/M-AdaGrad(D) with $x_0$ such that $\|x_0 - x^*\| \gg \gamma L$ and with $\beta_2 = 1 - 1/T$ for Adam(D) satisfy:*

$$\mathbb{P}\{f(x_T) - f(x^*) \geq \varepsilon\} \leq \delta \quad \implies \quad T = \Omega\left(\text{poly}(\varepsilon^{-1/2}, \delta^{-1/2})\right), \tag{7}$$

*i.e., the complexity of Adam(D)/M-AdaGrad(D) has inverse-power dependence on $\delta$.*

*Sketch of the proof.* To construct our example, we consider the Huber loss function (Huber, 1992)

$$f(x) = \begin{cases} \frac{1}{2}x^2, & \text{if } |x| \le \nu, \\ \nu\left(|x| - \frac{1}{2}\nu\right), & \text{otherwise,} \end{cases} \tag{8}$$

and design two specific sequences of noises (one for Adam/M-AdaGrad and the second one for AdamD/M-AdaGradD). For Adam/M-AdaGrad, we consider a discrete additive noise for the first step such that Markov's inequality holds as equality, and for the remaining steps, noise equals zero. Then, with high probability, $b_t$ becomes large after the first step, which slowdowns the method. As for AdamD/M-AdaGradD, similarly to Sadiev et al. (2023), we add the noise only to the last step: since $b_t$ is constructed using the norm of the previous stochastic gradient, the noise is independent of the stepsize and can spoil the last iterate. See the complete proofs and details in Appendix B. □

Interestingly, in the above example, it is sufficient to consider the noise with bounded variance to show that the high-probability convergence rates of Adam(D)/M-AdaGrad(D) depend polynomially on $\varepsilon^{-1}$ and $\delta^{-1/2}$. Moreover, following a similar argument to (Zhang et al., 2020, Remark 1), one can show the non-convergence of AdamD/M-AdaGradD when $\alpha < 2$. We also conjecture that for $\alpha < 2$ one can show even worse dependence on on $\varepsilon$ and $\delta$ for Adam/AdaGrad (or even non-convergence) since $b_t$ will grow with high probability even faster in this case. Moreover, we also emphasize that the negative result for Adam(D) is established only for $\beta_2 = 1 - 1/T$, which is a standard assumption to ensure convergence of Adam-type methods. Nevertheless, the negative result of Theorem 1 provides necessary evidence that Adam(D)/M-AdaGrad(D) do not achieve desired high-probability convergence rates and motivates us to apply clipping to Adam(D)/M-AdaGrad(D).

## 3 NEW RESULTS FOR Adam AND AdaGrad WITH CLIPPING

---

**Algorithm 2** Clip-Adam/Clip-AdamD and Clip-M-AdaGrad/Clip-M-AdaGradD

---

**Input:** Stepsize $\gamma > 0$, starting point $x_0 \in \mathbb{R}^d$, initial constant $b_{-1} > 0$ (for Adam and M-AdaGrad) or $b_0 > 0$ (for AdamD and M-AdaGradD), momentum parameters $\beta_1, \beta_2 \in [0, 1]$, level of clipping $\lambda > 0$

1: Set $m_{-1} = 0$
2: **for** $t = 0, 1, \dots$ **do**
3:    $m_t = \beta_1 m_{t-1} + (1 - \beta_1)\texttt{clip}\left(\nabla f_{\xi_t}(x_t), \lambda\right)$
4:    **if** no delay **then**
5:      $b_t = \begin{cases} \sqrt{\beta_2 b_{t-1}^2 + (1 - \beta_2)\|\texttt{clip}\left(\nabla f_{\xi_t}(x_t), \lambda\right)\|^2} & \text{for Clip-Adam} \\ \sqrt{b_{t-1}^2 + \|\texttt{clip}\left(\nabla f_{\xi_t}(x_t), \lambda\right)\|^2} & \text{for Clip-M-AdaGrad} \end{cases}$
6:    **else**
7:      $b_{t+1} = \begin{cases} \sqrt{\beta_2 b_t^2 + (1 - \beta_2)\|\texttt{clip}\left(\nabla f_{\xi_t}(x_t), \lambda\right)\|^2} & \text{for Clip-AdamD} \\ \sqrt{b_t^2 + \|\texttt{clip}\left(\nabla f_{\xi_t}(x_t), \lambda\right)\|^2} & \text{for Clip-M-AdaGradD} \end{cases}$
8:    **end if**
9:    $x_{t+1} = x_t - \frac{\gamma}{b_t}m_t$
10: **end for**

---

**Methods.** To address the issue indicated in Theorem 1, we consider Clip-Adam(D)/Clip-M-AdaGrad(D) (see Algorithm 2). In contrast to the existing practice (Pan & Li, 2023), we use clipping of the stochastic gradient not only in the update rule for momentum buffer $m_t$ (Line 3 in Algorithm 2), but also in the computation of the scaling factor $b_t$ (Lines 5 and 7 in Algorithm 2). The role of clipping in $m_t$ is similar to the role of clipping in Clip-SGD-type methods: it prevents the method from too large steps that may occur due to the presence of the heavy-tailed noise in the gradients. In this regard, it is important to select clipping level in such a way that bias and variance of the estimator are balanced. However, the role of clipping in $b_t$ is different: clipping prevents $b_t$ from growing too quickly since such a growth can lead to poor high-probability guarantees (see the proof's sketch of Theorem 1). We note that clipping is also used in Clip-AdaGrad (without momentum) for both $m_t$ and $b_t$ computation by Li & Liu (2023) but the authors do not comment about the role of clipping in $b_t$ and use restrictive assumptions as we explain later in this section.

**Convergence results.** We derive new high-probability convergence bounds for the generalized method formalized as Algorithm 2 in the convex and non-convex cases. The following theorem gives the main result for Clip-AdamD/Clip-AdaGradD in the convex case.

**Theorem 2** (Convex Case). *Let $K > 0$ and $\delta \in (0,1]$ and Assumptions 1, 2, and 3 hold for $Q = B_{2R}(x^*)$ for some $R \geq \|x_0 - x^*\|$. Assume that $\beta_1 \in [0,1)$, $\beta_2 = \frac{K}{K+1}$ (for Clip-AdamD)*

$$\gamma = \Theta\left(\min\left\{\frac{(1-\beta_1)^2 b_0}{LA}, \frac{\sqrt{1-\beta_1}Rb_0}{\sigma(K+1)^{\frac{1}{\alpha}}A^{\frac{\alpha-1}{\alpha}}}\right\}\right), \quad \lambda = \Theta\left(\frac{\sqrt{1-\beta_1}b_0 R}{\gamma A}\right), \quad (9)$$

*where $A = \ln\left(4(K+1)/\delta\right)$. Then, to guarantee $f(\overline{x}_K) - f(x^*) \leq \varepsilon$ with probability at least $1 - \delta$ for $\overline{x}_K = \frac{1}{K+1}\sum_{t=0}^{K} x_t$ Clip-AdamD/Clip-M-AdaGradD requires :*

$$\widetilde{O}\left(\max\left\{\frac{LR^2}{(1-\beta_1)^3\varepsilon}, \left(\frac{\sigma R}{(1-\beta_1)^{\frac{3}{2}}\varepsilon}\right)^{\frac{\alpha}{\alpha-1}}\right\}\right) \quad \text{iterations/oracle calls.} \quad (10)$$

*Moreover, with probability at least $1 - \delta$, all iterates $\{x_t\}_{t=0}^{K}$ stay in $Q$.*

Next, we present our main results for Clip-AdamD/Clip-M-AdaGradD and Clip-Adam/Clip-M-AdaGrad in the non-convex case.

**Theorem 3** (Non-Convex Case: Methods with Delay). *Let $K > 0$ and $\delta \in (0,1]$ and Assumptions 1 and 2 hold for $Q = \left\{x \in \mathbb{R}^d \mid \exists y \in \mathbb{R}^d : f(y) \leq f_* + 2\Delta \text{ and } \|x - y\| \leq \sqrt{\Delta}/20\sqrt{L}\right\}$ for some $\Delta \geq f(x^0) - f_*$. Assume that $\beta_1 \in [0,1)$, $\beta_2 = \frac{K}{K+1}$ (for Clip-AdamD)*

$$\gamma = \Theta\left(\min\left\{\frac{(1-\beta_1)^2 b_0}{L(K+1)^{\frac{\alpha-1}{3\alpha-2}}A}, \frac{\sqrt{1-\beta_1}b_0\sqrt{\Delta}}{\sqrt{L}\sigma(K+1)^{\frac{\alpha}{3\alpha-2}}A^{\frac{\alpha-1}{\alpha}}}, \right.\right. \quad (11)$$

$$\left.\left.\frac{(1-\beta_1)^{\frac{\alpha-1}{2\alpha-1}}b_0\Delta^{\frac{\alpha}{2\alpha-1}}}{\sigma^{\frac{2\alpha}{2\alpha-1}}L^{\frac{\alpha-1}{2\alpha-1}}(K+1)^{\frac{\alpha}{3\alpha-2}}A^{\frac{2\alpha-2}{2\alpha-1}}}\right\}\right), \quad \lambda = \Theta\left(\frac{\sqrt{1-\beta_1}b_0\sqrt{\Delta}}{\sqrt{L}\gamma A(K+1)^{\frac{\alpha-1}{3\alpha-2}}}\right), \quad (12)$$

*where $A = \ln\left(4(K+1)/\delta\right)$. Then, to guarantee $\frac{1}{K+1}\sum_{t=0}^{K}\|\nabla f(x_t)\|^2 \leq \varepsilon$ with probability at least $1 - \delta$ Clip-AdamD/Clip-M-AdaGradD requires the following number of iterations/oracle calls:*

$$\widetilde{O}\left(\max\left\{\left(\frac{L\Delta}{(1-\beta_1)^3\varepsilon}\right)^{\frac{3\alpha-2}{2\alpha-1}}, \left(\frac{\sigma\sqrt{L\Delta}}{(1-\beta_1)^{\frac{3}{2}}\varepsilon}\right)^{\frac{3\alpha-2}{2\alpha-2}}, \left(\frac{\sigma^{\frac{2\alpha}{2\alpha-1}}(L\Delta)^{\frac{\alpha-1}{2\alpha-1}}}{(1-\beta_1)^{\frac{3\alpha-2}{2\alpha-1}}\varepsilon}\right)^{\frac{3\alpha-2}{2\alpha-2}}\right\}\right). \quad (13)$$

*Moreover, with probability at least $1 - \delta$, all iterates $\{x_t\}_{t=0}^{K}$ stay in $Q$.*

**Theorem 4** (Non-Convex Case: Methods without Delay). *Let $K > 0$ and $\delta \in (0,1]$ and Assumptions 1, 2, 4 hold for $Q = \mathbb{R}^d$. Assume that $\beta_1 \in [0,1)$, $\beta_2 = 1 - \frac{1}{K}$ (for Clip-Adam)*

$$\gamma = \Theta\left(\min\left\{\frac{b_{-1}}{L(K+1)^{\frac{\alpha-1}{3\alpha-2}}A}, \frac{b_{-1}\sqrt{M}}{\sqrt{L}\sigma(K+1)^{\frac{\alpha}{3\alpha-2}}A^{\frac{\alpha-1}{\alpha}}}, \right.\right. \quad (14)$$

$$\left.\left.\frac{b_{-1}M^{\frac{\alpha}{2\alpha-1}}}{\sigma^{\frac{2\alpha}{2\alpha-1}}L^{\frac{\alpha-1}{2\alpha-1}}(K+1)^{\frac{\alpha}{3\alpha-2}}A^{\frac{2\alpha-2}{2\alpha-1}}}\right\}\right), \quad \lambda = \Theta\left(\frac{b_{-1}\sqrt{M}}{\sqrt{L}\gamma A(K+1)^{\frac{\alpha-1}{3\alpha-2}}}\right), \quad (15)$$

*where $A = \ln\left(4/\delta\right)$. Then, to guarantee $\frac{1}{K+1}\sum_{t=0}^{K}\|\nabla f(x_t)\|^2 \leq \varepsilon$ with probability at least $1 - \delta$ Clip-Adam/Clip-M-AdaGrad requires the following number of iterations/oracle calls:*

$$\widetilde{O}\left(\frac{1}{(1-\beta_1)^{\frac{3}{2}}}\max\left\{\left(\frac{LM}{\varepsilon}\right)^{\frac{3\alpha-2}{2\alpha-1}}, \left(\frac{\sigma\sqrt{LM}}{\varepsilon}\right)^{\frac{3\alpha-2}{2\alpha-2}}, \left(\frac{\sigma^{\frac{2\alpha}{2\alpha-1}}(L\Delta)^{\frac{\alpha-1}{2\alpha-1}}}{\varepsilon}\right)^{\frac{3\alpha-2}{2\alpha-2}}\right\}\right). \quad (16)$$

**Discussion of the results.** Theorems 2, 3, and 4 provide high-probability complexities for Clip-Adam(D)Clip-M-AdaGrad(D) with *polylogarithmic* dependence on the confidence level $\delta$. Up to the differences in logarithmic factors, these complexities coincide with the best-known ones for

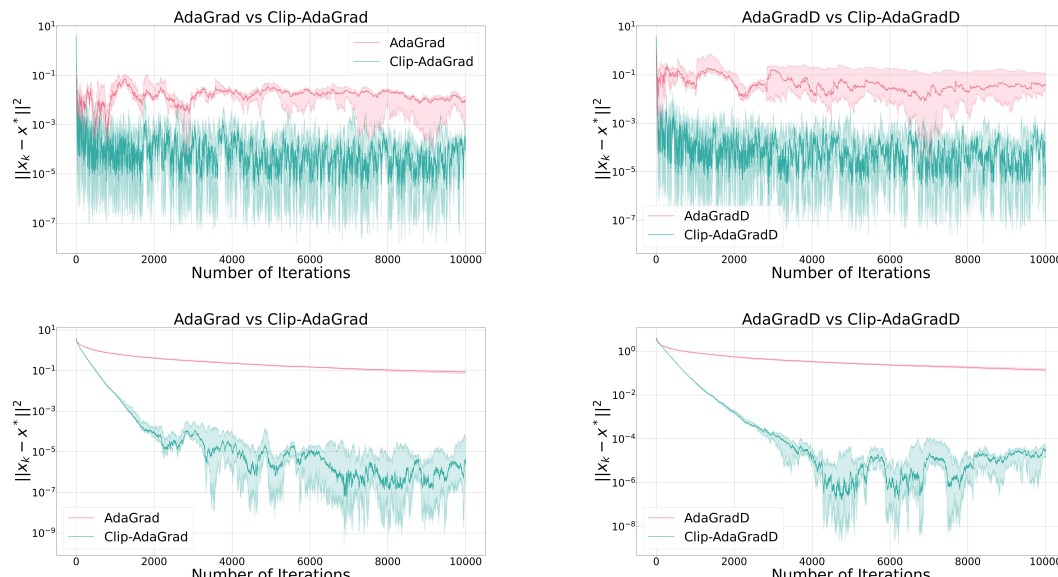

Figure 1: Performance of different versions of AdaGrad (with and without clipping/delay) with stepsizes $\gamma = 1$ (first row) and $\gamma = 1/16$ (second row) on the quadratic problem.

Clip-SGD (Sadiev et al., 2023; Nguyen et al., 2023). Moreover, the leading terms in (13) and (16) are optimal up to logarithmic factors (Zhang et al., 2020), though the first terms in (13) and (16) can be improved (Arjevani et al., 2023). In the convex case, the first term in (10) is not optimal (Nemirovskij & Yudin, 1983) and can be improved (Gorbunov et al., 2020; Sadiev et al., 2023). The optimality of the second term in (10) is still an open question.

It is also worth mentioning that the existing high-probability complexities for Adam/AdaGrad-type (without clipping) methods either have inverse power dependence on $\delta$ (Wang et al., 2023) or have polylogarithmic dependence on $\delta$ but rely on the assumption that the noise is sub-Gaussian/bounded (Li & Orabona, 2020; Liu et al., 2023; Li et al., 2023), which is stronger than bounded variance assumption. Under the additional assumption that the emprical risk is bounded and the (non-central) $\alpha$-th moment of the stochastic gradient are bounded and the empirical risk is smooth, which are stronger than Assumptions 4, 1 and 2 respectively, Li & Liu (2023) derive a similar bound to (16) for Clip-AdaGrad. We emphasize that boundedness and smoothness of the empirical risk imply the boundedness and smoothness of all $f_\xi(x)$ in the worst case (e.g., when the distribution $\mathcal{D}$ is discrete). Therefore, in the worst case, these assumptions imply the boundedness of $\nabla f_\xi(x)$ (in view of the second part of (3) for function $f_\xi$), meaning that the noise is bounded and, thus, sub-Gaussian. In this case, clipping is not needed for AdaGrad to achieve good high-probability convergence guarantees as shown by Li & Orabona (2020); Liu et al. (2023). Our Theorem 4 extends this result to the momentum version of Clip-AdaGrad under less restrictive assumptions (not implying sub-Gaussianity of the noise) and gives the first high-probability convergence bounds for Clip-Adam with polylogarithmic dependence on $\delta$. Moreover, to the best of our knowledge, Theorems 2 and 3 are the first results showing high-probability convergence of Adam/AdaGrad-type methods with polylogarithmic dependence on the confidence level in the case of the heavy-tailed noise without extra assumptions such as Assumption 4. Moreover, we also show that the iterates of Clip-AdamD/Clip-M-AdaGradD do not leave set $Q$ with high probability, where $Q = B_{2R}(x^*)$ in the convex case and $Q = \left\{ x \in \mathbb{R}^d \mid \exists y \in \mathbb{R}^d : f(y) \leq f_* + 2\Delta \text{ and } \|x - y\| \leq \sqrt{\Delta}/20\sqrt{L} \right\}$ in the non-convex case. Further details and proofs are deferred to Appendix C.

## 4 NUMERICAL EXPERIMENTS

In this section, we illustrate numerically that clipping indeed helps AdaGrad and Adam to achieve better high-probability convergence.

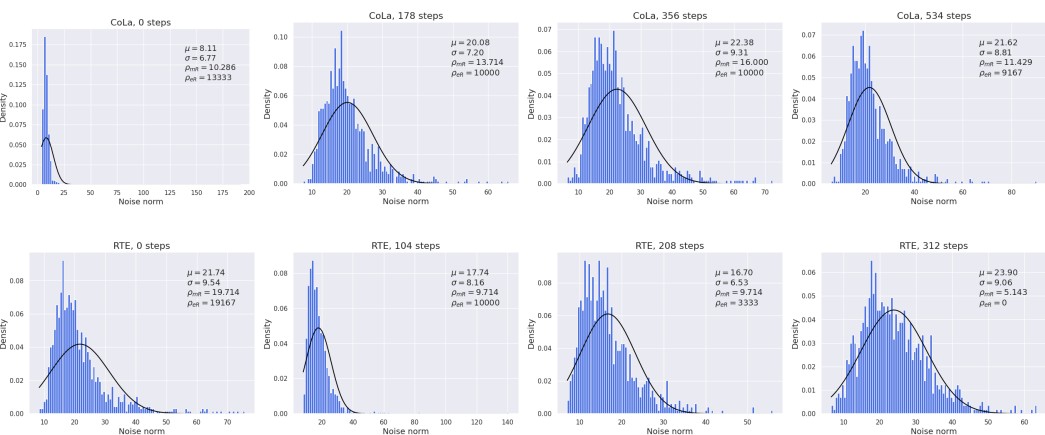

Figure 2: Gradient noise evolution for Adam on CoLa (the first row) and RTE (the second row) datasets. Histograms were evaluated after 0 steps, after $\approx 1/3$ and $\approx 2/3$ of all steps, and in the end.

**Quadratic problem.** In the first experiment, we test the performance of different versions of AdaGrad with and without clipping on the 1-dimensional quadratic objective with additive heavy-tailed noise: $f(x) = x^2/2$, $\nabla f_\xi(x) = x + \xi$, where the noise $\xi$ has probability density function $p(t) = \frac{3}{4(1+|t|)^{2.5}}$. In this case, Assumption 1 is satisfied with any $\alpha \in (1, 1.5)$ and the $\alpha$-th moment is unbounded for $\alpha \geq 1.5$. Moreover, the function is strongly convex and $L$-smooth with $L = 1$. We choose $x_0 = 2$, $b_0 = 3$ (for the versions of AdaGrad with delay), $b_{-1} = 3$ (for other cases), $\lambda = 1/2$ for the methods with clipping, and choose $\gamma$ from $\{1, 1/16, 1/128\}$. Each method was run 100 times with different seeds.

The results are given in Figure 1, where for each method, we show its trajectory in terms of the squared distance to the solution for $\gamma = 1$ and $\gamma = 1/16$ (the results for $\gamma = 1/128$ are given in Appendix D.1). More precisely, solid lines correspond to the median value of the squared distances, and the error bands cover the areas from the 10-th to 90-th percentiles of $(x_t - x^*)^2$. These results show that clipped versions of AdaGrad (with and without delay) achieve better convergence with higher probability than their non-clipped counterparts. Moreover, versions with clipping exhibit similar behavior to each other. That is, the error bands for Clip-AdaGrad(D) are lower than for AdaGrad(D) (note that the vertical axis is shown in the logarithmic scale making the error bands for Clip-AdaGrad(D) look wider than for AdaGrad(D), while they are not). In general, the observed results for AdaGrad-type methods are perfectly aligned with the theory developed in this paper. We provide the results for Adam with and without clipping/delay in Appendix D.1.

**ALBERT Base v2 fine-tuning.** In the second part of our experiments, we consider fine-tuning the pre-trained ALBERT Base v2 model (Lan et al., 2019) on CoLa and RTE datasets (Wang et al., 2018). Since Adam-based algorithms are the methods of choice for NLP tasks, in the main part of the paper, we focus on Adam and its clipped versions – Clip-Adam and Clip-AdamD – and provide additional experiments with AdaGrad-based methods in Appendix D.2. We took a pre-trained model from the Hugging Face library. Then, the model was fine-tuned following the methodology suggested by Mosbach et al. (2020). More precisely, we used linear warmup with warmup ratio being 0.1, and hyperparameters were $\beta_1 = 0.9$, $\beta_2 = 0.999$, $b = \epsilon\mathbf{1}$, where $\mathbf{1} = (1, 1, \ldots, 1)^\top \in \mathbb{R}^d$. We tuned batchsize and stepsize $\gamma$ for Adam and selected best values from $\{4, 8, 16, 32\}$ for the batchsize and from $\{10^{-6}, 3 \cdot 10^{-6}, 10^{-5}, 3 \cdot 10^{-5}, 10^{-4}\}$ for $\gamma$. For the CoLa dataset, the best batchsize was 16 and $\gamma = 10^{-5}$, and for the RTE dataset, the best batchsize was 8 and $\gamma = 10^{-5}$. For the methods with clipping, we used the same batchsize and stepsize as for Adam and tuned the clipping level for the two types of clipping[6]. We tested coordinate-wise clipping with $\lambda \in \{0.001, 0.002, 0.005, 0.01, 0.02, 0.05, 0.1, 0.2, 0.5, 1\}$ and layer-wise clipping with

---

[6]We did not consider the global/norm clipping (the considered in theory), since typically coordinate-wise or layer-wise clipping work better in training neural networks.

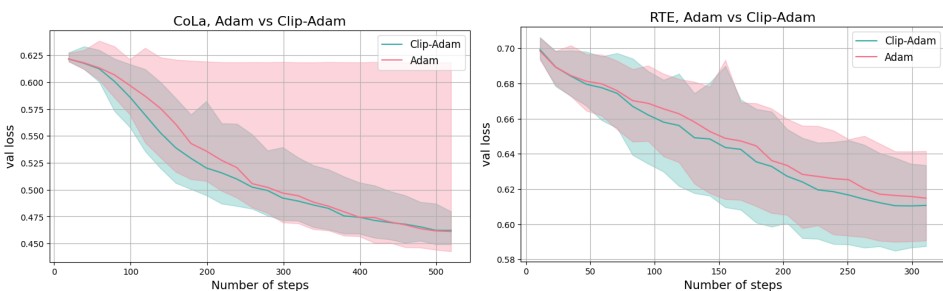

Figure 3: Validation loss for ALBERT Base v2 fine-tuning task on the CoLa and RTE datasets.

$\lambda \in \{0.1, 0.2, 0.5, 1, 2, 5, 10\}$. For the CoLa dataset, the best results were achieved with $\lambda = 1$ for layer-wise clipping and $\lambda = 0.02$ for coordinate-wise clipping, and for the RTE dataset, the best results were achieved with $\lambda = 2$ for layer-wise clipping and $\lambda = 0.005$ for coordinate-wise clipping. In the main text, we show the results with layer-wise clipping and defer the results with coordinate-wise clipping to Appendix D.2.

Before comparing the methods, we ran Adam and checked how heavy-tailed the noise in the stochastic gradients is along the trajectory. In particular, for both tasks, we selected 4 iterates corresponding to the starting point, points generated after $\approx 1/3$ and $\approx 2/3$ of all steps, and the last iterate. Then, for each of these points, we sampled size-16 (for CoLa) and size-8 (for RTE) mini-batched estimator $\nabla f_\xi(x)$ of the gradient 1000 times, saved the resulting norms of the differences $\|\nabla f_\xi(x) - \nabla f(x)\|$, and plotted their histogram, i.e., we plotted the histograms of the noise norm. Moreover, we also measure the heavy-tailedness of the noise following the approach from (Gorbunov et al., 2022): we compute two metrics $p_{mR} = F_{1.5}(\|\nabla f_\xi(x) - \nabla f(x)\|)$, which quantifies "mild" heavy tails, and $p_{eR} = F_3(\|\nabla f_\xi(x) - \nabla f(x)\|)$ introduced by Jordanova & Petkova (2017), which quantifies "extreme" heavy tails, where $F_a(\|\nabla f_\xi(x) - \nabla f(x)\|) = \mathbb{P}\{\|\nabla f_\xi(x) - \nabla f(x)\| > Q_3 + a(Q_3 - Q_1)\}$ and $Q_i$ is the $i$-th quartile of $\|\nabla f_\xi(x) - \nabla f(x)\|$. To illustrate the heavy-tailedness clearly, we divide these metrics to the ones computed for the standard normal distribution ($p_{mR\mathcal{N}}$ and $p_{eR\mathcal{N}}$) and show $\rho_{mR} = p_{mR}/p_{mR\mathcal{N}}$ and $\rho_{eR} = p_{eR}/p_{eR\mathcal{N}}$ on the plots.

The histograms are provided in Figure 2, where we additionally estimate the mean and standard deviation and plot the density of the normal distribution with these parameters (black curve). For the CoLa dataset, the noise distribution changes significantly after the start of the training, and its mean drifts to the right. However, the standard deviation does not change significantly, and, more importantly, metrics $\rho_{mR}$ and $\rho_{eR}$ remain quite large, showing that the distribution is significantly heavy-tailed. In contrast, for the RTE dataset, the noise distribution does not drift significantly, and, interestingly, $\rho_{eR}$ decreases towards the end of training and becomes zero, while $\rho_{mR}$ stays in the interval $[5, 10]$. Therefore, the noise distribution has much heavier tails for CoLa than for RTE.

Then, similarly to the experiments with the quadratic problem, we ran the methods 100 times, and for each step, we computed the median value of the validation loss and its 5-th and 95-th percentiles. The results are presented in Figure 3, where the solid lines correspond to the medians and the error bands cover the areas between 5-th and 95-th percentiles. As expected, Adam exhibits poor high-probability convergence on the CoLa datasets where the noise is significantly heavy-tailed, and Clip-Adam shows much better performance: the area between 5-th and 95-th percentiles is relatively narrow for Clip-Adam. In contrast, for the RTE dataset, Clip-Adam performs similarly to Adam. This is also expected since the noise is much less heavy for RTE, as Figure 2 shows. Taking into account the negative results from Section 2, and the upper bounds from Section 3, we conclude that these numerical results are well-aligned with the theory developed in the paper.

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

CONTENTS

## A  TECHNICAL DETAILS AND AUXILIARY RESULTS

**Additional notation.**  For the ease of exposition, we introduce the following notation for the proofs:

$$g_t = \texttt{clip}\left(\nabla f_{\xi_t}(x_t), \lambda\right),$$
$$\theta_t = g_t - \nabla f(x_t),$$
$$\theta_t^u = g_t - \mathbb{E}_{\xi_t}[g_t],$$
$$\theta_t^b = \mathbb{E}_{\xi_t}[g_t] - \nabla f(x_t),$$
$$R_t = \|x_t - x^*\|,$$
$$\Delta_t = f(x_t) - f_*.$$

**Auxiliary results.**  We also use the following standard results.

**Proposition 1** (Young's inequality.).  *For any $x, y \in \mathbb{R}^d$ and $p > 0$ the following inequality holds:*

$$\|x + y\|^2 \le (1 + p) \|x\|^2 + \left(1 + \frac{1}{p}\right) \|y\|^2.$$

*In particular, for $p = 1$*

$$\|x + y\|^2 \le 2\|x\|^2 + 2\|y\|^2.$$

**Lemma 1** (Lemma B.2 from (Défossez et al., 2022)).  *Let $0 \le a \le b$ be some non-negative integers and $0 \le q < 1$. Then,*

$$\sum_{k=a}^{b} q^k k \le \frac{q}{(1-q)^2}.$$

**Lemma 2** (Lemma 1 from (Streeter & McMahan, 2010)).  *Let $\{a_i\}_{i=1}^n$ and $c$ be non-negative reals. Then,*

$$\sum_{k=1}^{n} \frac{a_k}{\sqrt{c + \sum_{i=1}^{k} a_i}} \le 2\sqrt{c + \sum_{k=1}^{n} a_k}$$

The following lemma by Sadiev et al. (2023) helps to estimate bias and variance of the clipped stochastic gradient satisfying Assumption 1.

**Lemma 3** (Lemma 5.1 from (Sadiev et al., 2023)).  *Let $X$ be a random vector from $\mathbb{R}^d$ and $\widehat{X} = clip(X, \lambda)$. Then, $\left\|\widehat{X} - \mathbb{E}\left[\widehat{X}\right]\right\| \le 2\lambda$. Moreover, if for some $\sigma \ge 0$ and $\alpha \in (1, 2]$ we have $\mathbb{E}[X] = x \in \mathbb{R}^d$, $\mathbb{E}\left[\|X - x\|^\alpha\right] \le \sigma^\alpha$, and $\|x\| \le \frac{\lambda}{2}$, then*

$$\left\|\mathbb{E}\left[\widehat{X}\right] - x\right\| \le \frac{2^\alpha \sigma^\alpha}{\lambda^{\alpha-1}},$$

$$\mathbb{E}\left[\left\|\widehat{X} - x\right\|^2\right] \le 18\lambda^{2-\alpha}\sigma^\alpha,$$

$$\mathbb{E}\left[\left\|\widehat{X} - \mathbb{E}\left[\widehat{X}\right]\right\|^2\right] \le 18\lambda^{2-\alpha}\sigma^\alpha.$$

Finally, in the analysis of Clip-RAdaGradD, we face the sums of martingale-difference sequences. One of the tools that we use to handle them is Bernstein's inequality (Bennett, 1962; Dzhaparidze & Van Zanten, 2001; Freedman et al., 1975).

**Lemma 4** (Bernstein's inequality).  *Let the sequence of random variables $\{X_i\}_{i \ge 1}$ form a martingale difference sequence, i.e., $\mathbb{E}[X_i \mid X_{i-1}, \ldots, X_1] = 0$ for all $i \ge 1$. Assume that conditional variances $\sigma_i^2 = \mathbb{E}\left[X_i^2 \mid X_{i-1}, \ldots, X_1\right]$ exist and are bounded and also assume that there exists deterministic constant $c > 0$ such that $|X_i| \le c$ almost surely for all $i \ge 1$. Then for all $b > 0$, $G > 0$ and $n \ge 1$*

$$\mathbb{P}\left\{\left|\sum_{i=1}^{n} X_i\right| > b \text{ and } \sum_{i=1}^{n} \sigma_i^2 \le G\right\} \le 2\exp\left(-\frac{b^2}{2G + \frac{2cb}{3}}\right).$$

## B    MISSING PROOFS FROM SECTION 2

In this section, we provide further details regarding Theorem 1 giving a negative result about high-probability convergence of Adam/M-AdaGrad and AdamD/M-AdaGradD. For all methods, we use the 1-dimensional Huber loss function:

$$f(x) = \begin{cases} \frac{1}{2}x^2, & \text{if } |x| \leq \nu, \\ \nu\left(|x| - \frac{1}{2}\nu\right), & \text{otherwise.} \end{cases}$$

This function is convex and $L$-smooth with $L = 1$. However, the construction of noises and proofs are different for Adam, M-AdaGrad, AdamD, and M-AdaGradD. Therefore, we provide the negative results for these methods separately in the following subsections.

### B.1    FAILURE OF M-AdaGrad

We start with the following lemma giving a closed-form expression for the iterates of deterministic M-AdaGrad applied to (8).

**Lemma 5.** *Suppose that the starting point $x_0$ is such that $x_0 > 0$. If after $T$ iterations of deterministic M-AdaGrad with initial momentum $m_{-1}$ we have $|x_t| > \nu$ and $x_t > 0$ for all $t = \overline{1, T-1}$, then*

$$x_T = x_0 - \gamma\nu \sum_{t=0}^{T-1} \frac{1 - \beta_1^{t+1} + \beta_1^{t+1}\frac{m_{-1}}{\nu}}{\sqrt{b_{-1}^2 + (t+1)\nu^2}}.$$

*Proof.* Since $|x_t| > \nu$ and $x_t$ is positive, the gradient at $x_t$ is equal to $\nu$. Hence, by substituting the gradient into the algorithm, we get the final result. $\square$

The above lemma relies on the condition that $|x_t| > \nu$ and $x_t > 0$ for all $t = \overline{1, T-1}$. For any $\gamma, b_{-1}$ and $T$ this condition can be achieved if we choose sufficiently small $\nu$.

Next, we estimate the interval where $x_T$ lies.

**Lemma 6.** *Let the conditions of Lemma 5 hold. Then, we have*

$$x_T \geq x_0 - \gamma\left(1 + \frac{\max\{m_{-1}, 0\}}{\nu}\right)\left(\frac{1}{\sqrt{1+a_0}} + 2\sqrt{a_0 + T} - 2\sqrt{a_0 + 1}\right),$$

$$x_T \leq x_0 - \gamma\left(1 - \beta_1 + \beta_1\frac{\min\{m_{-1}, 0\}}{\nu}\right)\left(2\sqrt{a_0 + T + 1} - 2\sqrt{a_0 + 1}\right),$$

*where $a_0 = \frac{b_{-1}^2}{\nu^2}$.*

*Proof.* From Lemma 5 we have:

$$x_T = x_0 - \gamma \sum_{t=0}^{T-1} \frac{1 - \beta_1^{t+1} + \beta_1^{t+1}\frac{m_{-1}}{\nu}}{\sqrt{a_0 + (t+1)}},$$

where $a_0 = \frac{b_{-1}^2}{\nu^2}$. Next, we bound the second term in the following way:

$$\sum_{t=0}^{T-1} \frac{1 - \beta_1^{t+1} + \beta_1^{t+1}\frac{m_{-1}}{\nu}}{\sqrt{a_0 + (t+1)}} \geq \left(1 - \beta_1 + \beta_1\frac{\min\{m_{-1}, 0\}}{\nu}\right)\int_{a_0}^{a_0+T} \frac{1}{\sqrt{1+x}}\,dx$$

$$= \left(1 - \beta_1 + \beta_1\frac{\min\{m_{-1}, 0\}}{\nu}\right)(2\sqrt{a_0 + T + 1} - 2\sqrt{a_0 + 1}), \quad (17)$$

$$\sum_{t=0}^{T-1} \frac{1 - \beta_1^{t+1} + \beta_1^{t+1}\frac{m_{-1}}{\nu}}{\sqrt{a_0 + (t+1)}} \leq \frac{1 + \frac{\max\{m_{-1}, 0\}}{\nu}}{\sqrt{1+a_0}} + \left(1 + \frac{\max\{m_{-1}, 0\}}{\nu}\right)\int_{a_0}^{a_0+T-1} \frac{1}{\sqrt{1+x}}\,dx$$

$$= \left(1 + \frac{\max\{m_{-1}, 0\}}{\nu}\right)\left(\frac{1}{\sqrt{1+a_0}} + 2\sqrt{a_0 + T} - 2\sqrt{a_0 + 1}\right).$$

$$(18)$$

Combining (17) and (18), we get the final result. □

**Corollary 1.** *If* $x_0 - \gamma > \nu > 0$, $\hat{\gamma} = \gamma \left(1 + \frac{\max\{m_{-1},0\}}{\nu}\right)$ *and*

$$T < \frac{(x_0 - \nu - \hat{\gamma})^2 + 4\hat{\gamma}(x_0 - \nu - \hat{\gamma})\sqrt{a_0 + 1}}{4\hat{\gamma}^2} + 1,$$

*then* $x_T > \nu$ *for deterministic* M-AdaGrad*. Alternatively,* $|x_T| \leq \nu$ *implies that*

$$T \geq \frac{(x_0 - \nu - \hat{\gamma})^2 + 4\hat{\gamma}(x_0 - \nu - \hat{\gamma})\sqrt{a_0 + 1}}{4\hat{\gamma}^2} + 1.$$

*Proof.* First, let us show that

$$\nu < x_0 - \hat{\gamma}\left(1 + 2\sqrt{a_0 + T} - 2\sqrt{a_0 + 1}\right) \tag{19}$$

is equivalent to

$$T < \frac{(x_0 - \nu - \hat{\gamma})^2 + 4\gamma(x_0 - \nu - \hat{\gamma})\sqrt{a_0 + 1}}{4\hat{\gamma}^2} + 1.$$

Rewriting the (19), one can obtain

$$2\hat{\gamma}\sqrt{a_0 + T} < x_0 - \nu - \hat{\gamma} + 2\hat{\gamma}\sqrt{a_0 + 1}.$$

Squaring both parts of the inequality above and expressing $T$, we get the alternative equivalent formula. Noticing that $1 \geq \frac{1}{\sqrt{1 + a_0}}$ and applying Lemma 6, we get the final result. The second part of the corollary is just a negation of the implication stated in the first part of the corollary. □

**Theorem 5.** *For any* $\varepsilon, \delta \in (0,1), \sigma > 0$ *such that* $\sigma/\sqrt{\varepsilon\delta} \geq 4$, *there exists convex L-smooth minimization problem* (8) *and stochastic gradient oracle such that Assumption 1 holds with* $\alpha = 2$ *and the iterates produced by* M-AdaGrad *after K steps with stepsize* $\gamma$ *and starting point* $x_0$ *such that* $R := x_0 - \sqrt{2\varepsilon} - 3\gamma > 0$ *satisfy the following implication:*

$$\mathbb{P}\{f(x_K) - f(x^*) \geq \varepsilon\} \leq \delta \quad \implies \quad K = \Omega\left(\frac{b_{-1}R}{\sqrt{\varepsilon}\gamma} + \frac{\sigma R}{\gamma\sqrt{\varepsilon\delta}}\right), \tag{20}$$

*i.e., the high-probability complexity of* M-AdaGrad *has inverse-power dependence on* $\delta$.

*Proof.* Before we delve into the technical details, we provide an intuition behind the proof. We want to use the lower bound from Corollary 1 and estimate the bound for the number of iterations required to achieve the desired optimization error $\varepsilon$ with probability at least $1 - \delta$. Moreover, we need to set $\nu$ depending on the accuracy $\varepsilon$ ($\nu$ is analytically clarified later). We denote the output of deterministic M-AdaGrad after $t$ iterations as $\hat{x}_t$. Then, we introduce the noise in the stochastic gradient in the following way

$$g_k = \nabla f(x_k) - \sigma\xi_k,$$

where

$$\xi_k = \begin{cases} 0, & \text{for } k > 0, \\ \begin{cases} -A, & \text{with probability } \frac{1}{2A^2} \\ 0, & \text{with probability } 1 - \frac{1}{A^2} \\ A, & \text{with probability } \frac{1}{2A^2} \end{cases} & \text{otherwise,} \end{cases} \tag{21}$$

where the formula for $A$ is given later. The noise construction (21) implies that stochasticity appears only at the first iteration of M-AdaGrad, and then it only affects the stepsizes. Therefore,

$$x_1 = x_0 - \frac{\gamma}{b_0}m_0,$$

where $b_0 = \sqrt{b_{-1}^2 + (\nu - \sigma\xi_0)^2}$ and $m_0 = (1 - \beta_1)(\nu - \sigma\xi_0)$. Moreover, $x_1$ can be bounded in the following way

$$x_0 + \gamma > x_1 > x_0 - \gamma.$$

Choosing $x_0$ in such a way that $x_0 - 2\gamma > \nu$, we apply Corollary 1 and get that the algorithm needs to make at least

$$K_0 = \frac{\left(x_1 - \nu - \gamma\left(1 + \frac{\max\{m_0, 0\}}{\nu}\right)\right)\sqrt{a_1}}{\gamma\left(1 + \frac{\max\{m_0, 0\}}{\nu}\right)}$$

iterations to reach $\varepsilon$-accuracy, where $a_1 = \frac{b_0^2}{\nu^2}$ and $\varepsilon = \frac{\nu^2}{2}$. Let us specify that this estimate depends on the stochasticity at the first iteration, i.e., the bound on the number of iterations is random. Consequently, if M-AdaGrad achieves $\varepsilon$-solution after $K$ steps, we should have $K \geq K_0$. Therefore, $\mathbb{P}\{K \geq K_0\} \geq \mathbb{P}\{f(x_K) - f(x^*) \leq \varepsilon\}$ and we want to estimate $K$ such that

$$\mathbb{P}\{K_0 \leq K\} \geq 1 - \delta.$$

Bounding the left-hand side,

$$\mathbb{P}\{K_0 \leq K\} = \mathbb{P}\{K_0 \leq K | \xi_0 = -A\}\mathbb{P}\{\xi_0 = -A\} + \mathbb{P}\{K_0 \leq K | \xi_0 \neq -A\}\mathbb{P}\{\xi_0 \neq -A\}$$

$$\leq \mathbb{P}\left\{\frac{\left(x_1 - \nu - \gamma\left(1 + \frac{\max\{m_0, 0\}}{\nu}\right)\right)\sqrt{a_1}}{\gamma\left(1 + \frac{\max\{m_0, 0\}}{\nu}\right)} \leq K \middle| \xi_0 \neq -A\right\}\mathbb{P}\{\xi_0 \neq -A\}$$

$$+ \mathbb{P}\{\xi_0 = -A\}.$$

If we choose $R = x_0 - \nu - 3\gamma$ and $A = A = \frac{\frac{\gamma K \nu}{R} + \nu}{\sigma}$, then $m_0$ can be bounded as

$$m_0 \leq \nu,$$

where we substitute $\xi_0 = 0, A$. Therefore, we get

$$\mathbb{P}\{K_0 \leq K\} \leq \mathbb{P}\left\{\frac{\left(x_1 - \nu - \gamma\left(1 + \frac{\max\{m_0, 0\}}{\nu}\right)\right)\sqrt{a_1}}{\gamma\left(1 + \frac{\max\{m_0, 0\}}{\nu}\right)} \leq K \middle| \xi_0 \neq -A\right\}\mathbb{P}\{\xi_0 \neq -A\}$$

$$+ \mathbb{P}\{\xi_0 = -A\}$$

$$\leq \mathbb{P}\left\{\frac{(x_0 - \nu - 3\gamma)\sqrt{a_1}}{2\gamma} \leq K \middle| \xi_0 \neq -A\right\}\mathbb{P}\{\xi_0 \neq -A\} + \mathbb{P}\{\xi_0 = -A\}$$

$$\leq \mathbb{P}\left\{\frac{R\sqrt{a_1}}{2\gamma} \leq K \middle| \xi_0 \neq -A\right\}\mathbb{P}\{\xi_0 \neq -A\} + \mathbb{P}\{\xi_0 = -A\}.$$

We notice that condition $K \geq \frac{b_{-1}R}{\nu\gamma}$ is necessary, since otherwise it leads to the contradiction. Indeed, it is enough to choose $\delta = \frac{1}{4}$:

$$\frac{3}{4} = 1 - \delta \leq \mathbb{P}\{\xi_0 = -A\} = \frac{1}{2A^2} \leq \frac{1}{2}.$$

Substituting the analytical form of $b_0$, with $K \geq \frac{b_{-1}R}{\nu\gamma}$ we get

$$\mathbb{P}\{K_0 \leq K\} \leq \mathbb{P}\left\{b_{-1}^2 + (\nu - \sigma\xi_0)^2 \leq \frac{\gamma^2 K^2 \nu^2}{R^2} \middle| \xi_0 \neq -A\right\}\mathbb{P}\{\xi_0 \neq -A\} + \mathbb{P}\{\xi_0 = -A\}$$

$$= \mathbb{P}\left\{|\sigma\xi_0 - \nu| \leq \sqrt{\frac{\gamma^2 K^2 \nu^2}{R^2} - b_{-1}^2} \middle| \xi_0 \neq -A\right\}\mathbb{P}\{\xi_0 \neq -A\} + \mathbb{P}\{\xi_0 = -A\}$$

$$\leq \mathbb{P}\left\{\sigma|\xi_0| \leq \sqrt{\frac{\gamma^2 K^2 \nu^2}{R^2} - b_{-1}^2} + \nu \middle| \xi_0 \neq -A\right\}\mathbb{P}\{\xi_0 \neq -A\} + \mathbb{P}\{\xi_0 = -A\}$$

Therefore, $\mathbb{P}\{K_0 \leq K\} \geq 1 - \delta$ implies

$$\mathbb{P}\left\{\sigma|\xi_0| \leq \sqrt{\frac{\gamma^2 K^2 \nu^2}{R^2} - b_{-1}^2} + \nu \middle| \xi_0 \neq -A\right\}\mathbb{P}\{\xi_0 \neq -A\} + \mathbb{P}\{\xi_0 = -A\} \geq 1 - \delta.$$

Consequently, since $A = \frac{\frac{\gamma K \nu}{R} + \nu}{\sigma}$, the first probability in the inequality above is equal to $1 - \frac{1}{A^2}$, since the only $\xi_0 = 0$ satisfies the condition on random variable. Hence, we have

$$\left(1 - \frac{1}{A^2}\right)\left(1 - \frac{1}{2A^2}\right) + \frac{1}{2A^2} \geq 1 - \delta.$$

Denoting $\frac{1}{2A^2}$ as $x$, one can obtain

$$2x^2 - 2x + \delta \geq 0.$$

In the case $\delta \geq \frac{1}{2}$ we use that $\frac{1}{2A^2} \leq \frac{1}{2} \leq \delta$. For the $\delta < \frac{1}{2}$ we solve the quadratic inequality and get

$$\frac{1}{2A^2} \leq \frac{\delta}{1 + \sqrt{1 - 2\delta}} \leq \delta.$$

Consequently,

$$\frac{1}{A} = \frac{\sigma}{\frac{\gamma K \nu}{R} + \nu} \leq \sqrt{2\delta}.$$

Therefore,

$$K \geq \frac{R}{\gamma}\left(\frac{\sigma}{\nu\sqrt{2\delta}} - 1\right),$$

which concludes the proof since $\sigma/\sqrt{\varepsilon\delta} \geq 4$ and $\nu = \sqrt{2\varepsilon}$. $\qquad\square$

### B.2 FAILURE OF M-AdaGradD

Similarly to the case of M-AdaGrad, we start by obtaining the analytic form of iterations of the deterministic M-AdaGradD in the following lemma.

**Lemma 7.** *Suppose that starting point $x_0$ is such that $x_0 > 0$. If after $T$ iterations of deterministic M-AdaGradD we have $|x_t| > \nu$ and $x_t > 0$ for all $t = \overline{1, T-1}$ with , then*

$$x_T = x_0 - \gamma\nu \sum_{t=0}^{T-1} \frac{1 - \beta_1^{t+1}}{\sqrt{b_0^2 + t\nu^2}}.$$

*Proof.* The proof is similar to the proof of Lemma 5. Since $x_t > \nu$, the gradient at point $x_t$ is equal to $\nu$. Substituting that into the iteration of M-AdaGradD for each $t$, we finish the proof. $\qquad\square$

Now, let us estimate the interval where $x_T$ lies.

**Lemma 8.** *Let the conditions of Lemma 7 hold. Then, we have*

$$x_0 - \gamma\left(\frac{1}{\sqrt{a_0}} + 2\sqrt{a_0 + T - 1} - 2\sqrt{a_0}\right) \leq x_T \leq x_0 - \gamma(1 - \beta_1)\left(2\sqrt{a_0 + T} - 2\sqrt{a_0}\right),$$

*where $a_0 = \frac{b_0^2}{\nu^2}$.*

*Proof.* Let us start with Lemma 7:

$$x_T = x_0 - \gamma \sum_{t=0}^{T-1} \frac{1 - \beta_1^{t+1}}{\sqrt{a_0 + t}},$$

where $a_0 = \frac{b_0^2}{\nu^2}$. Next, we bound the second term in the following way:

$$\sum_{t=0}^{T-1} \frac{1 - \beta_1^{t+1}}{\sqrt{a_0 + t}} \geq (1 - \beta_1) \int_{a_0}^{a_0 + T} \frac{1}{\sqrt{x}}\, dx = (1 - \beta_1)(2\sqrt{a_0 + T} - 2\sqrt{a_0}), \tag{22}$$

$$\sum_{t=0}^{T-1} \frac{1 - \beta_1^{t+1}}{\sqrt{a_0 + t}} \leq \frac{1}{\sqrt{a_0}} + \int_{a_0}^{a_0 + T - 1} \frac{1}{\sqrt{x}}\, dx = \frac{1}{\sqrt{a_0}} + 2\sqrt{a_0 + T - 1} - 2\sqrt{a_0}. \tag{23}$$

Combining (22) and (23), we have the final result. $\qquad\square$

**Corollary 2.** *If $x_0 - \gamma > \nu > 0$, $b_0 \geq \nu$ and*

$$T < \frac{(x_0 - \nu - \gamma)^2 + 4\gamma(x_0 - \nu - \gamma)\sqrt{a_0}}{4\gamma^2} + 2,$$

*then $x_T > \nu$ for deterministic* M-AdaGradD. *Conversely, the case $|x_T| \leq \nu$ implies that*

$$T \geq \frac{(x_0 - \nu - \gamma)^2 + 4\gamma(x_0 - \nu - \gamma)\sqrt{a_0}}{4\gamma^2} + 2.$$

*Proof.* The proof is the same as for Corollary 1. $\square$

**Theorem 6.** *For any $\varepsilon, \delta \in (0, 1)$, $\sigma > 0$, there exists convex $L$-smooth minimization problem (8) and stochastic gradient oracle such that Assumption 1 holds with $\alpha = 2$ and the iterates produced by* M-AdaGradD *after $K$ steps with stepsize $\gamma$ and starting point $x_0$ such that $R := x_0 - \sqrt{2\varepsilon} - \gamma > 0$, $b_0 > \nu$ and $(1-\beta_1)\sigma R/\varepsilon\sqrt{\delta} \geq 16b_0^2$ satisfy the following implication*

$$\mathbb{P}\{f(x_K) - f(x^*) \geq \varepsilon\} \leq \delta \quad \Longrightarrow \quad K = \Omega\left(\frac{\sigma R}{\varepsilon\sqrt{\delta}}\right), \tag{24}$$

*i.e., the high-probability complexity of* M-AdaGradD *has inverse-power dependence on $\delta$.*

*Proof.* The overall idea of the proof resembles the one for Theorem 5 – we combine the lower bound for the number of iterations from Corollary 2 with the specific choice of stochasticity. Nevertheless, to prove this theorem, we construct the adversarial noise in another way. More precisely, we consider the following stochastic gradient

$$g_k = \nabla f(x_k) - \sigma\xi_k,$$

where

$$\xi_k = \begin{cases} 0, & \text{if } k < K - 1 \text{ or } |\hat{x}_K| > \nu, \\ \begin{cases} -A_k, & \text{with probability } \frac{1}{2A_k^2} \\ 0, & \text{with probability } 1 - \frac{1}{A_k^2} \\ A_k, & \text{with probability } \frac{1}{2A_k^2} \end{cases} & \text{otherwise,} \end{cases} \tag{25}$$

where $\hat{x}_K$ is the result of deterministic M-AdaGradD after $K$ iterations and $A_k = \max\left\{1, \frac{2\nu b_k}{(1-\beta_1)\gamma\sigma}\right\}$. What is more, $\mathbb{E}[\xi_k] = 0$ and $\mathbb{E}[\xi_k^2] \leq 1$ by the construction. Therefore, the stochastic gradient satisfies the Assumption 1 with $\alpha = 2$.

We want to prove that $\mathbb{P}\{f(x_K) - f(x^*) > \varepsilon\} \leq \delta$. For $\delta < 1$, this implies that $|\hat{x}_K| \leq \nu$ with $\varepsilon = \frac{\nu^2}{2}$. Indeed, assuming the contrary, the noise is equal to 0 for each iteration by the construction, meaning that

$$\mathbb{P}\{f(x_K) - f(x^*) > \varepsilon\} = \mathbb{P}\{f(\hat{x}_K) - f(x^*) > \varepsilon\} = \mathbb{P}\{|\hat{x}_K| > \nu\} = 1 > \delta.$$

As a result, $|\hat{x}_K| \leq \nu$ and, applying Corollary 2, we obtain

$$K \geq \frac{(x_0 - \nu - \gamma)^2 + 4\gamma(x_0 - \nu - \gamma)\sqrt{a_0}}{4\gamma^2} + 2.$$

What is more, $x_K$ can be written as

$$x_K = \hat{x}_{K-1} - \frac{\gamma}{b_{K-1}}m_{K-1} = \hat{x}_K + \frac{(1-\beta_1)\gamma\sigma\xi_{K-1}}{b_{K-1}}.$$

Hence,

$$\mathbb{P}\{f(x_K) - f(x^*) \geq \varepsilon\} = \mathbb{P}\{|x_K| \geq \nu\} = \mathbb{P}\left\{\left|\hat{x}_K + \frac{(1-\beta_1)\gamma\sigma\xi_{K-1}}{b_{K-1}}\right| \geq \nu\right\}$$

$$\geq \mathbb{P}\left\{\left|\frac{(1-\beta_1)\gamma\sigma\xi_{K-1}}{b_{K-1}}\right| \geq \nu + \hat{x}_K\right\} \geq \mathbb{P}\left\{\left|\frac{(1-\beta_1)\gamma\sigma\xi_{K-1}}{b_{K-1}}\right| \geq 2\nu\right\}$$

$$= \mathbb{P}\left\{|\xi_{K-1}| \geq \frac{2\nu b_{K-1}}{(1-\beta_1)\gamma\sigma}\right\}.$$

If $\max\left\{1, \frac{2\nu b_{K-1}}{(1-\beta_1)\gamma\sigma}\right\} = 1$, then

$$\delta \geq \mathbb{P}\left\{f(x_K) - f(x^*) \geq \varepsilon\right\} \geq \mathbb{P}\left\{|\xi_{K-1}| \geq \frac{2\nu b_{K-1}}{(1-\beta_1)\gamma\sigma}\right\} = 1,$$

which leads us to the contradiction. Therefore $\max\left\{1, \frac{2\nu b_{K-1}}{(1-\beta_1)\gamma\sigma}\right\} = \frac{2\nu b_{K-1}}{(1-\beta_1)\gamma\sigma}$, and

$$\delta \geq \mathbb{P}\left\{f(x_K) - f(x^*) \geq \varepsilon\right\} \geq \mathbb{P}\left\{|\xi_{K-1}| \geq \frac{2\nu b_{K-1}}{(1-\beta_1)\gamma\sigma}\right\} = \frac{1}{A_{K-1}^2} = \frac{(1-\beta_1)^2\gamma^2\sigma^2}{4\nu^2 b_{K-1}^2},$$

where we used that $A_{K-1} = \max\left\{1, \frac{2\nu b_{K-1}}{(1-\beta_1)\gamma\sigma}\right\}$ and the noise structure. Consequently, $\gamma \leq \frac{2\nu b_{K-1}\sqrt{\delta}}{(1-\beta_1)\sigma}$. What is more, $b_{K-1}$ can be bounded as

$$b_{K-1} \leq \sqrt{b_0^2 + K\nu^2}$$

since the gradient of $f$ is uniformly bounded by $\nu$. Hence, we obtain

$$K \geq \frac{(x_0 - \nu - \gamma)^2}{4\gamma^2} + \frac{4(x_0 - \nu - \gamma)\sqrt{a_0}}{4\gamma} \geq \frac{(x_0 - \nu - \gamma)^2}{4\gamma^2}$$
$$\geq \frac{(1-\beta_1)^2(x_0 - \nu - \gamma)^2\sigma^2}{16\nu^2(b_0^2 + K\nu^2)\delta}.$$

Multiplying both sides by $\nu^2(b_0^2 + K\nu^2)$, we get

$$(b_0^2 + K\nu^2)^2 \geq \nu^2 K(b_0^2 + K\nu^2) \geq \frac{(1-\beta_1)^2(x_0 - \nu - \gamma)^2\sigma^2}{16\delta},$$

implying that

$$K \geq \frac{(1-\beta_1)\sigma R}{4\nu^2\sqrt{\delta}} - b_0^2 = \frac{(1-\beta_1)\sigma R}{8\varepsilon\sqrt{\delta}} - b_0^2 \geq \frac{(1-\beta_1)\sigma R}{16\varepsilon\sqrt{\delta}},$$

which finishes the proof.

$\square$

### B.3 FAILURE OF Adam

Similarly to the case of M-AdaGrad, we start by obtaining the analytical form of iterations of the deterministic Adam in the following lemma.

**Lemma 9.** *Suppose that the starting point $x_0$ is such that $x_0 > 0$. If after $T$ iterations of deterministic Adam with initial momentum $m_{-1}$ we have $|x_t| > \nu$ and $x_t > 0$ for all $t = \overline{1, T-1}$, then*

$$x_T = x_0 - \gamma \sum_{t=0}^{T-1} \frac{\beta_1^{t+1} m_{-1} + \left(1 - \beta_1^{t+1}\right)\nu}{\sqrt{\beta_2^{t+1} b_{-1}^2 + \left(1 - \beta_2^{t+1}\right)\nu^2}}.$$

*Proof.* Since $|x_t| > \nu$ and $x_t$ is positive, the gradient at $x_t$ is equal to $\nu$. Hence, by substituting the gradient into the algorithm, we get the final result. $\square$

The above lemma relies on the condition that $|x_t| > \nu$ and $x_t > 0$ for all $t = \overline{1, T-1}$. For any $\gamma, b_{-1}$ and $T$ this condition can be achieved if we choose sufficiently small $\nu$.

Next, we estimate the interval where $x_T$ lies.

**Lemma 10.** *Let the conditions of Lemma 9 hold. Then, if $\beta_2 = 1 - 1/K$, where $K$ is the total number of iterations of deterministic Adam, we have*

$$x_0 - \frac{2\gamma(\max\{m_{-1}, 0\} + \nu)T}{b_{-1}} \leq x_T \leq x_0 - \frac{\gamma((1-\beta_1)\nu + \beta_1 \min\{m_{-1}, 0\})T}{\sqrt{b_{-1}^2 + \nu^2}}.$$

*Proof.* From Lemma 9 we have:

$$x_T = x_0 - \gamma \sum_{t=0}^{T-1} \frac{\beta_1^{t+1} m_{-1} + \left(1 - \beta_1^{t+1}\right)\nu}{\sqrt{\beta_2^{t+1} b_{-1}^2 + \left(1 - \beta_2^{t+1}\right)\nu^2}}.$$

Next, we bound the second term in the inequality above in the following way:

$$\sum_{t=0}^{T-1} \frac{\beta_1^{t+1} m_{-1} + \left(1 - \beta_1^{t+1}\right)\nu}{\sqrt{\beta_2^{t+1} b_{-1}^2 + \left(1 - \beta_2^{t+1}\right)\nu^2}} \leq \frac{2T(\max\{m_{-1}, 0\} + \nu)}{b_{-1}}, \tag{26}$$

$$\sum_{t=0}^{T-1} \frac{\beta_1^{t+1} m_{-1} + \left(1 - \beta_1^{t+1}\right)\nu}{\sqrt{\beta_2^{t+1} b_{-1}^2 + \left(1 - \beta_2^{t+1}\right)\nu^2}} \geq \frac{((1-\beta_1)\nu + \beta_1 \min\{m_{-1}, 0\})T}{\sqrt{b_{-1}^2 + \nu^2}}, \tag{27}$$

where we use the fact that with $K \geq 2$ next inequalities hold

$$1 \geq \beta_2^k = (1 - 1/K)^k \geq (1 - 1/K)^K \geq 1/4,$$

$$0 \leq 1 - \beta_2^k \leq 3/4 \leq 1.$$

Combining (26) and (27), we get the final result. $\qquad\square$

**Corollary 3.** *If $x_0 > \nu > 0$ and*

$$T < \frac{(x_0 - \nu)b_{-1}}{2\gamma(\max\{m_{-1}, 0\} + \nu)},$$

*then $x_T > \nu$ for deterministic* Adam. *Alternatively, $|x_T| \leq \nu$ implies that*

$$T \geq \frac{(x_0 - \nu)b_{-1}}{2\gamma(\max\{m_{-1}, 0\} + \nu)}.$$

*Proof.* Let us note that

$$\nu < x_0 - \frac{2\gamma(\max\{m_{-1}, 0\} + \nu)T}{b_{-1}}$$

is equivalent to

$$T < \frac{(x_0 - \nu)b_{-1}}{2\gamma(\max\{m_{-1}, 0\} + \nu)}.$$

The second part of the corollary is just a negation of the implication stated in the first part of the corollary. $\qquad\square$

**Theorem 7.** *For any $\varepsilon, \delta \in (0, 1), \sigma > 0$, there exists convex $L$-smooth minimization problem (8) and stochastic gradient oracle such that Assumption 1 holds with $\alpha = 2$ and the iterates produced by* Adam *after $K$ steps with stepsize $\gamma$ and starting point $x_0$ such that $R := x_0 - \nu > 0$ and $x_0 - \gamma/\sqrt{1-\beta_2} - \nu > 0$ satisfy the following implication:*

$$\mathbb{P}\{f(x_K) - f(x^*) \geq \varepsilon\} \leq \delta \quad \implies \quad K = \Omega\left(\min\left\{\frac{\sigma^2}{\varepsilon\delta}, \frac{b_{-1}R}{\sqrt{\varepsilon}\gamma} + \left(\frac{\sigma R}{\gamma\sqrt{\varepsilon\delta}}\right)^{2/3}\right\}\right), \tag{28}$$

*i.e., the high-probability complexity of* Adam *has inverse-power dependence on $\delta$.*

*Proof.* The main idea is quite similar to the proof of Theorem 5. We introduce the noise in the stochastic gradient in the following way

$$g_k = \nabla f(x_k) - \sigma\xi_k,$$

where

$$\xi_k = \begin{cases} 0, & \text{for } k > 0, \\ \begin{cases} -A, & \text{with probability } \frac{1}{2A^2} \\ 0, & \text{with probability } 1 - \frac{1}{A^2} \\ A, & \text{with probability } \frac{1}{2A^2} \end{cases} & \text{otherwise,} \end{cases} \tag{29}$$

where the formula for $A$ is given later. The noise construction (29) implies that stochasticity appears only at the first iteration of Adam, and then it only affects the stepsizes. Therefore,

$$x_1 = x_0 - \frac{\gamma}{b_0} m_0,$$

where $b_0 = \sqrt{\beta_2 b_{-1}^2 + (1 - \beta_2)(\nu - \sigma\xi_0)^2}$ and $m_0 = (1 - \beta_1)(\nu - \sigma\xi_0)$. Moreover, $x_1$ can be bounded in the following way

$$x_0 + \gamma/\sqrt{1-\beta_2} > x_1 > x_0 - \gamma/\sqrt{1-\beta_2}.$$

Choosing $x_0$ in such a way that $x_0 - \gamma/\sqrt{1-\beta_2} > \nu$, we apply Corollary 3 and get that the algorithm needs to make at least

$$K_0 = \frac{(x_1 - \nu)b_0}{2\gamma(\max\{m_0, 0\} + \nu)}$$

iterations to reach $\varepsilon$-accuracy, where $\varepsilon = \frac{\nu^2}{2}$. Let us specify that this estimate depends on the stochasticity at the first iteration, i.e., the bound on the number of iterations is random. Consequently, if Adam achieves $\varepsilon$-solution after $K$ steps, we should have $K \geq K_0$. Therefore, $\mathbb{P}\{K \geq K_0\} \geq \mathbb{P}\{f(x_K) - f(x^*) \leq \varepsilon\}$ and we want to estimate $K$ such that

$$\mathbb{P}\{K_0 \leq K\} \geq 1 - \delta.$$

Bounding the left-hand side,

$$\mathbb{P}\{K_0 \leq K\} = \mathbb{P}\{K_0 \leq K | \xi_0 = -A\}\mathbb{P}\{\xi_0 = -A\} + \mathbb{P}\{K_0 \leq K | \xi_0 \neq -A\}\mathbb{P}\{\xi_0 \neq -A\}$$

$$\leq \mathbb{P}\left\{\frac{(x_1 - \nu)b_0}{2\gamma(\max\{m_0, 0\} + \nu)} \leq K \Big| \xi_0 \neq -A\right\}\mathbb{P}\{\xi_0 \neq -A\} + \mathbb{P}\{\xi_0 = -A\}$$

$$= \mathbb{P}\left\{\frac{(x_0 - \gamma\frac{m_0}{b_0} - \nu)b_0}{2\gamma(\max\{m_0, 0\} + \nu)} \leq K \Big| \xi_0 \neq -A\right\}\mathbb{P}\{\xi_0 \neq -A\} + \mathbb{P}\{\xi_0 = -A\}.$$

Moreover, according to the analytical form of $m_0$, if $\xi_0 \neq -A$, then

$$m_0 \leq \nu.$$

Therefore,

$$\mathbb{P}\{K_0 \leq K\} \leq \mathbb{P}\left\{\frac{(x_0 - \nu)b_0 - 4\gamma\nu}{4\gamma\nu} \leq K \Big| \xi_0 \neq -A\right\}\mathbb{P}\{\xi_0 \neq -A\} + \mathbb{P}\{\xi_0 = -A\}$$

$$= \mathbb{P}\left\{\frac{Rb_0}{4\gamma\nu} \leq K + 1 \Big| \xi_0 \neq -A\right\}\mathbb{P}\{\xi_0 \neq -A\} + \mathbb{P}\{\xi_0 = -A\},$$

where $R = x_0 - \nu$. Substituting the analytical form of $b_0$, we get

$$\mathbb{P}\{K_0 \leq K\} \leq \mathbb{P}\left\{\beta_2 b_{-1}^2 + (1 - \beta_2)(\nu - \sigma\xi_0)^2 \leq \frac{16\gamma^2(K+1)^2\nu^2}{R^2} \Big| \xi_0 \neq -A\right\}\mathbb{P}\{\xi_0 \neq -A\}$$

$$+ \mathbb{P}\{\xi_0 = -A\}$$

We notice that condition $K + 1 \geq \frac{\sqrt{\beta_2} b_{-1} R}{\nu\gamma}$ is necessary for the convergence because of the similar idea from the proof of Theorem 5. Therefore, we have $K + 1 \geq \frac{\sqrt{\beta_2} b_{-1} R}{\nu\gamma}$ and can continue the

derivation as follows:

$$\mathbb{P}\{K_0 \le K\} \le \mathbb{P}\left\{ |\sigma\xi_0 - \nu| \le \frac{\sqrt{\frac{\gamma^2(K+1)^2\nu^2}{R^2} - \beta_2 b_{-1}^2}}{\sqrt{1-\beta_2}} \middle| \xi_0 \ne -A \right\} \mathbb{P}\{\xi_0 \ne -A\}$$

$$+ \mathbb{P}\{\xi_0 = -A\}$$

$$\le \mathbb{P}\left\{ \sigma|\xi_0| \le \frac{\sqrt{\frac{\gamma^2(K+1)^2\nu^2}{R^2} - \beta_2 b_{-1}^2}}{\sqrt{1-\beta_2}} + \nu \middle| \xi_0 \ne -A \right\} \mathbb{P}\{\xi_0 \ne -A\}$$

$$+ \mathbb{P}\{\xi_0 = -A\}.$$

Therefore, $\mathbb{P}\{K_0 \le K\} \ge 1 - \delta$ implies

$$\mathbb{P}\left\{ \sigma|\xi_0| \le \frac{\sqrt{\frac{\gamma^2(K+1)^2\nu^2}{R^2} - \beta_2 b_{-1}^2}}{\sqrt{1-\beta_2}} + \nu \middle| \xi_0 \ne -A \right\} \mathbb{P}\{\xi_0 \ne -A\} + \mathbb{P}\{\xi_0 = -A\} \ge 1 - \delta.$$

Consequently, if we choose $A = \frac{\gamma\nu(K+1)}{\sqrt{1-\beta_2}R\sigma} + \frac{\nu}{\sqrt{1-\beta_2}\sigma}$, then the only realization of the random variable $\xi_0$ at which the inequality in the first probability is satisfied is 0. Hence, we have the quadratic inequality:

$$\left(1 - \frac{1}{A^2}\right)\left(1 - \frac{1}{2A^2}\right) + \frac{1}{2A^2} \ge 1 - \delta.$$

Applying the idea similar to the proof of Theorem 5, we obtain

$$\frac{1}{A} = \frac{\sqrt{1-\beta_2}\sigma}{\frac{\gamma(K+1)\nu}{R} + \nu} \le \sqrt{2\delta}.$$

Therefore,

$$K + 1 \ge \frac{R}{\gamma}\left(\frac{\sqrt{1-\beta_2}\sigma}{\nu\sqrt{\delta}} - 1\right),$$

Applying the fact that $1 - \beta_2 = 1/K$, we conclude the proof since $\sqrt{1-\beta_2}\sigma/\sqrt{\varepsilon\delta} \ge 4$ (otherwise $K \ge \sigma^2/16\varepsilon\delta$) and $\nu = \sqrt{2\varepsilon}$. □

### B.4 FAILURE OF AdamD

We follow the idea for previous proofs and start by obtaining the analytical form of iterations of the deterministic AdamD in the following lemma.

**Lemma 11.** *Suppose that the starting point $x_0$ is such that $x_0 > 0$. If after $T$ iterations of deterministic AdamD we have $|x_t| > \nu$ and $x_t > 0$ for all $t = \overline{1, T-1}$, then*

$$x_T = x_0 - \gamma\nu \sum_{t=0}^{T-1} \frac{1 - \beta_1^{t+1}}{\sqrt{\beta_2^t b_0^2 + (1-\beta_2^t)\nu^2}}.$$

*Proof.* Since $|x_t| > \nu$ and $x_t$ is positive, the gradient at $x_t$ is equal to $\nu$. Hence, by substituting the gradient into the algorithm, we get the final result. □

The above lemma relies on the condition that $|x_t| > \nu$ and $x_t > 0$ for all $t = \overline{1, T-1}$. For any $\gamma, b_0$ and $T$ this condition can be achieved if we choose sufficiently small $\nu$.

Next, we estimate the interval where $x_T$ lies.

**Lemma 12.** *Let the conditions of Lemma 11 hold. Then, if $\beta_2 = 1 - 1/K$, where $K$ is the total number of iterations of deterministic AdamD, we have*

$$x_0 - \frac{2\gamma\nu T}{b_0} \le x_T \le x_0 - \frac{\gamma\nu(1-\beta_1)T}{\sqrt{b_0^2 + \nu^2}}.$$

*Proof.* From Lemma 11 we have:

$$x_T = x_0 - \gamma\nu \sum_{t=0}^{T-1} \frac{1 - \beta_1^{t+1}}{\sqrt{\beta_2^t b_0^2 + (1 - \beta_2^t)\nu^2}}.$$

Next, we bound the second term in the inequality above in the following way:

$$\sum_{t=0}^{T-1} \frac{1 - \beta_1^{t+1}}{\sqrt{\beta_2^t b_0^2 + (1 - \beta_2^t)\nu^2}} \leq \frac{2T}{b_0}, \tag{30}$$

$$\sum_{t=0}^{T-1} \frac{1 - \beta_1^{t+1}}{\sqrt{\beta_2^t b_0^2 + (1 - \beta_2^t)\nu^2}} \geq \frac{(1 - \beta_1)T}{\sqrt{b_0^2 + \nu^2}}, \tag{31}$$

where we use the fact that with $K \geq 2$ next inequalities hold

$$1 \geq \beta_2^k = (1 - 1/K)^k \geq (1 - 1/K)^K \geq 1/4,$$

$$0 \leq 1 - \beta_2^k \leq 3/4 \leq 1.$$

Combining (30) and (31), we get the final result. $\qquad\square$

**Corollary 4.** *If $x_0 > \nu > 0$ and*

$$T < \frac{(x_0 - \nu)b_0}{2\gamma\nu},$$

*then $x_T > \nu$ for deterministic* AdamD. *Alternatively, $|x_T| \leq \nu$ implies that*

$$T \geq \frac{(x_0 - \nu)b_0}{2\gamma\nu}.$$

*Proof.* The proof is the same as for Corollary 3. $\qquad\square$

**Theorem 8.** *For any $\varepsilon, \delta \in (0, 1)$, $\sigma > 0$, there exists convex L-smooth minimization problem (8) and stochastic gradient oracle such that Assumption 1 holds with $\alpha = 2$ and the iterates produced by* AdamD *after $K$ steps with stepsize $\gamma$ and starting point $x_0$ such that $R := x_0 - \nu > 0$, $b_0 > \nu$ and $\sigma R/\varepsilon\sqrt{\delta} \geq 16b_0^2$ satisfy the following implication*

$$\mathbb{P}\{f(x_K) - f(x^*) \geq \varepsilon\} \leq \delta \quad \implies \quad K = \Omega\left(\frac{\sigma R}{\varepsilon\sqrt{\delta}}\right), \tag{32}$$

*i.e., the high-probability complexity of* AdamD *has inverse-power dependence on $\delta$.*

*Proof.* The overall idea of the proof resembles the one for Theorem 7 – we combine the lower bound for the number of iterations from Corollary 4 with the specific choice of stochasticity. Nevertheless, to prove this theorem, we construct the adversarial noise in another way. More precisely, we consider the following stochastic gradient

$$g_k = \nabla f(x_k) - \sigma\xi_k,$$

where

$$\xi_k = \begin{cases} 0, & \text{if } k < K - 1 \text{ or } |\hat{x}_K| > \nu, \\ \begin{cases} -A_k, & \text{with probability } \frac{1}{2A_k^2} \\ 0, & \text{with probability } 1 - \frac{1}{A_k^2} \\ A_k, & \text{with probability } \frac{1}{2A_k^2} \end{cases} & \text{otherwise,} \end{cases} \tag{33}$$

where $\hat{x}_K$ is the result of deterministic AdamD after $K$ iterations and $A_k = \max\left\{1, \frac{2\nu b_k}{(1 - \beta_1)\gamma\sigma}\right\}$. What is more, $\mathbb{E}[\xi_k] = 0$ and $\mathbb{E}[\xi_k^2] \leq 1$ by the construction. Therefore, the stochastic gradient satisfies the Assumption 1 with $\alpha = 2$.

We want to prove that $\mathbb{P}\{f(x_K) - f(x^*) > \varepsilon\} \leq \delta$. For $\delta < 1$, this implies that $|\hat{x}_K| \leq \nu$ with $\varepsilon = \frac{\nu^2}{2}$. Indeed, assuming the contrary, the noise is equal to $0$ for each iteration by the construction, meaning that

$$\mathbb{P}\left\{f(x_K) - f(x^*) > \varepsilon\right\} = \mathbb{P}\left\{f(\hat{x}_K) - f(x^*) > \varepsilon\right\} = \mathbb{P}\left\{|\hat{x}_K| > \nu\right\} = 1 > \delta.$$

As a result, $|\hat{x}_K| \leq \nu$ and, applying Corollary 4, we obtain

$$K \geq \frac{(x_0 - \nu)b_0}{2\gamma\nu}.$$

What is more, $x_K$ can be written as

$$x_K = \hat{x}_{K-1} - \frac{\gamma}{b_{K-1}}m_{K-1} = \hat{x}_K + \frac{(1 - \beta_1)\gamma\sigma\xi_{K-1}}{b_{K-1}}.$$

Hence,

$$\mathbb{P}\left\{f(x_K) - f(x^*) \geq \varepsilon\right\} = \mathbb{P}\left\{|x_K| \geq \nu\right\} = \mathbb{P}\left\{\left|\hat{x}_K + \frac{(1 - \beta_1)\gamma\sigma\xi_{K-1}}{b_{K-1}}\right| \geq \nu\right\}$$

$$\geq \mathbb{P}\left\{\left|\frac{(1 - \beta_1)\gamma\sigma\xi_{K-1}}{b_{K-1}}\right| \geq \nu + \hat{x}_K\right\} \geq \mathbb{P}\left\{\left|\frac{(1 - \beta_1)\gamma\sigma\xi_{K-1}}{b_{K-1}}\right| \geq 2\nu\right\}$$

$$= \mathbb{P}\left\{|\xi_{K-1}| \geq \frac{2\nu b_{K-1}}{(1 - \beta_1)\gamma\sigma}\right\}.$$

If $\max\left\{1, \frac{2\nu b_{K-1}}{(1 - \beta_1)\gamma\sigma}\right\} = 1$, then

$$\delta \geq \mathbb{P}\left\{f(x_K) - f(x^*) \geq \varepsilon\right\} \geq \mathbb{P}\left\{|\xi_{K-1}| \geq \frac{2\nu b_{K-1}}{(1 - \beta_1)\gamma\sigma}\right\} = 1,$$

which leads us to the contradiction. Therefore $\max\left\{1, \frac{2\nu b_{K-1}}{\gamma\sigma}\right\} = \frac{2\nu b_{K-1}}{(1 - \beta_1)\gamma\sigma}$, and

$$\delta \geq \mathbb{P}\left\{f(x_K) - f(x^*) \geq \varepsilon\right\} \geq \mathbb{P}\left\{|\xi_{K-1}| \geq \frac{2\nu b_{K-1}}{(1 - \beta_1)\gamma\sigma}\right\} = \frac{1}{A_{K-1}^2} = \frac{(1 - \beta_1)^2\gamma^2\sigma^2}{4\nu^2 b_{K-1}^2},$$

where we used that $A_{K-1} = \max\left\{1, \frac{2\nu b_{K-1}}{(1 - \beta_1)\gamma\sigma}\right\}$ and the noise structure. Consequently, $\gamma \leq \frac{2\nu b_{K-1}\sqrt{\delta}}{(1 - \beta_1)\sigma}$. What is more, $b_{K-1}$ can be bounded as

$$b_{K-1} \leq \sqrt{b_0^2 + \nu^2}$$

since the gradient of $f$ is uniformly bounded by $\nu$. Hence, we obtain with $b_0 \geq \nu$

$$K \geq \frac{(x_0 - \nu)b_0}{2\gamma\nu} \geq \frac{(1 - \beta_1)(x_0 - \nu)\sigma b_0}{4\sqrt{b_0^2 + \nu^2}\nu^2\sqrt{\delta}} \geq \frac{(1 - \beta_1)(x_0 - \nu)\sigma}{8\nu^2\sqrt{\delta}} = \frac{(1 - \beta_1)R\sigma}{16\varepsilon\sqrt{\delta}},$$

which finishes the proof.

$\square$

## C  MISSING PROOFS FROM SECTION 3

In this section, we provide missing proofs for Algorithm 2 in the convex and non-convex cases. For each case, the proof consists of two parts – descent lemma and main theorem. Moreover, for convenience of the proofs, we consider a reweighted version of Algorithm 2 summarized in Algorithm 3, which has an additional parameter $\eta > 0$ appearing in the update rule for $b_t$. However, Algorithms 2 and 3 are equivalent: if we divide $b_t$ and $\gamma$ in Algorithm 3 by $\sqrt{\eta}$, the method reduces to Algorithm 2 but produces exactly the same points as before (given the same initialization and source of stochasticity, i.e., seed), since $\gamma/b_t$ remains unchanged.

---

**Algorithm 3** Reweighted Clip-Adam/Clip-AdamD and Clip-M-AdaGrad/Clip-M-AdaGradD

---

**Input:** Stepsize $\gamma > 0$, starting point $x_0 \in \mathbb{R}^d$, initial constant $b_{-1} > 0$ (for Adam and M-AdaGrad) or $b_0 > 0$ (for AdamD and M-AdaGradD), momentum parameters $\beta_1, \beta_2 \in [0,1]$, level of clipping $\lambda > 0$, reweighting parameter $\eta > 0$

1: Set $m_{-1} = 0$
2: **for** $t = 0, 1, \ldots$ **do**
3:     $m_t = \beta_1 m_{t-1} + (1 - \beta_1)\texttt{clip}\left(\nabla f_{\xi_t}(x_t), \lambda\right)$
4:     **if** no delay **then**
5:       $b_t = \begin{cases} \sqrt{\beta_2 b_{t-1}^2 + \eta(1-\beta_2)\|\texttt{clip}\left(\nabla f_{\xi_t}(x_t), \lambda\right)\|^2} & \text{for Clip-Adam} \\ \sqrt{b_{t-1}^2 + \eta\|\texttt{clip}\left(\nabla f_{\xi_t}(x_t), \lambda\right)\|^2} & \text{for Clip-M-AdaGrad} \end{cases}$
6:     **else**
7:       $b_{t+1} = \begin{cases} \sqrt{\beta_2 b_t^2 + \eta(1-\beta_2)\|\texttt{clip}\left(\nabla f_{\xi_t}(x_t), \lambda\right)\|^2} & \text{for Clip-AdamD} \\ \sqrt{b_t^2 + \eta\|\texttt{clip}\left(\nabla f_{\xi_t}(x_t), \lambda\right)\|^2} & \text{for Clip-M-AdaGradD} \end{cases}$
8:     **end if**
9:     $x_{t+1} = x_t - \frac{\gamma}{b_t}m_t$
10: **end for**

---

### C.1  TECHNICAL LEMMAS

Here we introduce technical lemmas for the future proofs.

**Lemma 13.** *Let the sequence $\{b_t\}_{t=0}$ is generated by Algorithm 3 in $K$ iterations. Then, for every $t, r$: $t \geq r$ we get*

$$b_t \geq c_m b_r,$$

*where the constant $c_m$ depends on the update rule for $b_t$. To be more precise, $c_m = 1$ for the Clip-M-AdaGrad/Clip-M-AdaGradD, and $c_m = 1/2$ for Clip-Adam/Clip-AdamD.*

*Proof.* The case of Clip-M-AdaGrad/Clip-M-AdaGradD is obvious since the sequence $\{b_t\}_{t=0}$ is non-decreasing. For the Clip-Adam/Clip-AdamD we obtain that

$$b_t^2 \geq \beta_2^{t-r} b_r^2 = \left(1 - \frac{1}{K}\right)^{t-r} b_r^2 \geq \left(1 - \frac{1}{K}\right)^K b_r^2 \geq \frac{1}{4}b_r^2,$$

where we, without loss of generality, assume that $K \geq 2$ and apply the analytical form of $\beta_2$ with fact that $g(K) = \left(1 - \frac{1}{K}\right)^K$ is increasing function. Taking the square root from both parts, we conclude the proof. $\square$

**Lemma 14.** *Let the sequence $\{m_t\}_{t=0}$ is generated by Algorithm 3 in $K$ iterations. Then, for every $0 \leq t \leq K - 1$ it holds that*

$$m_t = \sum_{k=0}^{t} \beta_1^{t-k}(1 - \beta_1)g_k.$$

*Moreover, $\|m_t\|^2$ can be bounded in the following way:*

$$\|m_t\|^2 \leq (1 - \beta_1^{t+1})\sum_{k=0}^{t} \beta_1^{t-k}(1 - \beta_1)\|g_k\|^2.$$

*Proof.* The first part of the lemma is the direct consequence of update rule of momentum $m_t$. For the second part we need to apply the Jensen's inequality as follows:

$$\left\| \sum_{k=0}^{t} \frac{\beta_1^{t-k}(1-\beta_1)}{1-\beta_1^{t+1}} g_k \right\|^2 \leq \sum_{k=0}^{t} \frac{\beta_1^{t-k}(1-\beta_1)}{1-\beta_1^{t+1}} \|g_k\|^2,$$

where we use the convexity of $\|\cdot\|^2$ and $\sum\limits_{k=0}^{t} \beta_1^{t-k}(1-\beta_1) = 1 - \beta_1^{t+1}$. Multiplying both sides by $(1-\beta_1^{t+1})^2$, we get the final result. $\qquad\square$

## C.2 NON-CONVEX CASE: METHODS WITH DELAY

**Lemma 15** (Descent lemma). *Let Assumption 2 hold on* $Q = \left\{ x \in \mathbb{R}^d \mid \exists y \in \mathbb{R}^d : f(y) \leq f_* + 2\Delta \text{ and } \|x - y\| \leq \frac{\sqrt{\Delta}}{20\sqrt{L}} \right\}$, *where* $f(x_0) - f_* = \Delta_0 \leq \Delta$. *Then, after* $T$ *iterations of* Clip-M-AdaGradD/Clip-AdamD *with* $b_0 \geq 2\gamma L/(1-\beta_1)^2 c_m^2$, *if* $x_t \in Q \ \forall t = \overline{0,T}$, *we have*

$$\sum_{t=0}^{T-1} \frac{\gamma C_t}{2} \|\nabla f(x_t)\|^2 \leq \Delta_0 - \Delta_T - \sum_{t=0}^{T-1} (\gamma C_t - 2A_t) \langle \nabla f(x_t), \theta_t^u \rangle$$

$$+ \sum_{t=0}^{T-1} \gamma C_t \|\theta_t^b\|^2 + \sum_{t=0}^{T-1} 2A_t \|\theta_t^u\|^2,$$

*where* $C_t = \sum\limits_{k=t}^{T-1} \frac{1-\beta_1}{b_k} \beta_1^{k-t}$, $A_t = \sum\limits_{k=t}^{T-1} \frac{L\gamma^2(1-\beta_1)}{c_m b_k b_0}(k-t+1)\beta_1^{k-t}$ *and* $c_m$ *is taken from Lemma 13.*

*Proof.* We start with the $L$-smoothness of $f$:

$$f(x_{t+1}) - f(x_t) \leq \langle \nabla f(x_t), x_{t+1} - x_t \rangle + \frac{L}{2} \|x_{t+1} - x_t\|^2$$

$$= -\frac{\gamma}{b_t} \langle \nabla f(x_t), m_t \rangle + \frac{L\gamma^2}{2b_t^2} \|m_t\|^2. \tag{34}$$

Using the update rule of Algorithm 3, we can obtain

$$-\langle \nabla f(x_t), m_t \rangle = -\beta_1 \langle \nabla f(x_t), m_{t-1} \rangle - (1-\beta_1) \langle \nabla f(x_t), g_t \rangle$$

$$= -\beta_1 \langle \nabla f(x_t) - \nabla f(x_{t-1}), m_{t-1} \rangle - \beta_1 \langle \nabla f(x_{t-1}), m_{t-1} \rangle$$

$$- (1-\beta_1) \langle \nabla f(x_t), g_t \rangle$$

$$\leq -\beta_1 \langle \nabla f(x_{t-1}), m_{t-1} \rangle + \beta_1 \|\nabla f(x_t) - \nabla f(x_{t-1})\| \|m_{t-1}\|$$

$$- (1-\beta_1) \langle \nabla f(x_t), g_t \rangle$$

$$\leq -\beta_1 \langle \nabla f(x_{t-1}), m_{t-1} \rangle + \beta_1 L \|x_t - x_{t-1}\| \|m_{t-1}\|$$

$$- (1-\beta_1) \langle \nabla f(x_t), g_t \rangle$$

$$= -\beta_1 \langle \nabla f(x_{t-1}), m_{t-1} \rangle + \frac{\gamma \beta_1 L}{b_{t-1}} \|m_{t-1}\|^2$$

$$- (1-\beta_1) \langle \nabla f(x_t), g_t \rangle,$$

where we use the Cauchy-Schwarz inequality and $L$-smoothness of $f$. Applying the same idea for the $t-1, t-2, \ldots, 0$ and noting that $m_{-1} = 0$, we get

$$-\langle \nabla f(x_t), m_t \rangle \leq -(1-\beta_1) \sum_{k=0}^{t} \beta_1^{t-k} \langle \nabla f(x_k), g_k \rangle + L\gamma \sum_{k=0}^{t-1} \frac{\beta_1^{t-k}}{b_k} \|m_k\|^2. \tag{35}$$

Therefore, substituting (35) into (34), we have

$$f(x_{t+1}) - f(x_t) \leq -\frac{(1-\beta_1)\gamma}{b_t} \sum_{k=0}^{t} \beta_1^{t-k} \langle \nabla f(x_k), g_k \rangle + \frac{L\gamma^2}{b_t} \sum_{k=0}^{t-1} \frac{\beta_1^{t-k}}{b_k} \|m_k\|^2 + \frac{L\gamma^2}{2b_t^2} \|m_t\|^2$$

$$\leq -\frac{(1-\beta_1)\gamma}{b_t} \sum_{k=0}^{t} \beta_1^{t-k} \langle \nabla f(x_k), g_k \rangle + \frac{L\gamma^2}{b_t} \sum_{k=0}^{t} \frac{\beta_1^{t-k}}{b_k} \|m_k\|^2.$$

Applying Lemma 14 with $1 - \beta_1^{k+1} \leq 1$, we can rewrite the inequality above as follows:

$$f(x_{t+1}) - f(x_t) \leq -\frac{(1-\beta_1)\gamma}{b_t} \sum_{k=0}^{t} \beta_1^{t-k} \langle \nabla f(x_k), g_k \rangle$$

$$+ \frac{L\gamma^2}{b_t} \sum_{k=0}^{t} \frac{\beta_1^{t-k}}{b_k} \sum_{j=0}^{k} \beta_1^{k-j}(1-\beta_1)\|g_j\|^2$$

$$= -\frac{(1-\beta_1)\gamma}{b_t} \sum_{k=0}^{t} \beta_1^{t-k} \langle \nabla f(x_k), g_k \rangle$$

$$+ \frac{L\gamma^2}{b_t} \sum_{j=0}^{t} \sum_{k=j}^{t} \frac{\beta_1^{t-k}}{b_k} \beta_1^{k-j}(1-\beta_1)\|g_j\|^2, \tag{36}$$

where we change the limits of summation. Now let us bound the second term. Applying Lemma 13, we obtain that $b_k \geq c_m b_0$ (the constant $c_m$ is taken from Lemma 13). Consequently,

$$\frac{L\gamma^2}{b_t} \sum_{j=0}^{t} \sum_{k=j}^{t} \frac{\beta_1^{t-k}}{b_k} \beta_1^{k-j}(1-\beta_1)\|g_j\|^2 \leq \frac{L\gamma^2(1-\beta_1)}{c_m b_t b_0} \sum_{j=0}^{t} \sum_{k=j}^{t} \beta_1^{t-k}\beta_1^{k-j}\|g_j\|^2$$

$$= \frac{L\gamma^2(1-\beta_1)}{c_m b_t b_0} \sum_{j=0}^{t} \beta_1^{t-j}(t-j+1)\|g_j\|^2. \tag{37}$$

Thus, substituting (37) into (36), we get

$$f(x_{t+1}) - f(x_t) \leq -\frac{(1-\beta_1)\gamma}{b_t} \sum_{k=0}^{t} \beta_1^{t-k} \langle \nabla f(x_k), g_k \rangle$$

$$+ \frac{L\gamma^2(1-\beta_1)}{c_m b_t b_0} \sum_{k=0}^{t} \beta_1^{t-k}(t-k+1)\|g_k\|^2.$$

After summing over $t = 0, \ldots T-1$,

$$f(x_T) - f(x_0) \leq -\sum_{t=0}^{T-1} \frac{(1-\beta_1)\gamma}{b_t} \sum_{k=0}^{t} \beta_1^{t-k} \langle \nabla f(x_k), g_k \rangle$$

$$+ \sum_{t=0}^{T-1} \frac{L\gamma^2(1-\beta_1)}{c_m b_t b_0} \sum_{k=0}^{t} \beta_1^{t-k}(t-k+1)\|g_k\|^2.$$

The main idea is to estimate the coefficients corresponding to $\langle \nabla f(x_r), g_r \rangle$ and $\|g_r\|^2$. These multiplicative factors can be estimated as

$$-\sum_{t=r}^{T-1} \frac{\gamma(1-\beta_1)}{b_t} \beta_1^{t-r} \tag{38}$$

for the scalar product and

$$\sum_{t=r}^{T-1} \frac{L\gamma^2(1-\beta_1)}{c_m b_t b_0}(t-r+1)\beta_1^{t-r} \tag{39}$$

for the squared norm, respectively. For (39) we can apply Lemma 13 in the following way:

$$\sum_{t=r}^{T-1} \frac{L\gamma^2(1-\beta_1)}{c_m b_t b_0}(t-r+1)\beta_1^{t-r} \leq \sum_{t=r}^{T-1} \frac{L\gamma^2(1-\beta_1)}{c_m^2 b_r b_0}(t-r+1)\beta_1^{t-r}$$

$$= \frac{L\gamma^2(1-\beta_1)}{c_m^2 b_r b_0} \sum_{t=r}^{T-1}(t-r+1)\beta_1^{t-r}.$$

Applying Lemma 1, and using that $\sum_{t=r}^{T-1} \beta_1^{t-r} \leq \frac{1}{1-\beta_1}$, we get

$$A_r = \sum_{t=r}^{T-1} \frac{L\gamma^2(1-\beta_1)}{c_m b_t b_0}(t-r+1)\beta_1^{t-r} \leq \frac{L\gamma^2}{c_m^2 b_k b_0 (1-\beta_1)} \tag{40}$$

for each $k = 0, \ldots, r$. Moreover, let us denote the factor corresponding to the scalar product (38) as $-\gamma C_r$. $C_r$ can be bounded as follows:

$$\frac{(1-\beta_1)}{b_r} \leq \sum_{t=r}^{T-1} \frac{(1-\beta_1)}{b_t}\beta_1^{t-r} \leq \sum_{t=r}^{T-1} \frac{(1-\beta_1)}{c_m b_0}\beta_1^{t-r} \leq \frac{1}{c_m b_0},$$

where we apply Lemma 13. Therefore, the descent lemma can be formulated as

$$f(x_T) - f(x_0) \leq -\sum_{t=0}^{T-1} \gamma C_t \langle \nabla f(x_t), g_t \rangle + \sum_{t=0}^{T-1} A_t \|g_t\|^2.$$

Substituting the analytical form of $g_t$, we have

$$\begin{aligned}
f(x_T) - f(x_0) &\leq -\sum_{t=0}^{T-1} \gamma C_t \langle \nabla f(x_t), g_t \rangle + \sum_{t=0}^{T-1} A_t \|g_t\|^2 \\
&= -\sum_{t=0}^{T-1} \gamma C_t \left( \langle \nabla f(x_t), \theta_t \rangle + \|\nabla f(x_t)\|^2 \right) \\
&\quad + \sum_{t=0}^{T-1} A_t \left( \|\theta_t\|^2 + 2\langle \nabla f(x_t), \theta_t \rangle + \|\nabla f(x_t)\|^2 \right) \\
&= -\sum_{t=0}^{T-1}(\gamma C_t - A_t)\|\nabla f(x_t)\|^2 - \sum_{t=0}^{T-1}(\gamma C_t - 2A_t)\langle \nabla f(x_t), \theta_t \rangle \\
&\quad + \sum_{t=0}^{T-1} A_t \|\theta_t\|^2.
\end{aligned}$$

Choosing $\gamma \leq \frac{(1-\beta_1)^2 c_m^2 b_0}{2L}$, we get that $\gamma C_t - 2A_t \geq 0$ since the boundary $C_t \geq \frac{1-\beta_1}{b_t}$ and (40) hold with $k = t$. Therefore, using that $\theta_t = \theta_t^u + \theta_t^b$, one can obtain

$$\begin{aligned}
f(x_T) - f(x_0) &\leq -\sum_{t=0}^{T-1}(\gamma C_t - A_t)\|\nabla f(x_t)\|^2 - \sum_{t=0}^{T-1}(\gamma C_t - 2A_t)\langle \nabla f(x_t), \theta_t \rangle \\
&\quad + \sum_{t=0}^{T-1} A_t \|\theta_t\|^2 \\
&\leq -\sum_{t=0}^{T-1}(\gamma C_t - A_t)\|\nabla f(x_t)\|^2 - \sum_{t=0}^{T-1}(\gamma C_t - 2A_t)\langle \nabla f(x_t), \theta_t^u \rangle \\
&\quad + \sum_{t=0}^{T-1} 2A_t \left( \|\theta_t^u\|^2 + \|\theta_t^b\|^2 \right) + \sum_{t=0}^{T-1} \left( \frac{\gamma C_t}{2} - A_t \right) \|\nabla f(x_t)\|^2 \\
&\quad + \sum_{t=0}^{T-1} \left( \frac{\gamma C_t}{2} - A_t \right) \|\theta_t^b\|^2 \\
&= -\sum_{t=0}^{T-1} \frac{\gamma C_t}{2}\|\nabla f(x_t)\|^2 - \sum_{t=0}^{T-1}(\gamma C_t - 2A_t)\langle \nabla f(x_t), \theta_t^u \rangle \\
&\quad + \sum_{t=0}^{T-1} 2A_t \|\theta_t^u\|^2 + \sum_{t=0}^{T-1} \left( \frac{\gamma C_t}{2} + A_t \right) \|\theta_t^b\|^2.
\end{aligned}$$

Using that $\frac{\gamma C_t}{2} \geq A_t$, and rearranging terms with $\Delta_t = f(x_t) - f_*$, we get the final result. $\qquad\square$

**Remark 1.** *It is important to note that $Q$ can be any non-empty subset of $\mathbb{R}^d$ as long as the iterates belong to it. In this sense, the form of $Q$ is not that important for the proof (a similar observation holds for Lemma 16 in the convex case). Nevertheless, $Q$ plays a key role in the next part of the proof.*

**Theorem 9.** *Let Assumptions 1 and 2 hold on $Q = \left\{ x \in \mathbb{R}^d \mid \exists y \in \mathbb{R}^d : f(y) \leq f_* + 2\Delta \text{ and } \|x - y\| \leq \frac{\sqrt{\Delta}}{20\sqrt{L}} \right\}$ with $f(x_0) - f_* = \Delta_0 \leq \Delta$. Then, after $K + 1$ iterations of* Clip-M-AdaGradD/Clip-AdamD *with*

$$
\gamma \leq \min \left\{ \frac{(1-\beta_1)^2 c_m^2 b_0 (K+1)^{\frac{1-\alpha}{3\alpha-2}}}{80 L \ln \frac{4(K+1)}{\delta}}, \frac{c_m \sqrt{1-\beta_1} 35^{\frac{1}{\alpha}} b_0 \sqrt{\Delta}}{432^{\frac{1}{\alpha}} \cdot 20 \sqrt{L} \sigma (K+1)^{\frac{\alpha}{3\alpha-2}} \ln^{\frac{\alpha-1}{\alpha}} \frac{4(K+1)}{\delta}}, \right.
$$

$$
\left. \frac{c_m (1-\beta_1)^{\frac{\alpha-1}{2\alpha-1}} b_0 \Delta^{\frac{\alpha}{2\alpha-1}}}{4^{\frac{\alpha+1}{2\alpha-1}} \cdot 20^{\frac{2\alpha-2}{2\alpha-1}} \sigma^{\frac{2\alpha}{2\alpha-1}} L^{\frac{\alpha-1}{2\alpha-1}} (K+1)^{\frac{\alpha}{3\alpha-2}} \ln^{\frac{2\alpha-2}{2\alpha-1}} \left( \frac{4(K+1)}{\delta} \right)} \right\}, \quad \eta = \frac{L\gamma^2(1-\beta_1)^2}{\Delta},
$$

$$
\tag{41}
$$

*and*

$$
\lambda = \frac{c_m \sqrt{1-\beta_1} b_0 \sqrt{\Delta} (K+1)^{\frac{1-\alpha}{3\alpha-2}}}{20 \sqrt{L} \gamma \ln \left( \frac{4(K+1)}{\delta} \right)}
\tag{42}
$$

*the bound*

$$
\sum_{k=0}^{K} \frac{\gamma C_k}{2} \|\nabla f(x_k)\|^2 \leq 2\Delta
$$

*holds with probability at least $1 - \delta$. In particular, when $\gamma$ equals the minimum from (41), the iterates produced by* Clip-M-AdaGradD/Clip-AdamD *satisfy*

$$
\frac{1}{K+1} \sum_{k=0}^{K} \|\nabla f(x_k)\|^2
$$

$$
= \mathcal{O} \left( \max \left\{ \frac{L\Delta \ln \frac{K+1}{\delta}}{(1-\beta_1)^3 (K+1)^{\frac{2\alpha-1}{3\alpha-2}}}, \frac{\sqrt{L\Delta} \sigma \ln^{\frac{\alpha-1}{\alpha}} \frac{K+1}{\delta}}{(1-\beta_1)^{\frac{3}{2}} (K+1)^{\frac{2\alpha-2}{3\alpha-2}}}, \frac{\sigma^{\frac{2\alpha}{2\alpha-1}} (L\Delta)^{\frac{\alpha-1}{2\alpha-1}} \ln^{\frac{2\alpha-2}{2\alpha-1}} \frac{K+1}{\delta}}{(1-\beta_1)^{\frac{3\alpha-2}{2\alpha-1}} (K+1)^{\frac{2\alpha-2}{3\alpha-2}}} \right\} \right)
$$

*with probability at least $1 - \delta$.*

*Proof.* Our proof is induction-based (similarly to the one for Clip-SGD by Sadiev et al. (2023)). We introduce probability event $E_k$ as follows: inequalities

$$
-\sum_{l=0}^{t-1} (\gamma C_l - 2A_l) \langle \nabla f(x_l), \theta_l^u \rangle + \sum_{l=0}^{t-1} \gamma C_l \|\theta_l^b\|^2 + \sum_{l=0}^{T-1} 2A_l \|\theta_l^u\|^2 \leq \Delta,
$$

$$
\Delta_t \leq 2\Delta
$$

hold simultaneously $\forall t = 0, 1, \ldots, k$. We want to show that $\mathbb{P}\{E_k\} \geq 1 - \frac{k\delta}{K+1}$ $\forall k = 0, 1, \ldots, K + 1$. The case when $k = 0$ is obvious. Now let us make an induction step: let the statement hold for some $k = T - 1 \leq K$: $\mathbb{P}\{E_{T-1}\} \geq 1 - \frac{(T-1)\delta}{K+1}$. It remains to prove that $\mathbb{P}\{E_T\} \geq 1 - \frac{T\delta}{K+1}$. The event $E_{T-1}$ implies that $x_t \in \{y \in \mathbb{R}^d : f(y) \leq f_* + 2\Delta\}$ $\forall t = 0, \ldots, T - 1$ and

$$
\|x_T - x_{T-1}\| = \frac{\gamma}{b_t} \|m_{T-1}\| \leq \frac{\gamma \lambda}{b_0} \leq \frac{c_m \sqrt{\Delta}}{20 \sqrt{L} \ln \frac{4(K+1)}{\delta}} \leq \frac{\sqrt{\Delta}}{20 \sqrt{L}}
$$

since $c_m \leq 1$. Hence, event $E_{T-1}$ implies $\{x_t\}_{t=0}^{T} \subseteq Q$ and we can apply Lemma 15:

$$
\sum_{l=0}^{t-1} \frac{\gamma C_l}{2} \|\nabla f(x_l)\|^2 \leq \Delta_0 - \Delta_t - \sum_{l=0}^{t-1} (\gamma C_l - 2A_l) \langle \nabla f(x_l), \theta_l^u \rangle + \sum_{l=0}^{t-1} \gamma C_l \|\theta_l^b\|^2
$$

$$
+ \sum_{l=0}^{t-1} 2A_l \|\theta_l^u\|^2
$$

$\forall t = 1, \ldots, T$ and $\forall t = 1, \ldots T - 1$ it implies that

$$\sum_{l=0}^{t-1} \frac{\gamma C_l}{2} \|\nabla f(x_l)\|^2 \leq \Delta_0 - \Delta_t - \sum_{l=0}^{t-1} (\gamma C_l - 2A_l) \langle \nabla f(x_l), \theta_l^u \rangle + \sum_{l=0}^{t-1} \gamma C_l \|\theta_l^b\|^2$$

$$+ \sum_{l=0}^{t-1} 2A_l \|\theta_l^u\|^2 \leq 2\Delta.$$

Taking into account that $\sum_{l=0}^{t-1} \frac{\gamma C_l}{2} \|\nabla f(x_l)\|^2 \geq 0$ for all $t$, we get that $E_{T-1}$ implies

$$\Delta_T \leq \Delta_0 - \sum_{t=0}^{T-1} (\gamma C_t - 2A_t) \langle \nabla f(x_t), \theta_t^u \rangle + \sum_{t=0}^{T-1} \gamma C_t \|\theta_t^b\|^2 + \sum_{t=0}^{T-1} 2A_t \|\theta_t^u\|^2$$

$$= \Delta_0 - \sum_{t=0}^{T-1} (\gamma C_t - 2A_t) \langle \nabla f(x_t), \theta_t^u \rangle + \sum_{t=0}^{T-1} \gamma C_t \|\theta_t^b\|^2$$

$$+ \sum_{t=0}^{T-1} 2A_t \left( \|\theta_t^u\|^2 - \mathbb{E}_{\xi_t} \|\theta_t^u\|^2 \right) + \sum_{t=0}^{T-1} 2A_t \mathbb{E}_{\xi_t} \|\theta_t^u\|^2.$$

Next, for vectors

$$\eta_t = \begin{cases} \nabla f(x_t), & \|\nabla f(x_t)\| \leq 2\sqrt{L\Delta} \\ 0, & \text{otherwise} \end{cases}$$

for all $t = 0, 1, \ldots, T - 1$, we have that that with probability 1

$$\|\eta_t\| \leq 2\sqrt{L\Delta}. \tag{43}$$

What is more, for all $t = 0, \ldots T - 1$ $E_{T-1}$ implies

$$\|\nabla f(x_t)\| \leq \sqrt{2L\Delta_t} \leq 2\sqrt{L\Delta} \overset{(42)}{\leq} \frac{\lambda}{2}$$

Thus, $E_{T-1}$ implies $\eta_t = \nabla f(x_t)$ for $t = 0, 1, \ldots, T - 1$ and

$$\Delta_T \leq \Delta_0 \underbrace{- \sum_{t=0}^{T-1} (\gamma C_t - 2A_t) \langle \eta_t, \theta_t^u \rangle}_{①} + \underbrace{\sum_{t=0}^{T-1} \gamma C_t \|\theta_t^b\|^2}_{②}$$

$$+ \underbrace{\sum_{t=0}^{T-1} 2A_t \left( \|\theta_t^u\|^2 - \mathbb{E}_{\xi_t} \|\theta_t^u\|^2 \right)}_{③} + \underbrace{\sum_{t=0}^{T-1} 2A_t \mathbb{E}_{\xi_t} \|\theta_t^u\|^2}_{④}. \tag{44}$$

It remains to bound each term in (44) separately with high probability. Before we move on, we also note that event $E_{T-1}$ implies $\|\nabla f(x_t)\| \leq \frac{\lambda}{2}$. Therefore, one can apply Lemma 3 and get

$$\|\theta_t^u\| \leq 2\lambda, \tag{45}$$

$$\|\theta_t^b\| \leq \frac{2^\alpha \sigma^\alpha}{\lambda^{\alpha-1}}, \tag{46}$$

$$\mathbb{E}_{\xi_t} \|\theta_t^u\|^2 \leq 18\lambda^{2-\alpha} \sigma^\alpha. \tag{47}$$

**Bound for ①.** The definition of $\theta_t^u$ implies

$$\mathbb{E}_{\xi_t} \left[ - (\gamma C_t - 2A_t) \langle \eta_t, \theta_t^u \rangle \right] = 0.$$

What is more, since $C_t \leq \frac{1}{c_m b_0}$, we get

$$|(\gamma C_t - 2A_t) \langle \eta_t, \theta_t^u \rangle| \leq \gamma C_t \|\eta_t\| \|\theta_t^u\| \overset{(43),(45)}{\leq} \frac{4\gamma \lambda \sqrt{L\Delta}}{c_m b_0} \leq \frac{\Delta}{5 \ln \left( \frac{4(K+1)}{\delta} \right)} = c.$$

Let us define $\sigma_t^2 = \mathbb{E}_{\xi_t}\left[(\gamma C_t - 2A_t)^2 \langle \eta_t, \theta_t^u \rangle^2\right]$. Hence,

$$\sigma_t^2 \overset{(43)}{\leq} (\gamma C_t - 2A_t)^2 \cdot 4L\Delta \mathbb{E}_{\xi_t}\|\theta_t^u\|^2 \leq \frac{4\gamma^2 L\Delta}{c_m^2 b_0^2} \mathbb{E}_{\xi_t}\|\theta_t^u\|^2. \tag{48}$$

Therefore, we can apply Bernstein's inequality (Lemma 4) with $G = \frac{7\Delta^2}{480 \ln \frac{4(K+1)}{\delta}}$:

$$\mathbb{P}\left\{\left|-\sum_{t=0}^{T-1}(\gamma C_t - 2A_t)\langle \nabla f(x_t), \theta_t^u \rangle\right| > \frac{\Delta}{4} \text{ and } \sum_{t=0}^{T-1}\sigma_t^2 \leq G\right\} \leq 2\exp\left(-\frac{\Delta^2}{16\left(2G + \frac{\Delta c}{6}\right)}\right)$$

$$= \frac{\delta}{2(K+1)}.$$

Thus, we get

$$\mathbb{P}\left\{\text{either } \left|-\sum_{t=0}^{T-1}(\gamma C_t - 2A_t)\langle \nabla f(x_t), \theta_t^u \rangle\right| \leq \frac{\Delta}{4} \text{ or } \sum_{t=0}^{T-1}\sigma_t^2 > G\right\} \geq 1 - \frac{\delta}{2(K+1)}.$$

Moreover, event $E_{T-1}$ implies

$$\sum_{t=0}^{T-1}\sigma_t^2 \overset{(47)}{\leq} \frac{72\gamma^2\lambda^{2-\alpha}\sigma^\alpha L\Delta T}{c_m^2 b_0^2} \overset{(42)}{=} \frac{72c_m^{2-\alpha}(1-\beta_1)^{1-\frac{\alpha}{2}}\gamma^\alpha b_0^{2-\alpha}\sqrt{\Delta}^{2-\alpha}(K+1)^{\frac{\alpha^2-3\alpha+2}{3\alpha-2}}\sigma^\alpha L\Delta T}{c_m^2 20^{2-\alpha}\sqrt{L}^{2-\alpha}b_0^2 \ln^{2-\alpha}\frac{4(K+1)}{\delta}}$$

$$\overset{(41)}{\leq} \frac{7\Delta^2}{480 \ln \frac{4(K+1)}{\delta}}.$$

**Bound for ②.** For the second term, we get that $E_{T-1}$ implies

$$\sum_{t=0}^{T-1}\gamma C_t \|\theta_t^b\|^2 \leq \sum_{t=0}^{T-1}\frac{\gamma}{c_m b_0}\|\theta_t^b\|^2 \overset{(46)}{\leq} \frac{4^\alpha \sigma^{2\alpha}\gamma T}{c_m \lambda^{2\alpha-2}b_0}$$

$$\overset{(42)}{\leq} \frac{4^\alpha \sigma^{2\alpha}\gamma(K+1)}{c_m b_0} \cdot \frac{20^{2\alpha-2}L^{\alpha-1}\gamma^{2\alpha-2}(K+1)^{\frac{(\alpha-1)(2\alpha-2)}{3\alpha-2}}\ln^{2\alpha-2}\left(\frac{4(K+1)}{\delta}\right)}{c_m^{2\alpha-2}(1-\beta_1)^{\alpha-1}b_0^{2\alpha-2}\Delta^{\alpha-1}}$$

$$= \frac{4^\alpha \cdot 20^{2\alpha-2}\sigma^{2\alpha}L^{\alpha-1}(K+1)^{\frac{\alpha(2\alpha-1)}{3\alpha-2}}\ln^{2\alpha-2}\left(\frac{4(K+1)}{\delta}\right)}{c_m^{2\alpha-1}(1-\beta_1)^{\alpha-1}b_0^{2\alpha-1}\Delta^{\alpha-1}} \cdot \gamma^{2\alpha-1}$$

$$\overset{(41)}{\leq} \frac{\Delta}{4},$$

where in the last step, we apply the third condition on $\gamma$ from (41).

**Bound for ③.** Similarly to ①, we have unbiased and bounded terms in the sum:

$$\mathbb{E}_{\xi_t}\left[2A_t\left(\|\theta_t^u\|^2 - \mathbb{E}_{\xi_t}\|\theta_t^u\|^2\right)\right] = 0$$

and, since (40) from Lemma 15 hold with $k = 0$,

$$\left|2A_t\left(\|\theta_t^u\|^2 - \mathbb{E}_{\xi_t}\|\theta_t^u\|^2\right)\right| \overset{(45)}{\leq} \frac{16L\lambda^2\gamma^2}{c_m^2 b_0^2(1-\beta_1)} \leq \frac{\Delta}{25\ln\frac{4(K+1)}{\delta}} \leq \frac{15\Delta}{47\ln\frac{4(K+1)}{\delta}} = c. \tag{49}$$

Next, we define $\hat{\sigma}_t^2 = \mathbb{E}_{\xi_t}\left[4A_t^2\left(\|\theta_t^u\|^2 - \mathbb{E}_{\xi_t}\|\theta_t^u\|^2\right)^2\right]$. For the introduced quantities, we have

$$\hat{\sigma}_t^2 \overset{(49)}{\leq} c\mathbb{E}_{\xi_t}\left[2A_t\left|\left(\|\theta_t^u\|^2 - \mathbb{E}_{\xi_t}\|\theta_t^u\|^2\right)\right|\right] \leq \frac{4L\gamma^2 c}{c_m^2 b_0^2(1-\beta_1)}\mathbb{E}_{\xi_t}\|\theta_t^u\|^2. \tag{50}$$

Therefore, we can apply Bernstein's inequality (Lemma 4) with $G = \frac{7\Delta^2}{1504\ln\frac{4(K+1)}{\delta}}$:

$$\mathbb{P}\left\{\left|\sum_{t=0}^{T-1}2A_t\left(\|\theta_t^u\|^2 - \mathbb{E}_{\xi_t}\|\theta_t^u\|^2\right)\right| > \frac{\Delta}{4} \text{ and } \sum_{t=0}^{T-1}\hat{\sigma}_t^2 \leq G\right\} \leq 2\exp\left(-\frac{\Delta^2}{16\left(2G + \frac{\Delta c}{6}\right)}\right)$$

$$= \frac{\delta}{2(K+1)}.$$

Thus, we get

$$\mathbb{P}\left\{\text{either } \left|\sum_{t=0}^{T-1} 2A_t\left(\|\theta_t^u\|^2 - \mathbb{E}_{\xi_t}\|\theta_t^u\|^2\right)\right| \le \frac{\Delta}{4} \text{ or } \sum_{t=0}^{T-1}\hat{\sigma}_t^2 > G\right\} \ge 1 - \frac{\delta}{2(K+1)}.$$

Moreover, event $E_{T-1}$ implies

$$\sum_{t=0}^{T-1}\hat{\sigma}_t^2 \overset{(50),(45)}{\le} \frac{72L\gamma^2 c\lambda^{2-\alpha}\sigma^\alpha}{c_m^2 b_0^2(1-\beta_1)} \overset{(42)}{\le} \frac{72c\gamma^\alpha b_0^{2-\alpha}\sqrt{\Delta}^{2-\alpha}(K+1)^{\frac{\alpha^2-3\alpha+2}{3\alpha-2}}\sigma^\alpha LT}{20^{2-\alpha}c_m^\alpha(1-\beta_1)^{\frac{\alpha}{2}}\sqrt{L}^{2-\alpha}b_0^2 \ln^{2-\alpha}\frac{4(K+1)}{\delta}}$$

$$\overset{(41)}{\le} \frac{7\Delta c}{480} \le \frac{7\Delta^2}{1504\ln\frac{4(K+1)}{\delta}}.$$

**Bound for ④.** For the last term, we have that $E_{T-1}$ implies

$$\sum_{t=0}^{T-1} 2A_t\mathbb{E}_{\xi_t}\|\theta_t^u\|^2 \le \sum_{t=0}^{T-1}\frac{2L\gamma^2}{c_m^2 b_0^2(1-\beta_1)}\mathbb{E}_{\xi_t}\|\theta_t^u\|^2$$

$$\overset{(45)}{\le} \frac{36L\gamma^2\lambda^{2-\alpha}\sigma^\alpha T}{c_m^2 b_0^2(1-\beta_1)} \overset{(42)}{\le} \frac{36\gamma^\alpha b_0^{2-\alpha}\sqrt{\Delta}^{2-\alpha}(K+1)^{\frac{\alpha^2-3\alpha+2}{3\alpha-2}}\sigma^\alpha LT}{20^{2-\alpha}(1-\beta_1)^{\frac{\alpha}{2}}\sqrt{L}^{2-\alpha}b_0^2 \ln^{2-\alpha}\frac{4(K+1)}{\delta}}$$

$$\overset{(41)}{\le} \frac{7\Delta}{960\ln\frac{4(K+1)}{\delta}} \le \frac{\Delta}{4}.$$

Thus, taking into account the bounds above, the probability event $E_{T-1}\cap E_1\cap E_2$ implies that

$$\Delta_T \le \Delta + 4\frac{\Delta}{4} = 2\Delta,$$

where

$$E_1 = \left\{\text{either } \left|-\sum_{t=0}^{T-1}\left(\frac{\gamma}{b_t} - \frac{L\gamma^2}{b_t^2}\right)\langle\nabla f(x_t),\theta_t^u\rangle\right| \le \frac{\Delta}{4} \text{ or } \sum_{t=0}^{T-1}\sigma_t^2 > \frac{7\Delta^2}{480\ln\frac{4(K+1)}{\delta}}\right\},$$

$$E_2 = \left\{\text{either } \left|\sum_{t=0}^{T-1}\frac{L\gamma^2}{b_t^2}\left(\|\theta_t^u\|^2 - \mathbb{E}_{\xi_t}\|\theta_t^u\|^2\right)\right| \le \frac{\Delta}{4} \text{ or } \sum_{t=0}^{T-1}\hat{\sigma}_t^2 > \frac{7\Delta^2}{1504\ln\frac{4(K+1)}{\delta}}\right\}.$$

Therefore,

$$\mathbb{P}\{E_T\} \ge \mathbb{P}\{E_{T-1}\cap E_1\cap E_2\} = 1 - \mathbb{P}\{\overline{E}_{T-1}\cup\overline{E}_1\cup\overline{E}_2\}$$

$$\ge 1 - \mathbb{P}\{\overline{E}_{T-1}\} - \mathbb{P}\{\overline{E}_1\} - \mathbb{P}\{\overline{E}_2\} \ge 1 - \frac{T\delta}{K+1}.$$

Hence, for all $k = 0,\dots,K+1$ we get $\mathbb{P}(E_k) \ge 1 - \frac{k\delta}{K+1}$. As revision result, event $E_{K+1}$ implies that

$$\sum_{k=0}^{K}\frac{\gamma C_k}{2}\|\nabla f(x_k)\|^2 \le 2\Delta \tag{51}$$

holds with probability at least $1-\delta$.

Therefore, we get that with probability at least $1-\delta$

$$\sum_{k=0}^{K}\|\nabla f(x_k)\|^2 \le \frac{4\Delta}{\gamma}\max_{k\in[0,K]}\frac{1}{C_k}.$$

and, since $C_k \ge \frac{1-\beta_1}{b_k}$, we obtain

$$\sum_{k=0}^{K}\|\nabla f(x_k)\|^2 \le \frac{4\Delta}{\gamma(1-\beta_1)}\max_{k\in[0,K]}b_k. \tag{52}$$

Moreover,

$$b_k^2 \le b_0^2 + \eta \sum_{k=0}^{K} \left( 3\|\nabla f(x_k)\|^2 + 3\|\theta_k^u\|^2 + 3\|\theta_k^b\|^2 \right) \tag{53}$$

for the Clip-AdaGradD of $b_k$ and

$$b_k^2 \le b_0^2 + \frac{\eta}{K+1} \sum_{k=0}^{K} \left( 3\|\nabla f(x_k)\|^2 + 3\|\theta_k^u\|^2 + 3\|\theta_k^b\|^2 \right) \tag{54}$$

for the Clip-AdamD, respectively. Next, we use that the event $E_{K+1}$ implies

$$\sum_{k=0}^{K} \frac{\gamma}{c_m b_0} \|\theta_k^b\|^2 \le \frac{\Delta}{4};$$

$$\sum_{k=0}^{K} \frac{2L\gamma^2}{c_m^2 b_0^2 (1-\beta_1)} \|\theta_k^u\|^2 \le \frac{\Delta}{2}$$

because we could substitute bounds on $C_t$ and $A_t$ directly in Lemma 15 and all steps in ②, ③ and ④ will be the same. Therefore, with applying Lemma 13, next bounds

$$\sum_{k=0}^{K} \|\nabla f(x_k)\|^2 \le \frac{4\Delta}{\gamma(1-\beta_1)} \sqrt{b_0^2 + 3\eta \sum_{k=0}^{K} \|\nabla f(x_k)\|^2 + \frac{3\eta b_0 \Delta}{4\gamma} + \frac{3\eta b_0^2 (1-\beta_1)\Delta}{4L\gamma^2}};$$

$$\sum_{k=0}^{K} \|\nabla f(x_k)\|^2 \le \frac{4\Delta}{\gamma(1-\beta_1)} \sqrt{b_0^2 + \frac{3\eta}{K+1} \sum_{k=0}^{K} \|\nabla f(x_k)\|^2 + \frac{3\eta b_0 \Delta}{8\gamma(K+1)} + \frac{3\eta b_0^2 (1-\beta_1)\Delta}{16L\gamma^2(K+1)}}$$

hold with probability at least $1 - \delta$, where we substitute different $c_m$ from Lemma 13 and (53), (54) for Clip-M-AdaGradD and Clip-AdamD, respectively. Next, solving quadratic inequalities above with respect to $\sum_{k=0}^{K} \|\nabla f(x_k)\|^2$, we obtain

$$\sum_{k=0}^{K} \|\nabla f(x_k)\|^2 \le \frac{\frac{48\eta\Delta^2}{\gamma^2(1-\beta_1)^2} + \sqrt{\frac{9 \cdot 4^4 \eta^2 \Delta^4}{\gamma^4(1-\beta_1)^4} + \frac{16\Delta^2}{\gamma^2(1-\beta_1)^2} \left( \frac{3\eta b_0 \Delta}{4\gamma} + \frac{3\eta b_0^2(1-\beta_1)\Delta}{4L\gamma^2} + b_0^2 \right)}}{2}$$

$$= \frac{24\eta\Delta^2}{\gamma^2(1-\beta_1)^2}$$

$$+ \sqrt{\frac{576\eta^2\Delta^4}{\gamma^4(1-\beta_1)^4} + \left( \frac{3\eta b_0 \Delta^3}{\gamma^3(1-\beta_1)^2} + \frac{3\eta b_0^2 \Delta^3}{L\gamma^4(1-\beta_1)} + \frac{4b_0^2\Delta^2}{\gamma^2(1-\beta_1)^2} \right)}$$

$$= \frac{\Delta}{\gamma^2} \left( \frac{24\eta\Delta}{(1-\beta_1)^2} + \sqrt{\frac{576\eta^2\Delta^2}{(1-\beta_1)^4} + \left( \frac{3\eta b_0 \gamma\Delta}{(1-\beta_1)^2} + \frac{3\eta b_0^2 \Delta}{L(1-\beta_1)} + \frac{4b_0^2\gamma^2}{(1-\beta_1)^2} \right)} \right)$$

for Clip-M-AdaGradD and

$$\sum_{k=0}^{K} \|\nabla f(x_k)\|^2 \leq \frac{24\eta\Delta^2}{\gamma^2(1-\beta_1)^2(K+1)}$$

$$+ \sqrt{\frac{9 \cdot 4^3 \eta^2 \Delta^4}{\gamma^4(1-\beta_1)^4(K+1)^2} + \frac{4\Delta^2}{\gamma^2(1-\beta_1)^2}\left(\frac{3\eta b_0 \Delta}{8\gamma(K+1)} + \frac{3\eta b_0^2(1-\beta_1)\Delta}{16L\gamma^2(K+1)} + b_0^2\right)}$$

$$= \frac{24\eta\Delta^2}{\gamma^2(1-\beta_1)^2(K+1)}$$

$$+ \sqrt{\frac{576\eta^2\Delta^4}{\gamma^4(1-\beta_1)^4(K+1)^2} + \left(\frac{3\eta b_0 \Delta^3}{2\gamma^3(1-\beta_1)^2(K+1)} + \frac{3\eta b_0^2 \Delta^3}{4L\gamma^4(1-\beta_1)(K+1)} + \frac{4b_0^2\Delta^2}{\gamma^2(1-\beta_1)^2}\right)}$$

$$= \frac{\Delta}{\gamma^2}\left(\frac{24\eta\Delta}{(1-\beta_1)^2(K+1)}\right.$$

$$\left. + \sqrt{\frac{576\eta^2\Delta^2}{(1-\beta_1)^4(K+1)^2} + \left(\frac{3\eta b_0 \gamma \Delta}{2(1-\beta_1)^2(K+1)} + \frac{3\eta b_0^2 \Delta}{4L(1-\beta_1)(K+1)} + \frac{4b_0^2\gamma^2}{(1-\beta_1)^2}\right)}\right)$$

for the Clip-AdamD. Substituting $\eta = \frac{L\gamma^2(1-\beta_1)^2}{\Delta}$ and applying $\sqrt{a^2 + b^2 + c^2 + d^2} \leq a+b+c+d$ for non-negative numbers, one can obtain the bound for Clip-M-AdaGradD:

$$\frac{1}{K+1}\sum_{k=0}^{K}\|\nabla f(x_k)\|^2 \leq \frac{\Delta}{(K+1)\gamma^2}\left(48L\gamma^2 + \sqrt{3L\gamma^3 b_0} + \sqrt{3\gamma^2 b_0^2(1-\beta_1)} + \frac{2\gamma b_0}{1-\beta_1}\right)$$

$$\leq \frac{\Delta}{(K+1)\gamma^2}\left(49L\gamma^2 + 3\sqrt{\gamma^2 b_0^2(1-\beta_1)} + \frac{2\gamma b_0}{1-\beta_1}\right)$$

$$\leq \frac{\Delta}{(K+1)\gamma^2}\left(49L\gamma^2 + 3\gamma b_0 + \frac{2\gamma b_0}{1-\beta_1}\right)$$

$$\leq \frac{2\Delta}{(K+1)\gamma^2}\max\left\{49L\gamma^2, \frac{5\gamma b_0}{1-\beta_1}\right\}$$

$$= \max\left\{\frac{98L\Delta}{K+1}, \frac{10\Delta b_0}{\gamma(K+1)(1-\beta_1)}\right\} \tag{55}$$

and for Clip-AdamD:

$$\frac{1}{K+1}\sum_{k=0}^{K}\|\nabla f(x_k)\|^2 \leq \frac{\Delta}{(K+1)\gamma^2}\left(\frac{48L\gamma^2}{K+1} + \sqrt{\frac{3L\gamma^3 b_0}{2(K+1)}} + \sqrt{\frac{3\gamma^2 b_0^2(1-\beta_1)}{4(K+1)}} + \frac{2\gamma b_0}{1-\beta_1}\right)$$

$$\leq \frac{\Delta}{(K+1)\gamma^2}\left(\frac{48L\gamma^2}{K+1} + 2\sqrt{\frac{L\gamma^3 b_0}{(K+1)}} + \gamma b_0 + \frac{2\gamma b_0}{1-\beta_1}\right)$$

$$\leq \frac{\Delta}{(K+1)\gamma^2}\left(\frac{49L\gamma^2}{K+1} + \frac{4\gamma b_0}{1-\beta_1}\right)$$

$$\leq \frac{2\Delta}{(K+1)\gamma^2}\max\left\{\frac{49L\gamma^2}{K+1}, \frac{4\gamma b_0}{1-\beta_1}\right\}$$

$$= \max\left\{\frac{98L\Delta}{(K+1)^2}, \frac{8\Delta b_0}{\gamma(K+1)(1-\beta_1)}\right\}, \tag{56}$$

where we use that $2\sqrt{ab} \leq a + b$. Consequently, after substitution of (41) into (55), (56), we get final bounds for Clip-M-AdaGradD/Clip-AdamD:

$$\frac{1}{K+1}\sum_{k=0}^{K}\|\nabla f(x_k)\|^2$$

$$= \mathcal{O}\left(\max\left\{\frac{L\Delta\ln\frac{K+1}{\delta}}{(1-\beta_1)^3(K+1)^{\frac{2\alpha-1}{3\alpha-2}}}, \frac{\sqrt{L\Delta}\sigma\ln^{\frac{\alpha-1}{\alpha}}\frac{K+1}{\delta}}{(1-\beta_1)^{\frac{3}{2}}(K+1)^{\frac{2\alpha-2}{3\alpha-2}}}, \frac{\sigma^{\frac{2\alpha}{2\alpha-1}}(L\Delta)^{\frac{\alpha-1}{2\alpha-1}}\ln^{\frac{2\alpha-2}{2\alpha-1}}\frac{K+1}{\delta}}{(1-\beta_1)^{\frac{2\alpha-2}{2\alpha-1}}(K+1)^{\frac{2\alpha-2}{3\alpha-2}}}\right\}\right)$$

holds with probability at least $1 - \delta$. □

### C.3 CONVEX CASE: METHODS WITH DELAY

**Lemma 16** (Descent lemma). *Let Assumptions 2 and 3 hold on $Q = B_{2R}(x^*)$, where $\|x_0 - x^*\| \leq R$. Assume that $x_t \in Q \ \forall t = \overline{0, T}$. Then, after $T$ iterations of* Clip-M-AdaGradD/Clip-AdamD *with $b_0 \geq \frac{8\gamma L}{(1-\beta_1)^2 c_m^2}$, we have*

$$\sum_{t=0}^{T-1}\gamma C_t\left(f(x_t) - f_*\right) \leq R_0^2 - R_t^2 - \sum_{t=0}^{T-1}2\gamma C_t\left\langle x_t - x^*, \theta_t\right\rangle + \sum_{t=0}^{T-1}2A_t\|\theta_t\|^2,$$

*where $C_t = \sum\limits_{i=t}^{T-1}\frac{1-\beta_1}{b_i}\beta_1^{i-t}$ and $A_t = \sum\limits_{i=t}^{T-1}\frac{2\gamma^2(1-\beta_1)}{c_m b_i b_0}\beta_1^{i-t}(i-t+1)$.*

*Proof.* According to the update rule of Algorithm 3, we have

$$\|x_{t+1} - x^*\|^2 = \|x_t - x^*\|^2 - \frac{2\gamma}{b_t}\left\langle x_t - x^*, m_t\right\rangle + \frac{\gamma^2}{b_t^2}\|m_t\|^2.$$

To bound the scalar product, we substitute the update rule for $m_t$:

$$-\left\langle x_t - x^*, m_t\right\rangle = -\beta_1\left\langle x_t - x^*, m_{t-1}\right\rangle - (1-\beta_1)\left\langle x_t - x^*, g_t\right\rangle$$
$$= -\beta_1\left\langle x_t - x_{t-1}, m_{t-1}\right\rangle - \beta_1\left\langle x_{t-1} - x^*, m_{t-1}\right\rangle$$
$$- (1-\beta_1)\left\langle x_t - x^*, g_t\right\rangle$$
$$\leq -\beta_1\left\langle x_{t-1} - x^*, m_{t-1}\right\rangle - (1-\beta_1)\left\langle x_t - x^*, g_t\right\rangle$$
$$+ \beta_1\|x_t - x_{t-1}\|\|m_{t-1}\|$$
$$= -\beta_1\left\langle x_{t-1} - x^*, m_{t-1}\right\rangle - (1-\beta_1)\left\langle x_t - x^*, g_t\right\rangle$$
$$+ \frac{\gamma\beta_1}{b_{t-1}}\|m_{t-1}\|^2.$$

Applying the same idea for $t-1, t-2, \ldots, 0$ and using that $m_{-1} = 0$, one can obtain

$$-\left\langle x_t - x^*, m_t\right\rangle \leq -\sum_{k=0}^{t}(1-\beta_1)\beta_1^{t-k}\left\langle x_k - x^*, g_k\right\rangle + \sum_{k=0}^{t-1}\frac{\gamma\beta_1^{t-k}}{b_k}\|m_k\|^2.$$

Therefore, we get

$$\|x_{t+1} - x^*\|^2 \leq \|x_t - x^*\|^2 - \frac{2\gamma}{b_t}\sum_{k=0}^{t}(1-\beta_1)\beta_1^{t-k}\left\langle x_k - x^*, g_k\right\rangle + \frac{2\gamma^2}{b_t}\sum_{k=0}^{t}\frac{\beta_1^{t-k}}{b_k}\|m_k\|^2.$$

Substituting the bound for $\|m_k\|^2$ from Lemma 14 with $1 - \beta_1^{k+1} \leq 1$, we have

$$\|x_{t+1} - x^*\|^2 \leq \|x_t - x^*\|^2 - \frac{2\gamma}{b_t}\sum_{k=0}^{t}(1-\beta_1)\beta_1^{t-k}\left\langle x_k - x^*, g_k\right\rangle$$

$$+ \frac{2\gamma^2}{b_t}\sum_{k=0}^{t}\frac{\beta_1^{t-k}}{b_k}\sum_{j=0}^{k}\beta_1^{k-j}(1-\beta_1)\|g_j\|^2$$

$$= \|x_t - x^*\|^2 - \frac{2\gamma}{b_t}\sum_{k=0}^{t}(1-\beta_1)\beta_1^{t-k}\left\langle x_k - x^*, g_k\right\rangle$$

$$+ \frac{2\gamma^2}{b_t}\sum_{k=0}^{t}\sum_{j=0}^{k}\frac{\beta_1^{t-j}}{b_k}(1-\beta_1)\|g_j\|^2.$$

Applying the same technique as in Lemma 15 (see (37)), one can obtain

$$\|x_{t+1} - x^*\|^2 \leq \|x_t - x^*\|^2 - \frac{2\gamma(1-\beta_1)}{b_t} \sum_{k=0}^{t} \beta_1^{t-k} \langle x_k - x^*, g_k \rangle$$

$$+ \frac{2\gamma^2(1-\beta_1)}{c_m b_t b_0} \sum_{j=0}^{t} \beta_1^{t-j}(t-j+1)\|g_j\|^2.$$

After summing over $t$:

$$\|x_T - x^*\|^2 \leq \|x_0 - x^*\|^2 - \sum_{t=0}^{T-1} \frac{2\gamma(1-\beta_1)}{b_t} \sum_{k=0}^{t} \beta_1^{t-k} \langle x_k - x^*, g_k \rangle$$

$$+ \sum_{t=0}^{T-1} \frac{2\gamma^2(1-\beta_1)}{c_m b_t b_0} \sum_{j=0}^{t} \beta_1^{t-j}(t-j+1)\|g_j\|^2. \tag{57}$$

Therefore, multiplicative factors for $\langle x_r - x^*, g_r \rangle$ and $\|g_r\|^2$ are equal to

$$-\sum_{t=r}^{T-1} \frac{2\gamma(1-\beta_1)}{b_t} \beta_1^{t-r} \qquad \text{and} \qquad \sum_{t=r}^{T-1} \frac{2\gamma^2(1-\beta_1)}{c_m b_t b_0} \beta_1^{t-r}(t-r+1),$$

respectively. Let us denote them as $-2\gamma C_r$ and $A_r$. Using the same idea as in Lemma 15, we get

$$\frac{(1-\beta_1)}{b_r} \leq C_r \leq \frac{1}{c_m b_p}$$

and

$$A_r \leq \frac{2\gamma^2}{c_m^2 b_p b_0 (1-\beta_1)}$$

for all $p = 0, \ldots r$ because of Lemma 13. Rewriting (57) in terms of $C_r$, $A_r$,

$$\|x_T - x^*\|^2 \leq \|x_0 - x^*\|^2 - \sum_{t=0}^{T-1} 2\gamma C_t \langle x_t - x^*, g_t \rangle + \sum_{t=0}^{T-1} A_t \|g_t\|^2.$$

Consequently,

$$\|x_T - x^*\|^2 - \|x_0 - x^*\|^2 \leq -\sum_{t=0}^{T-1} 2\gamma C_t \langle x_t - x^*, g_t \rangle + \sum_{t=0}^{T-1} A_t \|g_t\|^2$$

$$= -\sum_{t=0}^{T-1} 2\gamma C_t \langle x_t - x^*, \nabla f(x_t) + \theta_t \rangle + \sum_{t=0}^{T-1} A_t \|\nabla f(x_t) + \theta_t\|^2$$

$$\leq -\sum_{t=0}^{T-1} 2\gamma C_t \langle x_t - x^*, \nabla f(x_t) \rangle - \sum_{t=0}^{T-1} 2\gamma C_t \langle x_t - x^*, \theta_t \rangle$$

$$+ \sum_{t=0}^{T-1} 2A_t \|\nabla f(x_t)\|^2 + \sum_{t=0}^{T-1} 2A_t \|\theta_t\|^2.$$

Using Assumptions 2 and 3, one can obtain

$$\sum_{t=0}^{T-1} (2\gamma C_t - 4LA_t)(f(x_t) - f_*) \leq \sum_{t=0}^{T-1} \left( 2\gamma C_t \langle x_t - x^*, \nabla f(x_t) \rangle - 2A_t \|f(x_t)\|^2 \right)$$

$$\leq \|x_0 - x^*\|^2 - \|x_T - x^*\|^2 - \sum_{t=0}^{T-1} 2\gamma C_t \langle x_t - x^*, \theta_t \rangle$$

$$+ \sum_{t=0}^{T-1} 2A_t \|\theta_t\|^2.$$

If we choose $\gamma \leq \frac{(1-\beta_1)^2 c_m^2 b_0}{8L}$, then $2\gamma C_t - 4LA_t \geq \gamma C_t$ because of lower bound on $C_t$ and upper bound for $A_t$. This finishes the proof. $\qquad\square$

**Theorem 10.** *Let Assumptions 1, 2, and 3 hold on $Q = B_{2R}(x^*)$ with $\|x_0 - x^*\| \leq R$, Then, after $K + 1$ iterations of* Clip-M-AdaGradD/Clip-AdamD *with*

$$
\gamma \leq \min \left\{ \frac{(1 - \beta_1)^2 c_m^2 b_0}{160 L \ln \left( \frac{4(K+1)}{\delta} \right)}, \frac{\sqrt{1 - \beta_1} c_m R b_0}{40 \cdot 9^{\frac{1}{\alpha}} \sigma (K + 1)^{\frac{1}{\alpha}} \ln^{\frac{\alpha-1}{\alpha}} \left( \frac{4(K+1)}{\delta} \right)} \right\}, \quad \eta = \frac{\gamma^2 (1 - \beta_1)^2}{R^2},
$$

(58)

*and*

$$
\lambda = \frac{\sqrt{1 - \beta_1} c_m b_0 R}{40 \gamma \ln \left( \frac{4(K+1)}{\delta} \right)}
$$

(59)

*the bound*

$$
\sum_{k=0}^{K} \gamma C_k \left( f(x_k) - f_* \right) \leq 2R^2
$$

*holds with probability at least $1 - \delta$. In particular, when $\gamma$ equals the minimum from (58), the iterates produced by* Clip-M-AdaGradD/Clip-AdamD *satisfy*

$$
f(\overline{x}_K) - f(x^*) = \mathcal{O} \left( \max \left\{ \frac{L R^2 \ln \frac{K+1}{\delta}}{(1 - \beta_1)^3 (K + 1)}, \frac{\sigma R \ln^{\frac{\alpha-1}{\alpha}} \frac{K+1}{\delta}}{(1 - \beta_1)^{\frac{3}{2}} (K + 1)^{\frac{\alpha-1}{\alpha}}} \right\} \right)
$$

*with probability at least $1 - \delta$, where $\overline{x}_K = \frac{1}{K+1} \sum_{k=0}^{K} x_k$.*

*Proof.* Our proof is induction-based (similarly to the one for Clip-SGD by Sadiev et al. (2023)). We introduce probability event $E_k$ as follows: inequalities

$$
- \sum_{l=0}^{t-1} 2\gamma C_l \left\langle x_l - x^*, \theta_l \right\rangle + \sum_{l=0}^{t-1} 2A_l \|\theta_l\|^2 \leq R^2,
$$

$$
R_t \leq \sqrt{2} R
$$

hold simultaneously $\forall t = 0, 1, \ldots, k$. We want to show that $\mathbb{P}\{E_k\} \geq 1 - \frac{k\delta}{K+1}$ $\forall k = 0, 1, \ldots, K + 1$. The case when $k = 0$ is obvious. Now let us make an induction step: let the statement hold for some $k = T - 1 \leq K$: $\mathbb{P}\{E_{T-1}\} \geq 1 - \frac{(T-1)\delta}{K+1}$. It remains to prove that $\mathbb{P}\{E_T\} \geq 1 - \frac{T\delta}{K+1}$. The event $E_{T-1}$ implies $x_t \in B_{\sqrt{2}R}(x^*)$ $\forall t = 0, \ldots, T - 1$. Hence, $E_{T-1}$ also implies

$$
\|x_T - x^*\| \leq \|x_{T-1} - x^*\| + \frac{\gamma}{b_{T-1}} \|m_{T-1}\| \leq \sqrt{2} R + \frac{\gamma \lambda}{b_{T-1}} \leq \sqrt{2} R + \frac{\gamma \lambda}{c_m b_0} \leq 2R.
$$

Therefore, $E_{T-1}$ implies $\{x_t\}_{t=0}^{T} \subseteq B_{2R}(x^*)$ and we can apply Lemma 16:

$$
\sum_{l=0}^{t-1} \gamma C_l \left( f(x_l) - f_* \right) \leq R_0^2 - R_t^2 - \sum_{l=0}^{t-1} 2\gamma C_l \left\langle x_l - x^*, \theta_l \right\rangle + \sum_{l=0}^{t-1} 2A_l \|\theta_l\|^2
$$

$\forall t = 1, \ldots, T$ and $\forall t = 1, \ldots T - 1$ it implies that

$$
\sum_{l=0}^{t-1} \gamma C_l \left( f(x_l) - f_* \right) \leq R_0^2 - \sum_{l=0}^{t-1} 2\gamma C_l \left\langle x_l - x^*, \theta_l \right\rangle + \sum_{l=0}^{t-1} 2A_l \|\theta_l\|^2 \leq 2R^2.
$$

Taking into account that $\sum_{l=0}^{t-1} \gamma C_l \left( f(x_l) - f_* \right) \geq 0$, we get that $E_{T-1}$ implies

$$
R_T^2 \leq R_0^2 - \sum_{t=0}^{T-1} 2\gamma C_t \left\langle x_t - x^*, \theta_t \right\rangle + \sum_{t=0}^{T-1} 2A_t \|\theta_t\|^2.
$$

(60)

Next, for vectors

$$\eta_t = \begin{cases} x_t - x^*, & \|x_t - x^*\| \le \sqrt{2}R \\ 0, & \text{otherwise} \end{cases}$$

for all $t = 0, 1, \ldots, T-1$, we have that with probability 1

$$\|\eta_t\| \le \sqrt{2}R. \tag{61}$$

Then, $E_{T-1}$ implies that $\eta_t = x_t - x^*$ for all $t = 0, \ldots T-1$. What is more, for all $t = 0, \ldots T-1$ $E_{T-1}$ implies

$$\|\nabla f(x_t)\| \le L \|x_t - x^*\| \le \sqrt{2}LR \overset{(59)}{\le} \frac{\lambda}{2}$$

Hence, using the notation from Appendix A, we have that $E_{T-1}$ implies

$$R_T^2 \le R_0^2 \underbrace{- \sum_{t=0}^{T-1} 2\gamma C_t \langle x_t - x^*, \theta_t^u \rangle}_{\text{①}} \underbrace{- \sum_{t=0}^{T-1} 2\gamma C_t \langle x_t - x^*, \theta_t^b \rangle}_{\text{②}} + \underbrace{\sum_{t=0}^{T-1} 4A_t \left( \|\theta_t^u\|^2 - \mathbb{E}_{\xi_t} \|\theta_t^u\|^2 \right)}_{\text{③}}$$

$$+ \underbrace{\sum_{t=0}^{T-1} 4A_t \mathbb{E}_{\xi_t} \|\theta_t^u\|^2}_{\text{④}} + \underbrace{\sum_{t=0}^{T-1} 4A_t \|\theta_t^b\|^2}_{\text{⑤}}. \tag{62}$$

Next, we bound each term separately with high probability. Before we move on, we also note that event $E_{T-1}$ implies $\|\nabla f(x_t)\| \le \frac{\lambda}{2}$. Therefore, one can apply Lemma 3 and get

$$\|\theta_t^u\| \le 2\lambda, \tag{63}$$

$$\|\theta_t^b\| \le \frac{2^\alpha \sigma^\alpha}{\lambda^{\alpha-1}}, \tag{64}$$

$$\mathbb{E}_{\xi_t} \|\theta_t^u\|^2 \le 18\lambda^{2-\alpha} \sigma^\alpha. \tag{65}$$

**Bound for ①.** The definition of $\theta_t^u$ implies

$$\mathbb{E}_{\xi_t} \left[ -2\gamma C_t \langle \eta_t, \theta_t^u \rangle \right] = 0.$$

Moreover, applying the bound on $C_t$: $C_t \le \frac{1}{c_m b_0}$ from Lemma 16,

$$|-2\gamma C_t \langle \eta_t, \theta_t^u \rangle| \le 2\gamma C_t \|\eta_t\| \|\theta_t^u\| \overset{(61),(63)}{\le} \frac{6\gamma\lambda R}{c_m b_0} \overset{(59)}{\le} \frac{3R^2}{20 \ln\left(\frac{4(K+1)}{\delta}\right)} = c.$$

For $\sigma_t^2 = \mathbb{E}_{\xi_t} \left[ 4\gamma^2 C_t^2 \langle \eta_t, \theta_t^u \rangle^2 \right]$ we also derive

$$\sigma_t^2 \le 4\gamma^2 C_t^2 \mathbb{E}_{\xi_t} \|\theta_t^u\|^2 \|\eta_t\|^2 \le \frac{8\gamma^2 R^2}{c_m^2 b_0^2} \mathbb{E}_{\xi_t} \|\theta_t^u\|^2. \tag{66}$$

Hence, we can apply Bernstein's inequality (Lemma 4) with $c$ defined above and $G = \frac{R^4}{100 \ln\left(\frac{4(K+1)}{\delta}\right)}$:

$$\mathbb{P}\left\{ -\sum_{t=0}^{T-1} \frac{2\gamma}{b_t} \langle x_t - x^*, \theta_t^u \rangle > \frac{R^2}{5} \text{ and } \sum_{t=0}^{T-1} \sigma_t^2 \le G \right\} \le 2\exp\left( -\frac{R^4}{25\left(2G + \frac{2cR^2}{15}\right)} \right)$$

$$= \frac{\delta}{2(K+1)}.$$

Therefore,

$$\mathbb{P}\left\{ \text{either } -\sum_{t=0}^{T-1} \frac{2\gamma}{b_t} \langle x_t - x^*, \theta_t^u \rangle \le \frac{R^2}{5} \text{ or } \sum_{t=0}^{T-1} \sigma_t^2 > G \right\} \ge 1 - \frac{\delta}{2(K+1)}.$$

In addition, event $E_{T-1}$ implies that (due to (66) and (65))

$$
\sum_{t=0}^{T-1} \sigma_t^2 \overset{(59)}{\leq} \frac{144\gamma^2\lambda^{2-\alpha}\sigma^\alpha R^2 T}{c_m^2 b_0^2} \overset{(59)}{\leq} \frac{144(1-\beta_1)^{1-\frac{\alpha}{2}}\gamma^\alpha b_0^{2-\alpha}\sigma^\alpha R^{4-\alpha}T}{40^{2-\alpha} c_m^\alpha b_0^2 \ln^{2-\alpha}\left(\frac{4(K+1)}{\delta}\right)}
$$

$$
\overset{(58)}{\leq} \frac{144(1-\beta_1)R^4 T}{9 \cdot 40^2 (K+1)\ln\left(\frac{4(K+1)}{\delta}\right)} \leq \frac{R^4}{100\ln\left(\frac{4(K+1)}{\delta}\right)}.
$$

**Bound for ②.** For the second term, one can obtain from (58), (59) and $\alpha \leq 2$ that $E_{T-1}$ implies

$$
-\sum_{t=0}^{T-1} 2\gamma C_t \langle x_t - x^*, \theta_t^b \rangle \leq \sum_{t=0}^{T-1} \frac{2\gamma}{c_m b_0}\|\eta_t\|\|\theta_t^b\| \overset{(61),(64)}{\leq} \frac{2\sqrt{2}\cdot 2^\alpha \sigma^\alpha \gamma T R}{c_m b_0 \lambda^{\alpha-1}}
$$

$$
\overset{(59)}{=} \frac{4 \cdot 2^\alpha 40^\alpha \sigma^\alpha \gamma^\alpha T R^{2-\alpha}}{40(1-\beta_1)^{\frac{\alpha}{2}-1} c_m^\alpha b_0^\alpha \ln^{1-\alpha}\left(\frac{4(K+1)}{\delta}\right)} \overset{(58)}{\leq} \frac{4 \cdot 2^\alpha (1-\beta_1)T R^2}{360 \cdot (K+1)}
$$

$$
\leq \frac{2R^2}{45} \leq \frac{R^2}{5}.
$$

**Bound for ③.** For the third part, we have

$$
\mathbb{E}_{\xi_t}\left[4A_t\left(\|\theta_t^u\|^2 - \mathbb{E}_{\xi_t}\|\theta_t^u\|^2\right)\right] = 0.
$$

What is more,

$$
\left|4A_t\left(\|\theta_t^u\|^2 - \mathbb{E}_{\xi_t}\|\theta_t^u\|^2\right)\right| \leq 4A_t\left(\|\theta_t^u\|^2 + \mathbb{E}_{\xi_t}\|\theta_t^u\|^2\right) \overset{(63)}{\leq} \frac{64\gamma^2\lambda^2}{c_m^2 b_0^2 (1-\beta_1)} \overset{(59)}{=} \frac{R^2}{25\ln^2\left(\frac{4(K+1)}{\delta}\right)}
$$

$$
\leq \frac{3R^2}{20\ln\left(\frac{4(K+1)}{\delta}\right)} = c. \tag{67}
$$

We also define

$$
\hat\sigma_t^2 = \mathbb{E}_{\xi_t}\left[16A_t^2\left(\|\theta_t^u\|^2 - \mathbb{E}_{\xi_t}\|\theta_t^u\|^2\right)^2\right].
$$

Hence,

$$
\hat\sigma_t^2 \overset{(67)}{\leq} \frac{3R^2}{20\ln\left(\frac{4(K+1)}{\delta}\right)}\mathbb{E}_{\xi_t}\left[\left|4A_t\left(\|\theta_t^u\|^2 - \mathbb{E}_{\xi_t}\|\theta_t^u\|^2\right)\right|\right]
$$

$$
\leq \frac{12\gamma^2 R^2}{5c_m^2 b_0^2 (1-\beta_1)\ln\left(\frac{4(K+1)}{\delta}\right)}\mathbb{E}_{\xi_t}\|\theta_t^u\|^2.
$$

Therefore, we can apply Bernstein's inequality (Lemma 4) with $c$ defined above and $G = \frac{R^4}{100\ln\left(\frac{4(K+1)}{\delta}\right)}$:

$$
\mathbb{P}\left\{\sum_{t=0}^{T-1} 4A_t\left(\|\theta_t^u\|^2 - \mathbb{E}_{\xi_t}\|\theta_t^u\|^2\right) > \frac{R^2}{5} \text{ and } \sum_{t=0}^{T-1}\hat\sigma_t^2 \leq G\right\} \leq 2\exp\left(-\frac{R^4}{25\left(2G + \frac{2cR^2}{15}\right)}\right)
$$

$$
= \frac{\delta}{2(K+1)}.
$$

Consequently,

$$
\mathbb{P}\left\{\text{either } \sum_{t=0}^{T-1} 4A_t\left(\|\theta_t^u\|^2 - \mathbb{E}_{\xi_t}\|\theta_t^u\|^2\right) \leq \frac{R^2}{5} \text{ or } \sum_{t=0}^{T-1}\hat\sigma_t^2 > G\right\} \geq 1 - \frac{\delta}{2(K+1)}.
$$

Moreover, event $E_{T-1}$ implies that

$$
\begin{aligned}
\sum_{t=0}^{T-1} \hat{\sigma}_t^2 &\leq \sum_{t=0}^{T-1} \frac{12\gamma^2 R^2}{5c_m^2 b_0^2 (1-\beta_1) \ln\left(\frac{4(K+1)}{\delta}\right)} \mathbb{E}_{\xi_t} \|\theta_t^u\|^2 \overset{(65)}{\leq} \frac{18 \cdot 12\gamma^2 \lambda^{2-\alpha} \sigma^\alpha R^2 T}{5c_m^2 b_0^2 (1-\beta_1) \ln\left(\frac{4(K+1)}{\delta}\right)} \\
&\overset{(59)}{=} \frac{18 \cdot 12 \cdot 40^\alpha \gamma^\alpha \sigma^\alpha R^{4-\alpha} T}{5 \cdot 40^2 c_m^\alpha (1-\beta_1)^{\frac{\alpha}{2}} b_0^\alpha \ln^{3-\alpha}\left(\frac{4(K+1)}{\delta}\right)} \overset{(58)}{\leq} \frac{18 \cdot 12 R^4 T}{9 \cdot 5 \cdot 40^2 (K+1) \ln^2\left(\frac{4(K+1)}{\delta}\right)} \\
&\leq \frac{R^4}{100 \ln\left(\frac{4(K+1)}{\delta}\right)}.
\end{aligned}
$$

**Bound for ④.** For the fourth part, we get that $E_{T-1}$ implies

$$
\begin{aligned}
\sum_{t=0}^{T-1} 4A_t E_{\xi_t} \|\theta_t^u\|^2 &\leq \sum_{t=0}^{T-1} \frac{8\gamma^2}{c_m^2 b_0^2 (1-\beta_1)} E_{\xi_t} \|\theta_t^u\|^2 \overset{(65)}{\leq} \frac{144\gamma^2 \lambda^{2-\alpha} \sigma^\alpha T}{c_m^2 b_0^2 (1-\beta_1)} \\
&\overset{(58)}{=} \frac{144\gamma^\alpha 40^\alpha R^{2-\alpha} \sigma^\alpha T}{40^2 c_m^\alpha b_0^\alpha (1-\beta_1)^{\frac{\alpha}{2}} \ln^{2-\alpha}\left(\frac{4(K+1)}{\delta}\right)} \overset{(58)}{\leq} \frac{144 R^2 T}{9 \cdot 40^2 (K+1) \ln\left(\frac{4(K+1)}{\delta}\right)} \\
&\leq \frac{R^2}{100} \leq \frac{R^2}{5}.
\end{aligned}
$$

**Bound for ⑤.** For the last term, $E_{T-1}$ implies

$$
\begin{aligned}
\sum_{t=0}^{T-1} 4A_t \|\theta_t^b\|^2 &\leq \sum_{t=0}^{T-1} \frac{8\gamma^2}{c_m^2 b_0^2 (1-\beta_1)} \|\theta_t^b\|^2 \overset{(64)}{\leq} \frac{8 \cdot 4^\alpha \sigma^{2\alpha} \gamma^2 T}{c_m^2 b_0^2 (1-\beta_1) \lambda^{2(\alpha-1)}} \\
&\overset{(59)}{=} \frac{8 \cdot 4^\alpha 40^{2\alpha} \sigma^{2\alpha} \gamma^{2\alpha} T \ln^{2(\alpha-1)}\left(\frac{4(K+1)}{\delta}\right)}{40^2 c_m^{2\alpha} b_0^{2\alpha} (1-\beta_1)^\alpha R^{2(\alpha-1)}} \\
&\overset{(58)}{\leq} \frac{8 \cdot 4^\alpha R^2 T}{360^2 (K+1)^2} \leq \frac{8R^2}{45^2} \leq \frac{R^2}{5}.
\end{aligned}
$$

Thus, taking into account the bounds above, the probability event $E_{T-1} \cap E_1 \cap E_2$ implies that

$$
R_T^2 \leq R^2 + 5\frac{R^2}{5} = 2R^2,
$$

where

$$
E_1 = \left\{ \text{either } -\sum_{t=0}^{T-1} \frac{2\gamma}{b_t} \langle x_t - x^*, \theta_t^u \rangle \leq \frac{R^2}{5} \text{ or } \sum_{t=0}^{T-1} \sigma_t^2 > \frac{R^4}{100 \ln\left(\frac{4(K+1)}{\delta}\right)} \right\},
$$

$$
E_2 = \left\{ \text{either } \sum_{t=0}^{T-1} \frac{4\gamma^2}{b_t^2} \left( \|\theta_t^u\|^2 - \mathbb{E}_{\xi_t} \|\theta_t^u\|^2 \right) \leq \frac{R^2}{5} \text{ or } \sum_{t=0}^{T-1} \hat{\sigma}_t^2 > \frac{R^4}{100 \ln\left(\frac{4(K+1)}{\delta}\right)} \right\}.
$$

Therefore,

$$
\mathbb{P}\{E_T\} \geq \mathbb{P}\{E_{T-1} \cap E_1 \cap E_2\} = 1 - \mathbb{P}\{\overline{E}_{T-1} \cup \overline{E}_1 \cup \overline{E}_2\}
$$

$$
\geq 1 - \mathbb{P}\{\overline{E}_{T-1}\} - \mathbb{P}\{\overline{E}_1\} - \mathbb{P}\{\overline{E}_2\} \geq 1 - \frac{T\delta}{K+1}.
$$

Hence, for all $k = 0, \ldots, K+1$ we get $\mathbb{P}\{E_k\} \geq 1 - \frac{k\delta}{K+1}$. As the result, event $E_{K+1}$ implies that

$$
\sum_{k=0}^{K} \gamma C_k \left( f(x_k) - f_* \right) \leq 2R^2 \tag{68}
$$

with probability at least $1 - \delta$. Next, from (68) we get that with probability at least $1 - \delta$

$$\sum_{k=0}^{K} (f(x_k) - f_*) \leq \frac{2R^2}{\gamma} \max_{k \in [0, K]} \frac{1}{C_k}.$$

Moreover, $\frac{1}{C_k}$ can be bounded in the following way (from Lemma 16):

$$\frac{1}{C_k} \leq \frac{b_k}{(1 - \beta_1)}.$$

Hence, we get

$$\sum_{k=0}^{K} (f(x_k) - f_*) \leq \frac{2R^2}{\gamma(1 - \beta_1)} \max_{k \in [0, K]} b_k. \tag{69}$$

Also we can bound $b_k$ for Clip-M-AdaGradD using that $g_k = \nabla f(x_k) + \theta_k$ and Assumption 2:

$$b_k^2 \leq b_0^2 + \eta \sum_{k=0}^{K} \left( 4L \left( f(x_k) - f_* \right) + 2\|\theta_k\|^2 \right)$$

and for Clip-AdamD, respectively

$$b_k^2 \leq b_0^2 + \frac{\eta}{K + 1} \sum_{k=0}^{K} \left( 4L \left( f(x_k) - f_* \right) + 2\|\theta_k\|^2 \right).$$

Therefore, due to the fact that the event $E_{K+1}$ implies (see the bounds for ③, ④ and ⑤)

$$\sum_{k=0}^{K} \frac{4\gamma^2}{c_m^2 b_0^2 (1 - \beta_1)} \|\theta_k\|^2 \leq \frac{3R^2}{5},$$

we get

$$b_k^2 \leq b_0^2 + \eta \sum_{k=0}^{K} 4L \left( (f(x_k) - f_*) \right) + \frac{3\eta(1 - \beta_1) b_0^2 R^2}{10\gamma^2}$$

for Clip-M-AdaGradD scheme and

$$b_k^2 \leq b_0^2 + \frac{\eta}{K + 1} \sum_{k=0}^{K} 4L \left( (f(x_k) - f_*) \right) + \frac{3\eta(1 - \beta_1) b_0^2 R^2}{40\gamma^2 (K + 1)}$$

for Clip-AdamD, where we substitute the constant $c_m$ from Lemma 13. Consequently, substituting bounds above in (69), we get

$$\left( \sum_{k=0}^{K} (f(x_k) - f_*) \right)^2 \leq \frac{4R^4}{\gamma^2 (1 - \beta_1)^2} \left( b_0^2 + \eta \sum_{k=0}^{K} (4L \left( f(x_k) - f_* \right)) + \frac{3\eta(1 - \beta_1) R^2 b_0^2}{10\gamma^2} \right)$$

for Clip-M-AdaGradD and

$$\left( \sum_{k=0}^{K} (f(x_k) - f_*) \right)^2 \leq \frac{4R^4}{\gamma^2 (1 - \beta_1)^2} \left( b_0^2 + \frac{\eta}{K + 1} \sum_{k=0}^{K} (4L \left( f(x_k) - f_* \right)) + \frac{3\eta(1 - \beta_1) R^2 b_0^2}{40\gamma^2 (K + 1)} \right)$$

for Clip-AdamD, respectively. Solving these quadratic inequalities, we have that $E_{K+1}$ implies

$$\sum_{k=0}^{K} (f(x_k) - f_*) \leq \frac{2R^2}{\gamma^2} \left( \frac{4L\eta R^2}{(1 - \beta_1)^2} + \sqrt{\frac{16L^2 \eta^2 R^4}{(1 - \beta_1)^4} + b_0^2 \left( \frac{\gamma^2}{(1 - \beta_1)^2} + \frac{3\eta R^2}{10(1 - \beta_1)} \right)} \right)$$

$$\leq \frac{6R^2}{\gamma^2} \max \left\{ \frac{8L\eta R^2}{(1 - \beta_1)^2}, \frac{b_0 \gamma}{1 - \beta_1}, b_0 R \sqrt{\frac{\eta}{1 - \beta_1}} \right\}$$

and

$$\sum_{k=0}^{K} (f(x_k) - f_*) \leq \frac{2R^2}{\gamma^2} \left( \frac{4L\eta R^2}{(1-\beta_1)^2(K+1)} \right.$$
$$\left. + \sqrt{\frac{16L^2\eta^2 R^4}{(1-\beta_1)^4(K+1)^2} + b_0^2 \left( \frac{\gamma^2}{(1-\beta_1)^2} + \frac{3\eta R^2}{40(1-\beta_1)(K+1)} \right)} \right)$$
$$\leq \frac{6R^2}{\gamma^2} \max \left\{ \frac{8L\eta R^2}{(1-\beta_1)^2(K+1)}, \frac{b_0\gamma}{1-\beta_1}, b_0 R \sqrt{\frac{\eta}{(1-\beta_1)(K+1)}} \right\}.$$

with probability at least $1 - \delta$. Choosing $\eta = \frac{\gamma^2(1-\beta_1)^2}{R^2}$, $\gamma$ equal to the minimum from (58) and using that $2\sqrt{ab} \leq a + b$, we obtain the bound for Clip-M-AdaGradD/Clip-AdamD for the convex case:

$$\frac{1}{K+1} \sum_{k=0}^{K} (f(x_k) - f_*) = \mathcal{O} \left( \max \left\{ \frac{LR^2 \ln \frac{K+1}{\delta}}{(1-\beta_1)^3(K+1)}, \frac{\sigma R \ln^{\frac{\alpha-1}{\alpha}} \frac{K+1}{\delta}}{(1-\beta_1)^{\frac{3}{2}}(K+1)^{\frac{\alpha-1}{\alpha}}} \right\} \right)$$

with probability at least $1 - \delta$. To get the final result, it remains to apply Jensen's inequality. $\square$

## C.4 NON-CONVEX CASE: METHODS WITHOUT DELAY

**Lemma 17** (Descent lemma). *Let Assumptions 2 and 4 hold. Then, after $T$ iterations of* Clip-M-AdaGrad/Clip-Adam*, we have*

$$\sum_{t=0}^{T-1} \frac{\gamma C_t}{2} \|\nabla f(x_t)\|^2 \leq \left( 2M + \frac{2L\gamma^2}{\eta(1-\beta_1)} \right) \sqrt{b_{-1}^2 + \eta \sum_{t=0}^{T-1} \|g_t\|^2} - \sum_{t=0}^{T-1} \gamma C_t \langle \nabla f(x_t), \theta_t^u \rangle$$
$$+ \sum_{t=0}^{T-1} \frac{\gamma C_t}{2} \|\theta_t^b\|^2$$

*for* Clip-M-AdaGrad*, where $C_t = \sum_{k=t}^{T-1} (1-\beta_1)\beta_1^{k-t}$, and*

$$\sum_{t=0}^{T-1} \frac{\gamma C_t}{2} \|\nabla f(x_t)\|^2 \leq \left( 3M + \frac{16KL\gamma^2}{\eta(1-\beta_1)} \right) \sqrt{b_{-1}^2 + \frac{\eta}{K} \sum_{t=0}^{T-1} \|g_t\|^2} - \sum_{t=0}^{T-1} \gamma C_t \langle \nabla f(x_t), \theta_t^u \rangle$$
$$+ \sum_{t=0}^{T-1} \frac{\gamma C_t}{2} \|\theta_t^b\|^2$$

Clip-Adam*, where $C_t = \sum_{k=t}^{T-1} (1-\beta_1)\beta_1^{k-t} / (\sqrt{\beta_2})^k$.*

*Proof.* The first part of the proof is similar to the Lemma 15. We start with the $L$-smoothness of $f$:

$$f(x_{t+1}) - f(x_t) \leq \langle \nabla f(x_t), x_{t+1} - x_t \rangle + \frac{L}{2} \|x_{t+1} - x_t\|^2$$
$$= -\frac{\gamma}{b_t} \langle \nabla f(x_t), m_t \rangle + \frac{L\gamma^2}{2b_t^2} \|m_t\|^2. \tag{70}$$

Using the update rule of Algorithm 3, we can obtain

$$
\begin{aligned}
-\langle \nabla f(x_t), m_t \rangle &= -\beta_1 \langle \nabla f(x_t), m_{t-1} \rangle - (1 - \beta_1) \langle \nabla f(x_t), g_t \rangle \\
&= -\beta_1 \langle \nabla f(x_t) - \nabla f(x_{t-1}), m_{t-1} \rangle - \beta_1 \langle \nabla f(x_{t-1}), m_{t-1} \rangle \\
&\quad - (1 - \beta_1) \langle \nabla f(x_t), g_t \rangle \\
&\leq -\beta_1 \langle \nabla f(x_{t-1}), m_{t-1} \rangle + \beta_1 \| \nabla f(x_t) - \nabla f(x_{t-1}) \| \| m_{t-1} \| \\
&\quad - (1 - \beta_1) \langle \nabla f(x_t), g_t \rangle \\
&\leq -\beta_1 \langle \nabla f(x_{t-1}), m_{t-1} \rangle + \beta_1 L \| x_t - x_{t-1} \| \| m_{t-1} \| \\
&\quad - (1 - \beta_1) \langle \nabla f(x_t), g_t \rangle \\
&= -\beta_1 \langle \nabla f(x_{t-1}), m_{t-1} \rangle + \frac{\gamma \beta_1 L}{b_{t-1}} \| m_{t-1} \|^2 \\
&\quad - (1 - \beta_1) \langle \nabla f(x_t), g_t \rangle,
\end{aligned}
$$

where we use the Cauchy-Schwarz inequality and $L$-smoothness of $f$. Applying the same idea for the $t-1, t-2, \ldots, 0$ and noting that $m_{-1} = 0$, we get

$$
-\langle \nabla f(x_t), m_t \rangle \leq -(1 - \beta_1) \sum_{k=0}^{t} \beta_1^{t-k} \langle \nabla f(x_k), g_k \rangle + L\gamma \sum_{k=0}^{t-1} \frac{\beta_1^{t-k}}{b_k} \| m_k \|^2. \tag{71}
$$

Therefore, substituting (71) into (70), we have

$$
\begin{aligned}
f(x_{t+1}) - f(x_t) &\leq -\frac{(1 - \beta_1)\gamma}{b_t} \sum_{k=0}^{t} \beta_1^{t-k} \langle \nabla f(x_k), g_k \rangle + \frac{L\gamma^2}{b_t} \sum_{k=0}^{t-1} \frac{\beta_1^{t-k}}{b_k} \| m_k \|^2 + \frac{L\gamma^2}{2b_t^2} \| m_t \|^2 \\
&\leq -\frac{(1 - \beta_1)\gamma}{b_t} \sum_{k=0}^{t} \beta_1^{t-k} \langle \nabla f(x_k), g_k \rangle + \frac{L\gamma^2}{b_t} \sum_{k=0}^{t} \frac{\beta_1^{t-k}}{b_k} \| m_k \|^2.
\end{aligned}
$$

Applying Lemma 14 with $1 - \beta_1^{k+1} \leq 1$, we can rewrite the inequality above as follows:

$$
\begin{aligned}
f(x_{t+1}) - f(x_t) &\leq -\frac{(1 - \beta_1)\gamma}{b_t} \sum_{k=0}^{t} \beta_1^{t-k} \langle \nabla f(x_k), g_k \rangle + \frac{L\gamma^2}{b_t} \sum_{k=0}^{t} \frac{\beta_1^{t-k}}{b_k} \sum_{j=0}^{k} \beta_1^{k-j} (1 - \beta_1) \| g_j \|^2 \\
&= -\frac{(1 - \beta_1)\gamma}{b_t} \sum_{k=0}^{t} \beta_1^{t-k} \langle \nabla f(x_k), g_k \rangle + \frac{L\gamma^2}{b_t} \sum_{j=0}^{t} \sum_{k=j}^{t} \frac{\beta_1^{t-k}}{b_k} \beta_1^{k-j} (1 - \beta_1) \| g_j \|^2,
\end{aligned}
$$

where we change the limits of summation. Multiplying both sides of the inequality above by $\frac{b_t}{p_t}$, where

$$
p_t = \begin{cases} 1, & \text{for Clip-M-AdaGrad} \\ (\sqrt{\beta_2})^t, & \text{for Clip-Adam} \end{cases} \tag{72}
$$

and using that $b_k \geq c_m b_j$ (see Lemma 13), one can obtain

$$
\begin{aligned}
\frac{b_t}{p_t} (f(x_{t+1}) - f(x_t)) &\leq -\frac{(1 - \beta_1)\gamma}{p_t} \sum_{k=0}^{t} \beta_1^{t-k} \langle \nabla f(x_k), g_k \rangle \\
&\quad + \frac{L\gamma^2}{p_t} \sum_{j=0}^{t} \frac{\beta_1^{t-j}}{c_m b_j} (1 - \beta_1)(t - j + 1) \| g_j \|^2.
\end{aligned}
$$

After summing over $t$,

$$
\begin{aligned}
\sum_{t=0}^{T-1} \frac{b_t}{p_t} (f(x_{t+1}) - f(x_t)) &\leq -(1 - \beta_1)\gamma \sum_{t=0}^{T-1} \sum_{k=0}^{t} \frac{\beta_1^{t-k}}{p_t} \langle \nabla f(x_k), g_k \rangle \\
&\quad + L\gamma^2 \sum_{t=0}^{T-1} \sum_{j=0}^{t} \frac{\beta_1^{t-j}}{c_m b_j p_t} (1 - \beta_1)(t - j + 1) \| g_j \|^2.
\end{aligned}
$$

Next, applying the same idea as in Lemma 15, we get that multiplicative factors are equal to

$$-\gamma C_r = -\sum_{t=r}^{T-1} \frac{\gamma(1-\beta_1)\beta_1^{t-r}}{p_t} \tag{73}$$

for the scalar product $\langle \nabla f(x_r), g_r \rangle$ and

$$A_r = \sum_{t=r}^{T-1} \frac{L\gamma^2(1-\beta_1)}{c_m b_r p_t}(t-r+1)\beta_1^{t-r} \tag{74}$$

for the squared norm $\|g_r\|^2$, respectively. Moreover, it can be shown that $p_t \geq c_m$ for corresponding update rule of $b_t$. Hence, for (74) we apply Lemma 1 to obtain the next bound:

$$A_r \leq \frac{L\gamma^2}{c_m^2 b_r(1-\beta_1)}.$$

Therefore, rewriting the descent lemma in terms of (73) and (74), we have

$$\sum_{t=0}^{T-1} b_t(f(x_{t+1})-f(x_t)) \leq -\sum_{t=0}^{T-1} \gamma C_t \langle \nabla f(x_t), g_t \rangle + \frac{L\gamma^2}{c_m^2(1-\beta_1)}\sum_{t=0}^{T-1} \frac{\|g_t\|^2}{b_t}.$$

Using that $g_t = \nabla f(x_t) + \theta_t$, we get

$$\sum_{t=0}^{T-1} \gamma C_t\|\nabla f(x_t)\|^2 \leq \sum_{t=0}^{T-1} \frac{b_t}{p_t}(f(x_t)-f(x_{t+1})) - \sum_{t=0}^{T-1} \gamma C_t \langle \nabla f(x_t), \theta_t \rangle + \frac{L\gamma^2}{c_m^2(1-\beta_1)}\sum_{t=0}^{T-1} \frac{\|g_t\|^2}{b_t}$$

$$= \sum_{t=0}^{T-1} \frac{b_t}{p_t}(f(x_t)-f_* - (f(x_{t+1})-f_*)) - f(x_{t+1})) - \sum_{t=0}^{T-1} \gamma C_t \langle \nabla f(x_t), \theta_t \rangle$$

$$+ \frac{L\gamma^2}{c_m^2(1-\beta_1)}\sum_{t=0}^{T-1} \frac{\|g_t\|^2}{b_t}$$

$$\leq \frac{b_0}{p_0}(f(x_0)-f_*) + \sum_{t=1}^{T-1}\left(\frac{b_t}{p_t}-\frac{b_{t-1}}{p_{t-1}}\right)(f(x_t)-f_*) - \sum_{t=0}^{T-1} \gamma C_t \langle \nabla f(x_t), \theta_t \rangle$$

$$+ \frac{L\gamma^2}{c_m^2(1-\beta_1)}\sum_{t=0}^{T-1} \frac{\|g_t\|^2}{b_t}.$$

Since $p_t = 1$ for Clip-M-AdaGrad, we can use that $b_t \geq b_{t-1}$, and for Clip-Adam we get $b_t \geq \sqrt{\beta_2}b_{t-1}$, what is equal to $\frac{b_t}{p_t} \geq \frac{b_{t-1}}{p_{t-1}}$ with $p_t = (\sqrt{\beta_2})^t$. Therefore, applying Assumption 4, we obtain

$$\sum_{t=0}^{T-1} \gamma C_t\|\nabla f(x_t)\|^2 \leq \frac{b_0 M}{p_0} + \frac{b_{T-1}M}{p_{T-1}} - \sum_{t=0}^{T-1} \gamma C_t \langle \nabla f(x_t), \theta_t \rangle + \frac{L\gamma^2}{c_m^2(1-\beta_1)}\sum_{t=0}^{T-1} \frac{\|g_t\|^2}{b_t}.$$

Now we construct descent lemmas for each considering update separately. For Clip-M-AdaGrad we directly apply Lemma 2 to bound the last term:

$$\sum_{t=0}^{T-1} \gamma C_t\|\nabla f(x_t)\|^2 \leq 2Mb_{T-1} - \sum_{t=0}^{T-1} \gamma C_t \langle \nabla f(x_t), \theta_t \rangle + \frac{L\gamma^2}{\eta(1-\beta_1)}b_{T-1}$$

$$= \left(2M + \frac{2L\gamma^2}{\eta(1-\beta_1)}\right)b_{T-1} - \sum_{t=0}^{T-1} \gamma C_t \langle \nabla f(x_t), \theta_t \rangle$$

$$\leq \left(2M + \frac{2L\gamma^2}{\eta(1-\beta_1)}\right)b_{T-1} - \sum_{t=0}^{T-1} \gamma C_t \langle \nabla f(x_t), \theta_t^u \rangle$$

$$+ \sum_{t=0}^{T-1} \frac{\gamma C_t}{2}\|\nabla f(x_t)\|^2 + \sum_{t=0}^{T-1} \frac{\gamma C_t}{2}\|\theta_t^b\|^2, \tag{75}$$

where we use that $c_m = 1$ and $p_t = 1$ for Clip-M-AdaGrad. For the Clip-Adam, we get

$$
\sum_{t=0}^{T-1} \frac{\|g_t\|^2}{b_t} = \frac{1}{\eta} \sum_{t=0}^{T-1} \frac{\eta\|g_t\|^2}{\sqrt{\beta_2^{t+1} b_{-1}^2 + (1-\beta_2)\eta \sum_{k=0}^{t} \beta_2^{t-k} \|g_k\|^2}}
$$

$$
\leq \frac{K}{\eta} \sum_{t=0}^{T-1} \frac{2\frac{\eta}{K}\|g_t\|^2}{\sqrt{b_{-1}^2 + \frac{\eta}{K} \sum_{k=0}^{t} \|g_k\|^2}}
$$

$$
\leq \frac{4K}{\eta} \sqrt{b_{-1}^2 + \frac{\eta}{K} \sum_{t=0}^{T-1} \|g_t\|^2},
$$

where we use that $\beta_2^k \geq 1/4$ for all $k = 0, \ldots, K$. Consequently, with upper bound on $b_t$ and $c_m = 1/2$, for Clip-Adam one can obtain

$$
\sum_{t=0}^{T-1} \gamma C_t \|\nabla f(x_t)\|^2 \leq b_0 M + \frac{b_{T-1} M}{(\sqrt{\beta_2})^{T-1}} - \sum_{t=0}^{T-1} \gamma C_t \langle \nabla f(x_t), \theta_t \rangle
$$

$$
+ \frac{16KL\gamma^2}{\eta(1-\beta_1)} \sqrt{b_{-1}^2 + \frac{\eta}{K} \sum_{k=0}^{t} \|g_k\|^2}
$$

$$
\leq \left(3M + \frac{16KL\gamma^2}{\eta(1-\beta_1)}\right) \sqrt{b_{-1}^2 + \frac{\eta}{K} \sum_{t=0}^{T-1} \|g_t\|^2} - \sum_{t=0}^{T-1} \gamma C_t \langle \nabla f(x_t), \theta_t \rangle
$$

$$
\leq \left(3M + \frac{16KL\gamma^2}{\eta(1-\beta_1)}\right) \sqrt{b_{-1}^2 + \frac{\eta}{K} \sum_{t=0}^{T-1} \|g_t\|^2} - \sum_{t=0}^{T-1} \gamma C_t \langle \nabla f(x_t), \theta_t^u \rangle
$$

$$
+ \sum_{t=0}^{T-1} \frac{\gamma C_t}{2} \|\nabla f(x_t)\|^2 + \sum_{t=0}^{T-1} \frac{\gamma C_t}{2} \|\theta_t^b\|^2.
$$

After substitution of the analytical form of $b_{T-1}$ in (75) and different options of $p_t$, we claim the final result. $\qquad\square$

**Theorem 11.** *Let Assumptions 1, 2 and 4 hold. Then, after $K$ iterations of* Clip-M-AdaGrad/Clip-Adam *with*

$$
\gamma \leq \min\left\{ \frac{b_{-1} K^{\frac{1-\alpha}{3\alpha-2}}}{48L \ln\left(\frac{4}{\delta}\right)}, \frac{b_{-1}\sqrt{M}}{4^{\frac{1}{\alpha}} \cdot 12\sqrt{L}\sigma(K+1)^{\frac{\alpha}{3\alpha-2}} \ln^{\frac{\alpha-1}{\alpha}}\left(\frac{4}{\delta}\right)}, \right.
$$

$$
\left. \frac{b_{-1} M^{\frac{\alpha}{2\alpha-1}}}{4^{\frac{\alpha}{2\alpha-1}} \cdot 12^{\frac{2\alpha-2}{2\alpha-1}} \sigma^{\frac{2\alpha}{2\alpha-1}} L^{\frac{\alpha-1}{2\alpha-1}} (K+1)^{\frac{\alpha}{3\alpha-2}} \ln^{\frac{2\alpha-2}{2\alpha-1}}\left(\frac{4}{\delta}\right)} \right\}, \quad \eta = \frac{L\gamma^2}{M(1-\beta_1)}, \quad (76)
$$

*and*

$$
\lambda = \frac{b_{-1}\sqrt{M}(K+1)^{\frac{1-\alpha}{3\alpha-2}}}{12\sqrt{L}\gamma \ln\left(\frac{4}{\delta}\right)} \quad (77)
$$

*the bound*

$$
\frac{1}{K} \sum_{k=0}^{K-1} \|\nabla f(x_k)\|^2
$$

$$
= \mathcal{O}\left( \frac{1}{(1-\beta_1)^{\frac{3}{2}}} \max\left\{ \frac{LM \ln\left(\frac{4}{\delta}\right)}{K^{\frac{2\alpha-1}{3\alpha-2}}}, \frac{\sqrt{LM}\sigma \ln^{\frac{\alpha-1}{\alpha}}\left(\frac{4}{\delta}\right)}{K^{\frac{2\alpha-2}{3\alpha-2}}}, \frac{\sigma^{\frac{2\alpha}{2\alpha-1}}(LM)^{\frac{\alpha-1}{2\alpha-1}} \ln^{\frac{2\alpha-2}{2\alpha-1}}\left(\frac{4}{\delta}\right)}{K^{\frac{2\alpha-2}{3\alpha-2}}} \right\} \right)
$$

*holds with probability at least $1 - \delta$.*

*Proof.* The main idea of the proof is similar to the proof of Theorem 9, but we do not need to introduce any probabilistic events since according to Assumption 4 the norm of gradient is always bounded:

$$\|\nabla f(x_t)\| \leq \sqrt{2L\left(f(x_t) - f_*\right)} \leq \sqrt{2LM} \overset{(77)}{\leq} \frac{\lambda}{2}.$$

Therefore, one can apply Lemma 3 and get

$$\|\theta_t^u\| \leq 2\lambda, \tag{78}$$

$$\|\theta_t^b\| \leq \frac{2^\alpha \sigma^\alpha}{\lambda^{\alpha-1}}, \tag{79}$$

$$\mathbb{E}_{\xi_t} \|\theta_t^u\|^2 \leq 18\lambda^{2-\alpha}\sigma^\alpha. \tag{80}$$

According to the Lemma 17, we get

$$\sum_{t=0}^{T-1} \frac{\gamma C_t}{2} \|\nabla f(x_t)\|^2 \leq \left(2M + \frac{2L\gamma^2}{\eta(1-\beta_1)}\right) \sqrt{b_{-1}^2 + \eta \sum_{t=0}^{T-1} \|g_t\|^2} - \sum_{t=0}^{T-1} \gamma C_t \left\langle \nabla f(x_t), \theta_t^u \right\rangle$$

$$+ \sum_{t=0}^{T-1} \frac{\gamma C_t}{2} \|\theta_t^b\|^2$$

with $C_t = \sum_{k=t}^{T-1} (1-\beta_1)\beta_1^{k-t}$ for Clip-M-AdaGrad and

$$\sum_{t=0}^{T-1} \frac{\gamma C_t}{2} \|\nabla f(x_t)\|^2 \leq \left(3M + \frac{16KL\gamma^2}{\eta(1-\beta_1)}\right) \sqrt{b_{-1}^2 + \frac{\eta}{K} \sum_{t=0}^{T-1} \|g_t\|^2} - \sum_{t=0}^{T-1} \gamma C_t \left\langle \nabla f(x_t), \theta_t^u \right\rangle$$

$$+ \sum_{t=0}^{T-1} \frac{\gamma C_t}{2} \|\theta_t^b\|^2$$

with $C_t = \sum_{k=t}^{T-1} (1-\beta_1)\beta_1^{k-t}/(\sqrt{\beta_2})^k$ for Clip-Adam. Let us bound $C_t$ regardless of the method. In can be shown that

$$1 - \beta_1 \leq C_t(\text{Clip-M-AdaGrad}) \leq \sum_{k=0}^{\infty} (1-\beta_1)\beta_1^k = 1$$

and

$$1 - \beta_1 \leq C_t(\text{Clip-Adam}) \leq 2 \sum_{k=0}^{\infty} (1-\beta_1)\beta_1^k = 2,$$

since $(\sqrt{\beta_2})^{T-1} \geq 1/2$. Therefore, descent lemmas for Clip-M-AdaGrad and Clip-Adam can be rewritten in the following way:

$$\frac{\gamma(1-\beta_1)}{2} \sum_{t=0}^{T-1} \|\nabla f(x_t)\|^2 \leq \left(2M + \frac{2L\gamma^2}{\eta(1-\beta_1)}\right) \sqrt{b_{-1}^2 + \eta \sum_{t=0}^{T-1} \|g_t\|^2}$$

$$- \sum_{t=0}^{T-1} \gamma C_t \left\langle \nabla f(x_t), \theta_t^u \right\rangle + \sum_{t=0}^{T-1} \gamma \|\theta_t^b\|^2 \tag{81}$$

for Clip-M-AdaGrad and

$$\frac{\gamma(1-\beta_1)}{2} \sum_{t=0}^{T-1} \|\nabla f(x_t)\|^2 \leq \left(3M + \frac{16KL\gamma^2}{\eta(1-\beta_1)}\right) \sqrt{b_{-1}^2 + \frac{\eta}{K} \sum_{t=0}^{T-1} \|g_t\|^2}$$

$$- \sum_{t=0}^{T-1} \gamma C_t \left\langle \nabla f(x_t), \theta_t^u \right\rangle + \sum_{t=0}^{T-1} \gamma \|\theta_t^b\|^2 \tag{82}$$

for Clip-Adam. Moreover, $\sum_{t=0}^{T-1} \|g_t\|^2$ can be bounded as follows:

$$\sum_{t=0}^{T-1} \|g_t\|^2 \leq 3 \sum_{t=0}^{T-1} \left( \|\nabla f(x_t)\|^2 + \left( \|\theta_t^u\|^2 - \mathbb{E}_{\xi_t} \|\theta_t^u\|^2 \right) + \mathbb{E}_{\xi_t} \|\theta_t^u\|^2 + \|\theta_t^b\|^2 \right). \quad (83)$$

The main idea is to give upper bounds for the next terms for all $T \leq K$:

$$\underbrace{\sum_{t=0}^{T-1} \frac{L\gamma^2}{b_{-1}^2} \left( \|\theta_t^u\|^2 - \mathbb{E}_{\xi_t} \|\theta_t^u\|^2 \right)}_{①}, \underbrace{\sum_{t=0}^{T-1} \frac{L\gamma^2}{b_{-1}^2} \mathbb{E}_{\xi_t} \|\theta_t^u\|^2}_{②}, \underbrace{\sum_{t=0}^{T-1} \frac{\gamma}{b_{-1}} \|\theta_t^b\|^2}_{③}, \underbrace{-\sum_{t=0}^{T-1} \frac{\gamma}{b_{-1}} C_t \langle \nabla f(x_t), \theta_t^u \rangle}_{④}.$$

In cases of ①, ② and ③ we multiply sums from (83) to the factors to move to the corresponding type of sums from Theorem 9.

**Bound for ①.** We have bounded and unbiased terms in the sum:

$$\mathbb{E}_{\xi_t} \left[ \frac{L\gamma^2}{b_{-1}^2} \left( \|\theta_t^u\|^2 - \mathbb{E}_{\xi_t} \|\theta_t^u\|^2 \right) \right] = 0$$

and

$$\left| \frac{L\gamma^2}{b_{-1}^2} \left( \|\theta_t^u\|^2 - \mathbb{E}_{\xi_t} \|\theta_t^u\|^2 \right) \right| \overset{(78)}{\leq} \frac{8L\gamma^2\lambda^2}{b_{-1}^2} \leq \frac{24M}{19 \ln \frac{4}{\delta}} = c.$$

Next, we define $\hat{\sigma}_t^2 = \mathbb{E}_{\xi_t} \left[ \frac{L^2\gamma^4}{b_{-1}^4} \left( \|\theta_t^u\|^2 - \mathbb{E}_{\xi_t} \|\theta_t^u\|^2 \right) \right]$. For the introduced quantities, we have

$$\hat{\sigma}_t^2 \leq \frac{cL\gamma^2}{b_{-1}^2} \mathbb{E}_{\xi_t} \left| \|\theta_t^u\|^2 - \mathbb{E}_{\xi_t} \|\theta_t^u\|^2 \right| \leq \frac{2cL\gamma^2}{b_{-1}^2} \mathbb{E}_{\xi_t} \|\theta_t^u\|^2.$$

Therefore, we can apply Bernstein's inequality (Lemma 4) with $G = \frac{3M^2}{38 \ln\left(\frac{4}{\delta}\right)}$:

$$\mathbb{P} \left\{ \left| \sum_{t=0}^{T-1} \frac{L\gamma^2}{b_{-1}^2} \left( \|\theta_t^u\|^2 - \mathbb{E}_{\xi_t} \|\theta_t^u\|^2 \right) \right| > M \text{ and } \sum_{t=0}^{T-1} \hat{\sigma}_t^2 \leq G \right\} \leq 2 \exp \left( -\frac{M^2}{2G + \frac{2cM}{3}} \right) = \frac{\delta}{2}.$$

Thus, we get

$$\mathbb{P} \left\{ \text{either} \left| \sum_{t=0}^{T-1} \frac{L\gamma^2}{b_{-1}^2} \left( \|\theta_t^u\|^2 - \mathbb{E}_{\xi_t} \|\theta_t^u\|^2 \right) \right| \leq M \text{ or } \sum_{t=0}^{T-1} \hat{\sigma}_t^2 > G \right\} \geq 1 - \frac{\delta}{2}.$$

Moreover,

$$\sum_{t=0}^{T-1} \hat{\sigma}_t^2 \overset{(80)}{\leq} \frac{36cTL\gamma^2\lambda^{2-\alpha}\sigma^\alpha}{b_{-1}^2} \overset{(77)}{\leq} \frac{36cTL\gamma^\alpha\sqrt{M}^{2-\alpha}K^{\frac{(1-\alpha)(2-\alpha)}{3\alpha-2}}}{12^{2-\alpha}b_{-1}^\alpha\sqrt{L}^{2-\alpha}\ln^{2-\alpha}\left(\frac{4}{\delta}\right)}$$

$$\overset{(76)}{\leq} \frac{3M^2}{38 \ln\left(\frac{4}{\delta}\right)}.$$

**Bound for ②.** For the second term, we get

$$\sum_{t=0}^{T-1} \frac{L\gamma^2}{b_{-1}^2} \mathbb{E}_{\xi_t} \|\theta_t^u\|^2 \overset{(80)}{\leq} \frac{18TL\gamma^2\lambda^{2-\alpha}\sigma^\alpha}{b_{-1}^2} \overset{(77)}{\leq} \frac{18TL\gamma^\alpha\sqrt{M}^{2-\alpha}K^{\frac{(1-\alpha)(2-\alpha)}{3\alpha-2}}}{12^{2-\alpha}b_{-1}^\alpha\sqrt{L}^{2-\alpha}\ln^{2-\alpha}\left(\frac{4}{\delta}\right)}$$

$$\overset{(76)}{\leq} \frac{M}{32} \leq M.$$

**Bound for ③.** For the third sum, we obtain

$$\sum_{t=0}^{T-1} \frac{\gamma}{b_{-1}} \|\theta_t^b\|^2 \overset{(79)}{\leq} \frac{4^\alpha\sigma^{2\alpha}\gamma T}{b_{-1}\lambda^{2\alpha-2}} \overset{(77)}{=} \frac{4^\alpha 12^{2\alpha-2}\sigma^{2\alpha}\gamma^{2\alpha-1}TL^{\alpha-1}\ln^{2\alpha-2}\left(\frac{4}{\delta}\right)}{b_{-1}^{2\alpha-1}M^{\alpha-1}K^{\frac{(1-\alpha)(2\alpha-2)}{3\alpha-2}}} \overset{(76)}{\leq} M,$$

where we choose the third option for $\gamma$.

**Bound for** ④. Similarly to ①, we have unbiased and bounded terms in sum:

$$\mathbb{E}_{\xi_t}\left[-\frac{\gamma C_t}{b_{-1}}\langle\nabla f(x_t),\theta_t^u\rangle\right] = 0$$

and

$$\left|-\frac{\gamma C_t}{b_{-1}}\langle\nabla f(x_t),\theta_t^u\rangle\right| \le \frac{2\gamma}{b_{-1}}\|\nabla f(x_t)\|\,\|\theta_t^u\| \overset{(78)}{\le} \frac{4\gamma\lambda\sqrt{2LM}}{b_{-1}} \le \frac{3M}{4\ln\left(\frac{4}{\delta}\right)} = c.$$

Let us define $\sigma_t^2 = \mathbb{E}_{\xi_t}\left[\frac{\gamma^2 C_t^2}{b_{-1}^2}\langle\nabla f(x_t),\theta_t^u\rangle^2\right]$. Hence,

$$\sigma_t^2 \le \frac{8\gamma^2 LM}{b_{-1}^2}\mathbb{E}_{\xi_t}\|\theta_t^u\|^2.$$

Therefore, we can apply Bernstein's inequality (Lemma 4) with $G = \frac{M^2}{4\ln\left(\frac{4}{\delta}\right)}$:

$$\mathbb{P}\left\{\left|-\sum_{t=0}^{T-1}\frac{\gamma C_t}{b_{-1}}\langle\nabla f(x_t),\theta_t^u\rangle\right| > M \text{ and } \sum_{t=0}^{T-1}\sigma_t^2 \le G\right\} \le 2\exp\left(-\frac{M^2}{2G+\frac{2cM}{3}}\right) = \frac{\delta}{2}.$$

Thus, we get

$$\mathbb{P}\left\{\text{either } \left|-\sum_{t=0}^{T-1}\frac{\gamma C_t}{b_{-1}}\langle\nabla f(x_t),\theta_t^u\rangle\right| \le M \text{ or } \sum_{t=0}^{T-1}\sigma_t^2 > G\right\} \ge 1-\frac{\delta}{2}.$$

Moreover,

$$\sum_{t=0}^{T-1}\sigma_t^2 \overset{(80)}{\le} \frac{144\gamma^2 LMT\lambda^{2-\alpha}\sigma^\alpha}{b_{-1}^2} \overset{(77)}{=} \frac{144\sqrt{M}^{2-\alpha}K^{\frac{(1-\alpha)(2-\alpha)}{3\alpha-2}}\gamma^\alpha LMT\sigma^\alpha}{12^{2-\alpha}b_{-1}^\alpha\sqrt{L}^{2-\alpha}\ln^{2-\alpha}\left(\frac{4}{\delta}\right)} \overset{(76)}{\le} \frac{M^2}{4\ln\left(\frac{4}{\delta}\right)}.$$

Consequently, next inequality holds with probability at least $1-\delta$ for all $T \le K$:

$$\sum_{t=0}^{T-1}\|g_t\|^2 \le 3\sum_{t=0}^{T-1}\|\nabla f(x_t)\|^2 + \frac{6Mb_{-1}^2}{L\gamma^2} + \frac{3Mb_{-1}}{\gamma}.$$

Let us specify $\eta$ for each method. This parameter can be chosen as follows:

$$\eta = \begin{cases} \frac{L\gamma^2}{M(1-\beta_1)}, & \text{for Clip-M-AdaGrad} \\ \frac{KL\gamma^2}{M(1-\beta_1)}, & \text{for Clip-Adam} \end{cases}$$

Therefore, (81) and (82) can be rewritten in an unified form with $T = K$ and ①, ②, ③ and ④:

$$\frac{\gamma(1-\beta_1)}{2}\sum_{k=0}^{K-1}\|\nabla f(x_k)\|^2 \le 19M\sqrt{b_{-1}^2 + \frac{3L\gamma^2}{M(1-\beta_1)}\sum_{k=0}^{K-1}\|\nabla f(x_k)\|^2 + \frac{6b_{-1}^2}{1-\beta_1} + \frac{3L\gamma b_{-1}}{1-\beta_1}}$$
$$+ 2Mb_{-1}$$

holds with probability at least $1-\delta$ for both algorithms. Denoting $\sum\limits_{k=0}^{K-1}\|\nabla f(x_k)\|^2$ as $S_K$ and squaring the inequality above, we get

$$\frac{\gamma^2(1-\beta_1)^2}{4}S_K^2 \le \left(19M\sqrt{b_{-1}^2 + \frac{3L\gamma^2}{M(1-\beta_1)}S_K + \frac{6b_{-1}^2}{1-\beta_1} + \frac{3L\gamma b_{-1}}{1-\beta_1}} + 2M\right)^2$$

$$\le 762M^2\left(b_{-1}^2 + \frac{3L\gamma^2}{M(1-\beta_1)}S_K + \frac{6b_{-1}^2}{1-\beta_1} + \frac{3L\gamma b_{-1}}{1-\beta_1}\right) + 8M^2 b_{-1}^2,$$

where we use the fact that $(a + b)^2 \leq 2a^2 + 2b^2$. Rearranging the terms, we have

$$S_K^2 - \frac{6 \cdot 38^2 LM}{(1 - \beta_1)^3} S_K - \frac{2 \cdot 38^2 M^2}{\gamma^2 (1 - \beta_1)^2} \left( b_{-1}^2 + \frac{8b_{-1}^2}{762} + \frac{6b_{-1}}{1 - \beta_1} + \frac{3L\gamma b_{-1}}{1 - \beta_1} \right) \leq 0.$$

Solving the quadratic inequality and using that $\sqrt{a^2 + b^2} \leq a + b$, one can obtain

$$S_K \leq \frac{6 \cdot 38^2 LM}{(1 - \beta_1)^3} + \frac{38\sqrt{2}M}{\gamma(1 - \beta_1)} \sqrt{b_{-1}^2 + \frac{8b_{-1}^2}{762} + \frac{6b_{-1}}{1 - \beta_1} + \frac{3L\gamma b_{-1}}{1 - \beta_1}}$$

$$\leq \frac{6 \cdot 38^2 LM}{(1 - \beta_1)^3} + \frac{38\sqrt{2}M}{\gamma(1 - \beta_1)} \left( \frac{21b_{-1}}{19} + \frac{3b_{-1}}{\sqrt{1 - \beta_1}} \right),$$

because $L\gamma \leq \frac{b_{-1}}{48}$. Therefore, after division of both sides by $K$ and substitution of $\gamma$ from (76), we get the final bound for Clip-M-AdaGrad/Clip-Adam:

$$\frac{1}{K} \sum_{k=0}^{K-1} \|\nabla f(x_k)\|^2$$

$$= \mathcal{O} \left( \frac{1}{(1 - \beta_1)^{\frac{3}{2}}} \max \left\{ \frac{LM \ln\left(\frac{4}{\delta}\right)}{K^{\frac{2\alpha-1}{3\alpha-2}}}, \frac{\sqrt{LM}\sigma \ln^{\frac{\alpha-1}{\alpha}}\left(\frac{4}{\delta}\right)}{K^{\frac{2\alpha-2}{3\alpha-2}}}, \frac{\sigma^{\frac{2\alpha}{2\alpha-1}}(LM)^{\frac{\alpha-1}{2\alpha-1}} \ln^{\frac{2\alpha-2}{2\alpha-1}}\left(\frac{4}{\delta}\right)}{K^{\frac{2\alpha-2}{3\alpha-2}}} \right\} \right)$$

with probability at least $1 - \delta$. $\qquad\qquad\square$

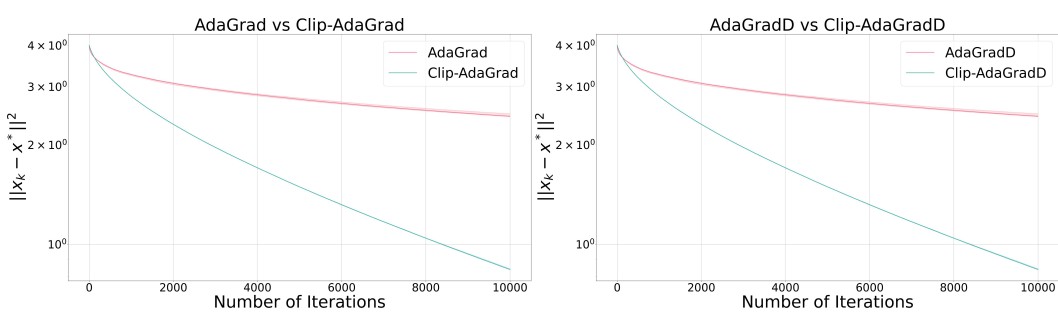

Figure 4: Performance of different versions of AdaGrad (with and without clipping/delay) with stepsize $\gamma = 1/128$ on the quadratic problem.

## D  NUMERICAL EXPERIMENTS: ADDITIONAL DETAILS AND RESULTS

### D.1  QUADRATIC PROBLEM

In addition to the results provided in the main text, we compare the performance of different versions of AdaGrad with $\gamma = 1/128$. The results are given in Figure 4. One can notice that methods with clipping consistently outperform the methods without clipping for this stepsize as well.

Moreover, we provide the results of similar experiments for Adam with and without clipping/delay in Figure 5 (for $\beta_1 = 0.9$ and $\beta_2 = 0.999$). In general, the observed results for Adam-based methods are very similar to the ones obtained for AdaGrad: clipped versions of Adam show better high-probability convergence than non-clipped ones.

### D.2  ALBERT BASE V2 FINE-TUNING

In our experiments with finetuning of the ALBERT Base v2 model on CoLa and RTE datasets, we follow a standard practice of usage Adam, we apply bias correction to Adam and Clip-Adam. For the delayed version – Clip-AdamD – we do not apply bias correction and tune $b_0$ instead.

In the main part of our work, we present the results for Clip-Adam with layer-wise clipping. In Figure 6, we provide the results in the case of coordinate-wise clipping. In general, they are quite similar to the ones given in Figure 3, indicating that both clipping strategies can be useful in practice and improve the high-probability convergence of Adam.

We also conducted experiments with Clip-AdamD and compared its performance with Clip-Adam. We tuned parameter $\epsilon$ defining $b$ as $b = \epsilon \mathbf{1}$, where $\mathbf{1} = (1, 1, \ldots, 1)^\top \in \mathbb{R}^d$. Tuning was performed in two phases: during the first phase, we selected the best values of $\epsilon$ from $\{10^{-8}, 10^{-7}, 10^{-6}, 10^{-5}, 10^{-4}, 10^{-3}, 10^{-2}\}$, and then for every selected $\hat{\epsilon}$ we tried $\epsilon \in \{0.2\hat{\epsilon}, 0.5\hat{\epsilon}, 0.8\hat{\epsilon}, 2\hat{\epsilon}, 5\hat{\epsilon}, 8\hat{\epsilon}\}$. In the case of CoLa dataset, the best $\epsilon$ was $2 \cdot 10^{-6}$, and in the case of RTE dataset, the best $\epsilon$ was $2 \cdot 10^{-6}$.

The results are presented[7] in Figure 7 and show that Clip-AdamD performs worse than Clip-Adam, especially on CoLa dataset. However, it is worth mentioning that the clipping level was selected the same for both Clip-Adam and Clip-AdamD. Moreover, we have not tried to use bias correction for Clip-AdamD that could also improve its performance. Finally, the tuning of $\epsilon$ parameter over multiple runs can also improve the result of Clip-AdamD.

Finally, we also conducted similar experiments with AdaGrad-based methods with and without clipping/delay. Parameter $\gamma$ and batchsize were tuned across the same values as in the case of Adam. Moreover, similarly to the experiments with Adam, we used standard layer-wise clipping for AdaGrad-based methods since it gave better results. The final parameters are (i) $\gamma = 10^{-4}$, batchsize 4, $\lambda = 5$ for (Clip-)AdaGrad on CoLa dataset, (ii) $\gamma = 10^{-4}$, batchsize 16, $\lambda = 1$ for (Clip-)AdaGrad on RTE dataset, (iii) $\gamma = 10^{-4}$, batchsize 4, $\lambda = 5$ for (Clip-)AdaGradD on CoLa

---

[7]In the plots, we use the name Clip-RAdamD, which is equivalent to Clip-AdamD as explained at the beginning of Appendix C.

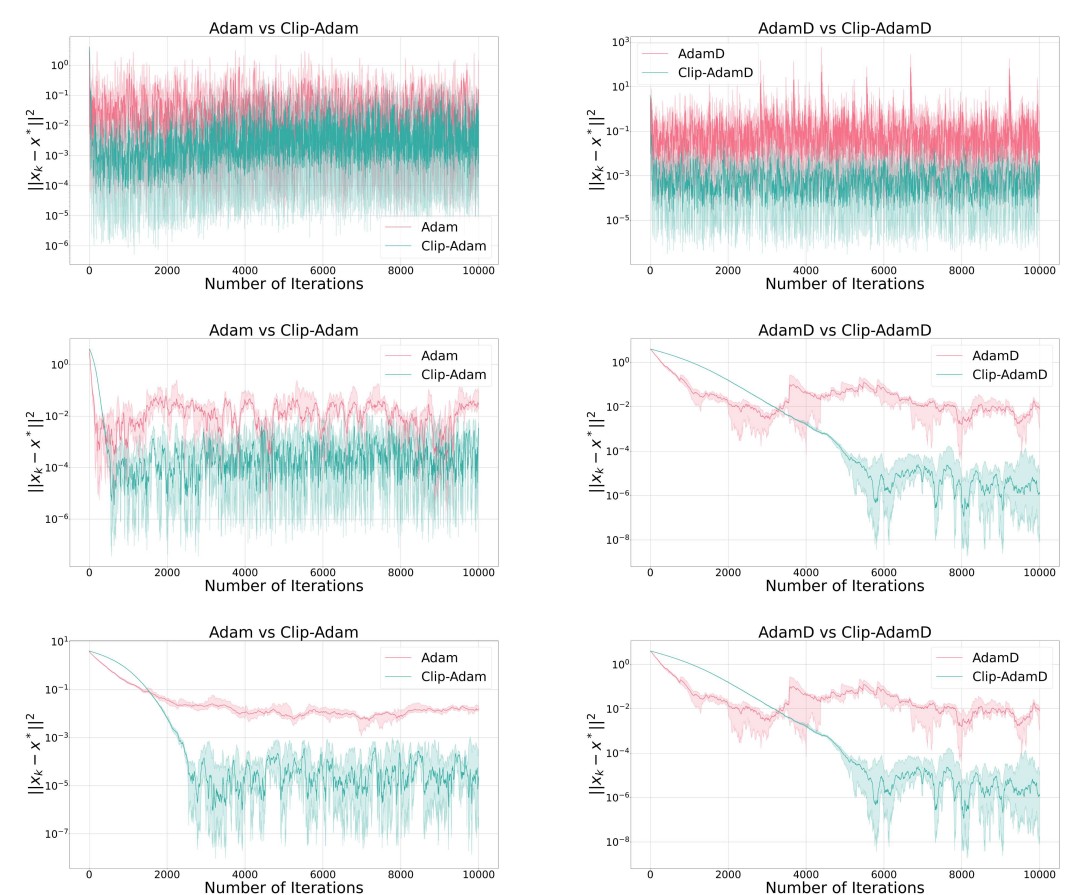

Figure 5: Performance of different versions of Adam (with and without clipping/delay) under the standard setting ($\beta_1 = 0.9$, $\beta_2 = 0.999$) with stepsizes $\gamma = 1$ (first row) and $\gamma = 1/16$ (second row) on the quadratic problem.

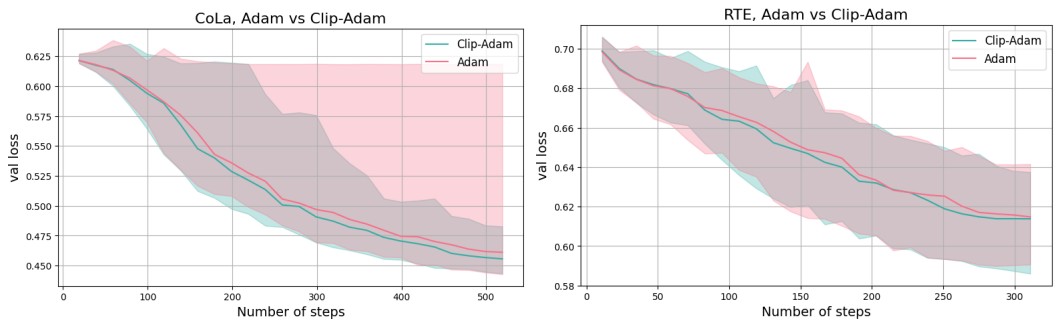

Figure 6: Validation loss for ALBERT Base v2 fine-tuning task on the CoLa and RTE datasets. Clip-Adam is used with coordinate-wise clipping ($\lambda = 0.02$ for CoLa and $\lambda = 0.005$ for RTE).

dataset, and (iv) $\gamma = 10^{-4}$, batchsize 16, $\lambda = 0.1$ for (Clip-)AdaGradD on RTE dataset. The results are presented in Figure 8. For this particular case, there is no big difference between versions of AdaGrad with and without clipping, and only for CoLa dataset we see that Clip-AdaGrad has much smaller error band than AdaGrad.

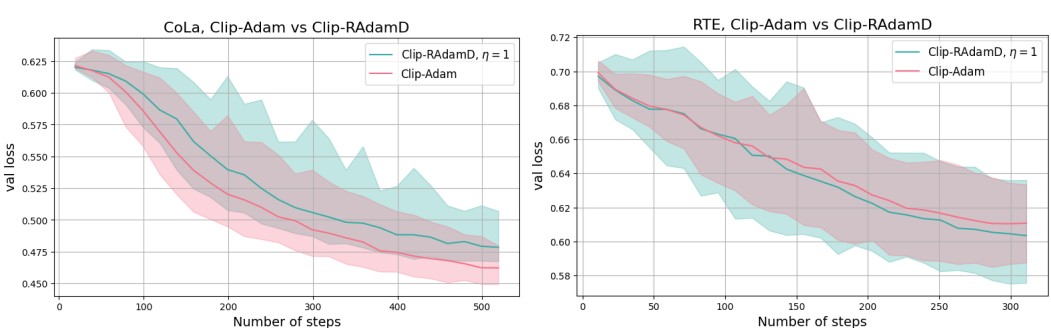

Figure 7: Validation loss for ALBERT Base v2 fine-tuning task on the CoLa and RTE datasets.

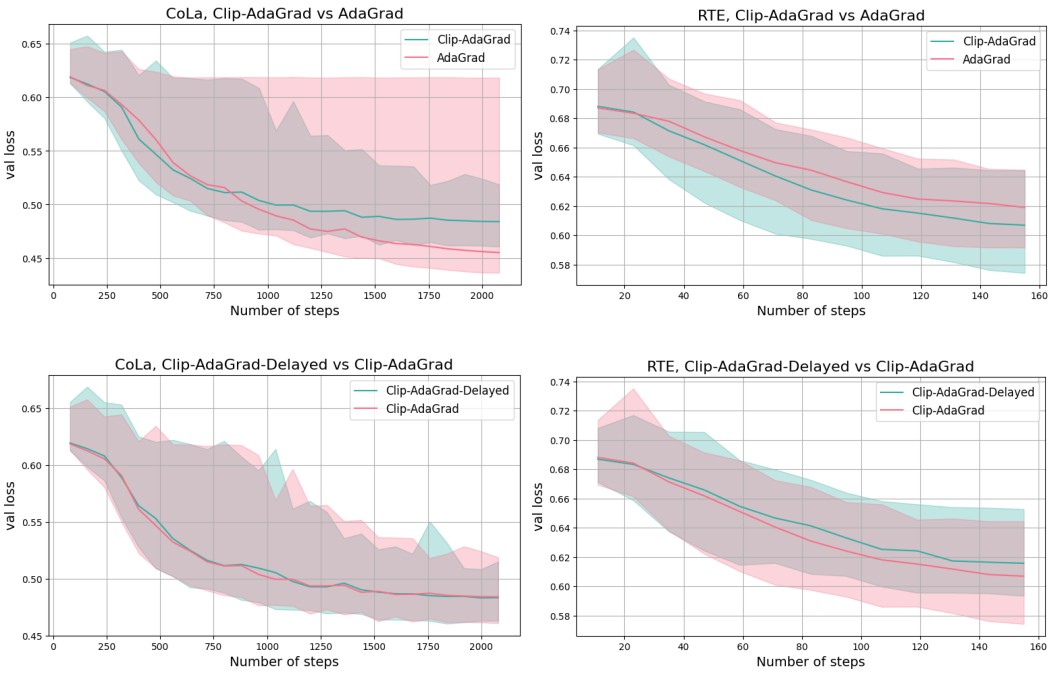

Figure 8: Validation loss for ALBERT Base v2 fine-tuning task on the CoLa and RTE datasets.

