# OpenReview forum: "Clipping Improves Adam and AdaGrad when the Noise Is Heavy-Tailed"
_ICLR.cc/2025/Conference — Submitted to ICLR 2025_

### Official Review · Reviewer_Sb6r · 2024-10-21

**Soundness:** 3
**Presentation:** 3
**Contribution:** 3
**Rating:** 6
**Confidence:** 2

**Summary:**

This paper theoretically analyzed the influence of heavy-tailed gradient noise on the convergence of AdaGrad/Adam (and their delayed version) and their clipping version. The authors found that clipping improves the convergence of AdaGrad/Adam under the heavy-tailed noise, which is validated by some experiments.

**Strengths:**

1.The authors proved that AdaGrad/Adam (and their delayed version) can have provably bad high-probability convergence if the noise is heavy-tailed.

2.They also derived new high-probability convergence bounds with polylogarithmic dependence on the confidence level for AdaGrad and Adam with clipping and with/without delay for smooth convex/non-convex stochastic optimization with heavy-tailed noise.

3.Some empirical evaluations validated their theoretical analysis.

**Weaknesses:**

1.The authors stated “Adam can be seen as Clip-SGD with momentum and iteration-dependent clipping level)”. And, the results of Adam/AdaGrad show that  their high-probability complexities don’t have polylogarithmic dependence on the confidence level in the worst case when the noise is heavy-tailed. However, they didn’t explain why the latent clipping brings the negative result (Theorem 1), which is not consistent with their rest results (Theorems 2-4).

2.Some comparisons of results are provided behind Theorems 1 and 4. These comparisons are not clear enough to emphasize the advantages of the results of this paper. Making a table to present all results of this paper and previous work may benefit readers' understanding.

**Questions:**

1.Line 13: What is the meaning of “for the later ones”? Does “the later one” denote Large Language Models, and why? Do other models not have heavy-tailed gradients？

2.Line 17: Which case do the authors want to state for the phrase “in this case”?

3.Lines 47-49: What is the meaning of the statement “Adam can be seen as Clip-SGD with momentum and iteration-dependent clipping level)”?

---

> ### Author Response · Authors · 2024-11-22
> **Response to the Reviewer Sb6r**
>
> Thank you for your thorough review and remarks. We are grateful that you emphazise the strengths of our work.
>
> **Weaknesses**
>
> >**The authors stated “Adam can be seen as Clip-SGD with momentum and iteration-dependent clipping level"**
>
> Indeed, Adam can be interpreted as some variant of Clip-SGD with momentum and time-varying clipping level. This is so because Adam has a parameter $\beta_1$, responsible for the momentum, and also a scaling factor normalizing the step direction in some way. Our results show that latent clipping (built into Adam or AdaGrad) is not sufficient, i.e., it does not allow us to deal with noise with heavy tails (to be more precise, our lower bounds in Theorems 5-8 show the absence of a logarithmic factor, which we would like to see in the bounds for the required number of iterations for convergence with high probability). This demonstrates why latent clipping in the Adam/AdaGrad methods is not enough to deal with heavy-tailed noise.
>
> >**Some comparisons of results are...**
>
> Thank you for your thoughtful comment. We will add a table with comparison of study of methods with and without clipping under different assumptions on stochasticity a little bit later.
>
> **Questions**
>
> 1) Yes, the later refers to LLMs as it was observed that stochastic gradients of these models exhibit heavy-tailness property. Another well-studied example is GANs. It is not the case that all the models have heavy tailed noise in stochastic gradients, but, in particular, for both LLMs and GANs, clipping is essential to make their training stable and our work explain why this is case from the theory point of view.
> 2) Here we refer to the case of stochasticity with heavy tails.
> 3) We suppose that we already describe the meaning of the phrase “Adam can be seen as Clip-SGD with momentum and iteration-dependent clipping level” in the **Weaknesses** section.

---

> > ### Comment · Reviewer_Sb6r · 2024-11-25
> >
> > Thanks for the responses of the authors. I don't have any questions. I will hold this score.

---

> > > ### Author Response · Authors · 2024-11-27
> > > **Response to the Reviewer Sb6r**
> > >
> > > Thank you for your response.

---

### Official Review · Reviewer_h1EE · 2024-10-25

**Soundness:** 2
**Presentation:** 2
**Contribution:** 2
**Rating:** 5
**Confidence:** 3

**Summary:**

This paper studies the convergence behavior of AdaGrad/Adam, considering heavy-tailed noise which is both significant in theoretical and empirical aspects. The authors prove AdaGrad/Adam failed in this case. To handle this issue, the authors study AdaGrad/Adam with clipping and derive a high probability convergence bound which owns a polylogarithmic dependence on the confidence level. Finally, they provide some experimental results.

**Strengths:**

This paper studies the convergence behavior of AdaGrad/Adam when the noise is heavy-tailed, both of which are quite important in the deep learning field. They find the convergence issue of both two algorithms, specifically the polynomial dependence of the confidence level inside the convergence bound. To solve this issue, they consider AdaGrad/Adam with clipping and show a convergence bound with polylogarithm dependence of the confidence level. Finally, some experimental results are provided, showing the superiority of adding clipping over the non-clipping versions.

**Weaknesses:**

I have the following major concerns.

**On negative results**

The main motivation in this paper comes from the potential failure of AdaGrad/Adam in the heavy-tailed noise case.  However, the main result to prove the failure, Theorem 1, is not convincing. First, the result shows a complexity of Adam/AdaGrad that has inverse-power dependence on $\delta$. However, this bound should require $\beta_2 = 1-1/T$ and $\\|x\_0-x^*\\| \ge \gamma L$ instead of arbitrary $\beta_2$ and $x\_0$. It's then questionable whether another setup of $\beta_2$ and $x\_0$ may achieve success. Second, I think it's not convincing to say Adam/AdaGrad is a failure given the inverse-power dependence on $\delta$ inside the convergence bound. Note that the dominated order in a convergence bound (or complexity) comes from the order of $T$ (or the accuracy $\epsilon$). I see that the complexity still achieves $\Omega(poly(\epsilon^{-1/2}))$ which leads to the convergence.

I suggest the author prove a negative result similar to [Remark 1,1], where they can show that for arbitrary step size and initialization, SGD has a non-convergence issue on a specific problem.

**On results regarding clipping**

First, the author claims that the main goal of incorporating the clipping is to improve the dependence of $\delta$ to polylogarithm order. However, I do not see clearly any polylogarithm order of $\delta$ in Theorem 2, 3, and 4, particularly in the complexity formulas. Second, I do not see the motivation for using a delayed step size. If we have the AdaGrad/Adam with clipping, why do we still need the delayed step-size version? Finally, the polylogarithm order of $\delta$ for AdaGrad with clipping has already been obtained in [2], although with a slightly stronger assumption. I suggest the author claim more on the proof difference with their results.

**Reference**

[1]. Zhang J, Karimireddy S P, Veit A, et al. Why are adaptive methods good for attention models? Advances in Neural Information Processing Systems, 2020, 33: 15383-15393.

[2]. Li S, Liu Y. High Probability Analysis for Non-Convex Stochastic Optimization with Clipping. ECAI 2023. IOS Press, 2023: 1406-1413.

**Questions:**

Please refer to **Weaknesses**.

---

> ### Author Response · Authors · 2024-11-22
> **Response to the Reviewer h1EE**
>
> We are grateful for your thorough review and comments. Thank you for acknowledging strengths of our paper.
>
> **Weaknesses**
>
> >**On negative results**
>
> We kindly disagree with this point. Indeed, the choice of $\beta_2$ as $1 - \frac{1}{T}$ is natural for the following reason - the divergence of the Adam method has already been shown in [3]. That is why, in order to show the advantage of Adam/AdaGrad with clipping over non-clipped versions, we need to consider exactly the same parameter $\beta_2$ as in the study of theoretical guarantees of convergence of Adam/AdaGrad with clipping (see Theorems 2-4). As for the constraint on the initial distance, it seems to us to be intuitive - it indicates that our starting point is not very close to the solution.
>
> As for $\Omega(\textit{poly}(\varepsilon^{-\frac{1}{2}}))$, we agree with the reviewer - our lower bounds suggest that the method converges. But this is what we need. Theorem 1 (see proof in Appendix, Theorems 5-8) says that the number of iterations required for high-probability convergence necessarily depends on $\frac{1}{\delta}$ in some power, that is, there is no logarithmic factor of $\frac{1}{\delta}$. This is enough to demonstrate the advantage of clipped Adam/AdaGrad over unclipped methods.
>
> If we discuss [1], Remark 1 demonstrates that SGD with heavy-tailed noise can diverge. But it does not say that it will always diverge. To understand this fact it is enough to see that the divergence in [1] is shown in the paradigm of convergence on mathematical expectation, i.e. SGD diverges on mathematical expectation. But in our work we consider convergence with high probability. And, to repeat, to demonstrate the outperformance of clipped Adam/AdaGrad over unclipped ones, the consideration of Theorems 5-8 (combined into Theorem 1 in the main part) is sufficient (see the explanation above).
>
> >**On results regarding clipping**
>
> Thank you for this important clarification. The logarithmic factor is written in the parameter $A$ for Theorems 2, 3 and 4 for convenience, since from our point of view they are rather large. The proof of each of these theorems has a convergence bound that depends on $\gamma$ (see Appendix). After substituting $\gamma$ as given in theorems’ formulations, we can get a bound for the required number of iterations in terms of $\tilde{\mathcal{O}}$, where the logarithmic factor is hidden.
>
> If we discuss the use of delayed stepizes, it is worth noting that this technique is not new. First, it is worth noting [3], where the instability of Adam behavior was studied. To combat this instability, the AMSGrad method was proposed, which is based on the idea of delayed stepsizes: it is enough to refer to the algorithm and look at the $\hat{v}_t$ update. The only thing worth clarifying is that the delayed stepsize has a slightly different structure. Moreover, delayed stepsizes can be mentioned in studies of distributed systems and parallelization, but that is beyond the scope of our study.
>
> Finally, let us discuss the result from [2]. As it has already been noticed, in [2] a similar result was obtained, except for a small difference. In fact, in this difference lies the whole difference of the considered problems. Indeed, let us turn to [2]. The convergence of AdaGrad with clipping is given in Theorem 13, which states that assumptions are made on the $L$-smoothness of the empirical risk, its uniform boundedness and boundedness of $\alpha$-th moment. Therefore, in the worst case, these assumptions imply the boundedness of $\nabla f_{\xi}(x)$ , meaning that the noise is bounded and, thus, sub-Gaussian. At the same time, AdaGrad analysis is already available for such noise (see [4]), and without clipping. That is, in [2] clipping for AdaGrad is applied unreasonably, since the polylogarithmic factor can be achieved without it under the assumptions made by the authors of [2]. Furthermore, we already discussed the result from [2] in our paper (see **Discussion of the results** after Theorem 4).
>
> **References**
>
> [1] Zhang J, Karimireddy S P, Veit A, et al. Why are adaptive methods good for attention models?
>
> [2] Li S, Liu Y. High Probability Analysis for Non-Convex Stochastic Optimization with Clipping.
>
> [3] Sashank J. Reddi, Satyen Kale & Sanjiv Kumar. On The Convergence of Adam and Beyond.
>
> [4] Zijian Liu. High Probability Convergence of Stochastic Gradient Methods.

---

> > ### Comment · Reviewer_h1EE · 2024-11-23
> >
> > Dear Authors,
> >
> > As far as I can understand, (if not please clarify for me), the failure of an algorithm in the paper means that one could find a parameter setup where the algorithm could have a polynomial dependence over $\delta$.
> >
> > First, I agree that $\beta_2 = 1-1/T$ is a commonly used setup in literature. However, could Theorem 1 answer the following question:
> > - If I use $\beta_2 = 0.999$ or $\beta_2 = 1-1/t$ or other commonly used setup, could Adam still have a polynomial dependence over $\delta$?
> >
> > I hope to see some results showing that, for a broader range of $\beta_1,\beta_2$, such as $\beta_1 < \sqrt{\beta_2}$ (the setup used in the divergence result in [Reddi et al., 2019]), Adam could diverge/or have bad dependency over $\delta$ under the heavy-tail noise. Otherwise, Theorem 1 is a bit limited.
> >
> > Second, I am not sure whether improving the dependency over $\delta$ may be a very important point in the convergence bound. The dominated order in the convergence bound is determined by $T$ whereas $\delta$ is secondary. For example, taking $\delta = 0.01$, meaning that the probability is at least $0.99$, the difference between $\delta^{-1/2}$ in Theorem 1 and $\text{poly}\log(1/\delta)$ could be ignored given a sufficiently large $T$.

---

> > > ### Author Response · Authors · 2024-11-23
> > > **Responce to the official comment of Reviewer h1EE**
> > >
> > > Dear Reviewer,
> > >
> > > Actually, not really. If we understand the question correctly, our negative result for Adam and Adam with delay says that if we want to converge with high probability (i.e., with probability $1 - \delta$), then when choosing $\beta_2$ as $1 - \frac{1}{T}$, we still have to do some number of iterations that depends on the factor $\frac{1}{\delta}$, and without logarithm. Therefore, it would be more accurate to say that the failure of the algorithm is not in the selection of parameters, but rather in the selection of objective function and stochasticity, for which we show that for $\beta_2 = 1 - \frac{1}{T}$ convergence requires a polynomial dependence on $\frac{1}{\delta}$. The parameters probably do not need any explanation, since the various dependencies and relationships of the parameters were derived from the minimized function and noise.
> > >
> > > Let us discuss the various options for $\beta_2$. Unfortunately, those typical examples you give are drastically different. Thus,
> > >
> > > $\bullet$ $\beta_2 = 0.999.$ If we represent an iterative process ($t = 0, 1, \ldots, T$), it is a constant both in terms of the total number of iterations and in terms of the current iteration. For this kind of constants there is already a result [Reddi et al., 2019], which describes exactly that there is no convergence in principle. That is, whatever constant $\beta_2$ we take (except $\beta_2 = 1$, since in that case we essentially have a heavy-ball method), there is always exists $\beta_1$ that there is no convergence, $\textit{even in the deterministic case}$, so we see no point in adding this result to our work (since it completely replicates Reddi's result).
> > >
> > > $\bullet$ $\beta_2(t) = 1 - \frac{1}{t}.$ In such a case, it will be possible to fully apply the idea of our theorems for Adam and Adam with delay. If we look carefully at the format of proofs of negative results, there are lemmas for each outcome, among which there are lemmas responsible for the analytic point Adam and Adam with delay for the deterministic case. It is enough just to substitute $\beta_2(t) = 1 - \frac{1}{t}$ instead of $\beta_2 = 1 - \frac{1}{T}$. In fact, the result will turn out to be similar. From our point of view it is illogical to add it to the paper, since Adam convergence with clipping is shown for $\beta_2 = 1 - \frac{1}{T}$, and so the negative result was constructed for such $\beta_2$.
> > >
> > > Finally, let's discuss the $\delta$ dependency improvement. We agree with the statement that if we take the usual $\delta$ (as you said, we can choose $\delta = 0.01$), there is almost no difference. The value of the probabilistic results is that we can estimate how challenging it is for us to reduce $\delta$. Since the logarithm grows slower than the polynomial, from a theoretical point of view it is much better to get $\log\left(\frac{1}{\delta}\right)$ in the estimation than a polynomial dependence. As an example, consider $\delta = 10^{-8}$, for this option the difference is already visible. Moreover, everyone tends to obtain bounds with the logarithm, since this is generally accepted in the literature.
> > >
> > > We would be happy to answer any questions you have.

---

> ### Comment · Reviewer_h1EE · 2024-11-25
>
> Dear Authors,
>
> Sorry for the delayed reply. I have read carefully of your rebuttal and I agree that the negative result is meaningful as $\beta_2 = 1-1/T$ commonly appears. I think that the concern comes from whether other setups of $\beta_2$, such as $1-1/t$ or $1-1/\sqrt{t}$ or any general setup that closes to one may still lead to a polynomial order of $\delta$. That's the reason why I think that Theorem 1 is limited. You comment that $1-1/t$ could be derived easily. I hope that this may be written down clearly in the new version if possible.
>
> Second, I hope to see a clear motivation for adding the delayed step-size as I do not see it clearly from the rebuttal. I do not agree that it's a very common mechanism.
>
> Third, I want to remind you that there are some recent works studying the high probability of AdaGrad and Adam with logarithm order of $\delta$, see e.g., [1,2].
>
> However, I recognize the value of this paper and thank you for the authors' detailed response. I will raise my score to 5.
>
> References.
>
> [1] Kavis A, Levy K Y, Cevher V. High probability bounds for a class of nonconvex algorithms with adagrad stepsize. ICLR, 2022.
>
> [2] Yusu Hong and Junhong Lin. Revisiting Convergence of AdaGrad with Relaxed Assumptions. UAI 2024.

---

> > ### Author Response · Authors · 2024-11-27
> >
> > We thank the reviewer for their reply and for participating in the active discussion with us.
> >
> > **Negative results for Adam/AdamD with $\beta_2(t) = 1 - 1/t$.** We are working on the proof and will share it as soon as possible. Preliminary derivations show inverse-power dependence on $\delta$ (though with a slightly worse exponent).
> >
> > **Motivation for the delayed stepsizes.** The main motivation for the usage of the delayed stepsizes is to obtain the convergence bounds under weaker assumptions. From the technical point of view, delayed stepsizes (at iteration $k$) are easier to analyze because they are conditionally independent of the stochastic gradient (at iteration $k$). We also note that we provide the analysis of Clip-Adagrad/Adam without the delay (Theorem 4) but this result relies on additional Assumption 4.
> >
> > **References to [1, 2].** We thank the reviewer for the references. Indeed, these papers are very relevant to our work, and they provide the results for Adagrad with logarithmic dependence on $\delta$ -- we will include the discussion of these results to the final version of our paper. However, these results are derived under additional assumptions. In particular, Kavis et al. [1] assume that the stochastic gradients are bounded almost surely, i.e., the noise and the gradient are bounded. Since the bounded noise has a sub-Gaussian distribution, their results do not cover the case of the heavy-tailed noise as our Theorem 4 does. Next, Hong & Lin [2] consider the case of relaxed almost surely affine noise, which is allowed to grow with $f(x) - f^\ast$ and $\|\| \nabla f(x) \|\|$ but has to be bounded for any fixed $x$. Therefore, the noise considered in [2] is also sub-Gaussian with sub-Gaussian variance dependent on $x$. This setting is not directly comparable to the setup we consider: in contrast to Assumption 1 from our paper, the noise considered from [2] can explicitly depend on $x$; however, the noise considered in our paper can be unbounded and have infinite variance even for fixed $x$. Therefore, in [2], the authors can use the concentration properties of sub-Gaussian random variables in the proof, but they cannot be applied in the setup we consider. Due to this reason, we use clipping and Bernstein's inequality.
> >
> > ---
> >
> > References
> >
> > [1] Kavis A, Levy K Y, Cevher V. High probability bounds for a class of nonconvex algorithms with adagrad stepsize. ICLR, 2022.
> >
> > [2] Yusu Hong and Junhong Lin. Revisiting Convergence of AdaGrad with Relaxed Assumptions. UAI 2024.

---

### Official Review · Reviewer_QDsp · 2024-11-06

**Soundness:** 3
**Presentation:** 2
**Contribution:** 2
**Rating:** 5
**Confidence:** 4

**Summary:**

This paper examines the high-probability convergence of adaptive optimizers like AdaGrad and Adam under heavy-tailed noise. Without gradient clipping, these methods can struggle with convergence. The authors show that gradient clipping significantly improves convergence bounds and empirical performance for AdaGrad and Adam, making them more robust to heavy-tailed noise.

**Strengths:**

The negative example for Adagrad's (actually, Adagrad-Norm) convergence is interesting. This could imply that heavy-tail noise is not handled by "adaptive" methods, which clarifies a misconception in the area. If this point is fully justified (given that the concerns I have below are resolved), I think it is a interesting contribution.

**Weaknesses:**

1. **Not analyzing Adam, but perhaps a twin of Adagrad**:  This paper does not analyze the original Adam, but Adam with beta2 = 1-1/K. The paper wrote "Therefore, the standard choice of beta2, in theory is, = 1 - 1/K where K is the total number of steps", but this is not the standard choice in theory. For instance, there are recent results proving convergece of Adam for constant beta2 (Zhang et al.'2022, cited in the submitted work). The analyzed algorithm with 1-1/K might be essentially Adagrad (note that for beta2 = 1/1/k, the algorithm becomes Adagrad, but for beta2 = 1 -1/K, it requires more discussion). The convergence properties of Adam and Aadgrad are quite different.

2. **Analyzed scalar-coefficient clipped version**, instead of regular clipping: The clipped version Algorithm 2 uses a scalar b_t, instead of a vector b_t in the typical adaptive gradient methods. This is because the update of b_t uses the norm of the clipped gradient instead of the gradient vector. This makes the algorithm quite different from the original adaptive gradient methods.
       The authors renamed the algortithm from '"Adagrad-Norm" to "Adagrad", and uses Adagrad-CW to describe the orginal version, as mentioned in a footnote. But this naming is quite misleading. If the paper analyzed Adagrad-norm, then the title and abstract should reflect it.
        Another example that renaming Adagrad-norm by Adam is misleading: for the experiments, I cannot tell for sure whether the authors use the original Adam or Adgrad-norm. My guess is the authors used Adagrad-norm for experiments, since the term "Adam" is already renamed.

3. **Contribution.**  Given the above modifications, the paper actually shows that Adagrad-norm-with-clipping works well while Adagrad-norm-without-clipping works not so well, for the heavy-tail-noise case. Thus the result is not about the orginal Adam. Nevertheless, there is still some chance that such an analysis could shed some light on the relation of clipping and Adam, if the experiments on Adam exhibit similar behavior to Adagrad-norm. However, the experiments are on "Adam", which, I guess, actually means Adagrad-norm in the context of this paper, thus the experiments may not be relevant to practitioners.

**Questions:**

In the experiments, does "Adam" mean the version of this paper, or the common version in the literature (i.e. the original version by Kingma and Ba)?

---

> ### Author Response · Authors · 2024-11-22
> **Response to the Reviewer QDsp**
>
> We are grateful for your review and comments. Thank you for emphasizing the strengths of our work.
>
> **Weaknesses**
>
> >**Not analyzing Adam, but a twin of Adagrad:**
>
> We kindly disagree with the provided statetement. To the best of our knowledge, $\beta_2$ close to $1$ is a common choice for analysis of Adam, e.g., see [1].  Furthermore, looking at the results of [2], it is clear that the constraint on $\beta_2$ is $\beta_2 \geq 1 - \mathcal{O}(\frac{1}{n^2})$, where $n$ is the size of the entire dataset (see Theorem 3.1). Even if we choose $\beta_2$ for $n = 1000$, this parameter does not correlate with what is used in practice. And, to be more precise, in most cases $\beta_2 = 1 - 1/K$ is smaller than $\beta_2$ from [2]. This means that $\beta_2$ cannot be called a constant. Therefore, we kindly disagree that our Adam analysis would be weaker when compared to prior work.
>
> >**Analyzed scalar-coefficient clipped version**
>
> We kindly disagree with the point that scalar-coefficient and vector-coefficient versions are quite different. Indeed, we can look at [3], where, for example, AdaGrad and AdaGrad-norm under Subgaussian noise are investigated. The ideas of the proofs are the same, except for the usage of norm for AdaGrad-norm and coordinate notation for AdaGrad. Nevertheless, we agree that it's worth renaming AdaGrad to AdaGrad-norm (Adam too) to avoid misunderstanding.
>
> >**Contribution**
>
> We appreciate the reviewer's feedback. Our work focuses on analyzing Adagrad-Norm and Adam-norm with clipping and demonstrates its effectiveness under heavy-tailed noise, which is a known practical challenge in optimization. While this differs from the original Adam, our analysis highlights critical insights into the role of clipping in adaptive methods. As noted earlier, the scalar-coefficient and vector-coefficient approaches share similar proof techniques, and our findings remain relevant to practitioners aiming to improve optimizer stability. We will revise the terminology in the manuscript to ensure clarity and alignment with our analysis.
>
> **References**
>
> [1] Manzil Zaheer. Adaptive Methods for Nonconvex Optimization
>
> [2] Yushun Zhang. Adam Can Converge Without Any Modification On Update Rules
>
> [3] Zijian Liu. High Probability Convergence of Stochastic Gradient Methods

---

> ### Comment · Reviewer_QDsp · 2024-12-02
> **disagree with the response**
>
> I've read the authors' response, but it is not convincing.
>
> 1) "To be more precise, in most cases, 1 - 1/K is smaller than $\beta_2$ from [2]. This means that cannot be called a constant":
>
> This comment does point out a gap between theory and practice, but the conclusion is not acceptable. For finite-sum problem where n is fixed, any parameter that only depends on n is a constant. In contrast, K is a changing parameter, which is more like a diminishing stepsize in classical analysis of gradient descent or stochastic gradient descent. One could say there is a gap between the constant in [2] and the practically used constant (a common thing in theory), and one can even say in a changing-sample-size problem 1-g(n) is not a constant, but one cannot say it is not a constant in the setting of fixed-n finite-sum problem.
>
> There are a few more examples to illustrate the point on "constant" v.s. "non-constant" in optimization.
> For instance, there are many optimization algorithms (including but not limited to affine scaling method, a few distributed optimization methods, etc.) whose convergence for constant stepsize are proved for small stepsize, while in practice people used large stepsize. **In this example, one could say there is a gap between the constant in theory and the constant in practice, but one cannot say the theory for these methods do not prove the result for a constant.**
>
> Another example is SGD's constant stepsize v.s. diminshing stepsize: it is well known that diminishing stepsize like 1/sqrt{K} is needed for SGD for finite-sum optimization to converge, and some researchers argued that 1/sqrt{K} is larger than the stepsize used for some practial problems. Howver, it is considered a good contribution when researchers proved that SGD with constant stepsize can converge under certain conditions -- people do not need to consider whether the constant of the proof is smaller than 1/sqrt{K} or not, as the first step towards convegence of constant-stepsize SGD. The precise charaterization of the constant is left to future work. **Again, in this example, one could say there is a gap between the constant in theory and the constant in practice, but one cannot say the theory for these methods do not prove the result for a constant.**
>
> 2)
> "We kindly disagree with the point that scalar-coefficient and vector-coefficient versions are quite different; we can look at [3], where, for example, AdaGrad and AdaGrad-norm under Subgaussian noise are investigated. The ideas of the proofs are the same, except for the usage of norm for AdaGrad-norm and coordinate notation for AdaGrad."
>
> This response is NOT convincing. The paper [3] is just analyzing one specialized setting (under a certain set of assumptions), and in this setting, for "scalar-coefficient and vector-coefficient", the ideas are the same do not mean in general the two settings are the same.
> I think the naming of analyzed algorithm should be more rigorous, and not be renamed just based on the authors' understanding of proof techniques.

---

> ### Comment · Reviewer_QDsp · 2024-12-02
> **Further clarification on the original comment & discussions on asymptotic convergence**
>
> After careful thought, I think the authors might not get the essence of my original comment and the response has led to a less relevant path. I'd like to clarify a bit more, since this may be useful for the community.
>
> My original comment says: "The analyzed algorithm with 1-1/K is essentially Adagrad."
> Let me clarify a bit. Consider three setttings of beta2 for Adam.
>
> **Setting A1 (diminshing 1 - beta2):**
>     Adam with beta2 = 1-1/k where k is the iteration index. It can be viewed as Adagrad, as this is the increasing-beta2 setting. The correspondence to Adagrad is not hard to show.
>
> **Setting A2 (MaxIter-dependent constant beta2):**
>     The analyzed algorithm with beta2 = 1-1/K where K is the pre-fixed iteration index is Adagrad:
>      This is a fixed-beta2 setting, but the algorithm will stop at K iterations.
>
> **Setting A3 (constant beta2 convergence):** [3]
>      Adam with large enough constant beta2 converges to stationary points, under strong growth condition.
>
> The three settings remind me of the settings in SGD (for convex case, for simplicity):
>
> **Setting S1 (diminshing stepsize convergence)**:  SGD with diminishing stepsize eta = 1/sqrt{k} converges to an error 0.
> The two settings are somewhat "equivalent" to each other.
>
> **Setting S2 (MaxIter-dependent stepsize)**: SGD with constant stepsize eta = 1/sqrt{K} converges to an error g(K);
>
> **Setting S3 (constant stepsize convergence)**: SGD with constant stepsize converges to error 0 under strong growth condition.
>
> I know that some researchers like to quote S1 for SGD, and some like to quote S2 for SGD.
> I'd like to explain a bit my interpretations of the three settings of Adam, based on the analogy with the three settings of SGD:
> 1) In SGD, S2, the convergence to a non-zero error for MaxIter-dependent eta = 1/sqrt{K}, can be somehow "transferred to" S1, the asymptotic convergence of SGD with diminishing eta = 1/sqrt{k}.
> 2) In Adam, it might be the case that A2, the convergence to a non-zero error for MaxIter-dependent beta2 = 1 - 1/K, can be "transferred to" A1, the asymptotic convergence of Adam with beta2 = 1 - 1/k.
>      This convergence is less surprising since people already know Adagrad converges.
> 3) S3, the convegence of Adam with beta2 = iteration-independent constant, is more like the result of S3 for SGD: it is a fundamentally different asymptotic convergence result from S1&S2.
>
> Due to the impression of "equivalence" of S1 and S2, I thought Setting A1 and A2 are somewhat "transferrable", and claimed Setting A1 (in this paper) is essentially Adagrad (via the bridge of A2).
>
> That being said, I did not spend time to show A1 and A2 would be "equivalent". I still think it is possible (based on reading the proof of Adam with 1-1/K, it does not seem as tricky as the one in [3]), but since I did not show the equivalence, I made the following modifications to the original review:
>   i) the sentence "The analyzed algorithm with 1-1/K is essentially Adagrad" in the original review is changed to
> "The analyzed algorithm with 1-1/K might just be Adagrad (note that for beta2 = 1/1/k, the algorithm becomes Adagrad, but for beta2 = 1 -1/K, it requires more discussion)";
>  ii) "a variant of Adagrad" changed to "perhaps a variant of Adagrad".
>
> I hope someone (maybe the authors, or other follow-up researchers) can clarify the relation between A1 and A2 for Adam.
> I'd be happy to see both positive or negative relations.
>
> **Additional point: **
> 1) Even putting aside the discussion on beta2, the authors did not make a convincing argument of NOT changing the name to Adam-Norm, as mentioned in the previous comment.
>
> 2) The authors said in response "We will revise the terminology in the manuscript to ensure clarity and alignment with our analysis."
> ---I checked the revised version (the authors have chance to modify PDF), the name Adam has not been changed to Adam-norm.

---

> > ### Author Response · Authors · 2024-12-03
> >
> > We thank the reviewer for the detailed responses. Below, we provide several further clarifications.
> >
> > **On the comparison with [1].** We kindly note here that the setup of our paper is significantly different from the setup considered in [1]. More precisely, we consider the general expectation minimization problem, while in [1], the authors focus on finite sums only. In our case, $n = \infty$ (informally speaking), i.e., the results from [1] are not applicable. We also note that even in the finite-sum regime $\beta \sim 1 - \frac{1}{n^2}$ considered in [1] is larger than $\beta_2 = 1 - \frac{1}{K}$ considered in our paper whenever $K < n^2$, which is typically the case in real-world datasets that might have $n \sim 10^6$ samples and much more. Last but not least, the result from [1] implies the convergence to $\mathcal{O}(\sqrt{D_0})$ neighborhood only. This neighborhood cannot be reduced by the choice of the parameters of the method (e.g., stepsize), meaning that the result from [1] does not imply convergence to any predefined optimization error unless the problem satisfies so-called strong growth condition ($D_0 = 0$).
> >
> > We promise to add a detailed comparison to the final version of the paper. As one can see from our explanations above, the results from [1] do not undermine our contribution since the results from [1] are shown for a different problem under different assumptions.
> >
> > **On the norm- and coordinate-wise versions of the methods.** We can add new proofs for the versions of AdaGrad and Adam with coordinate-wise stepsizes -- the proofs are almost identical to the ones we have in the paper. We are not aware of any paper on AdaGrad/Adam where the proof for norm-versions and coordinate-wise versions differ significantly. We also promise to indicate these differences in the names of the methods.
> >
> >
> > **On the different choices of $\beta_2$.** We thank the reviewer for sharing this analogy. Indeed, the cases of $\beta_2(k) = 1 - \frac{1}{k}$ and $\beta_2 = 1 - \frac{1}{K}$ seem to be close but we are not aware of the technique showing an equivalence between two regimes. Nevertheless, this question is orthogonal to the main focus of our paper -- high-probability convergence of AdaGrad/Adam-based methods under the heavy-tailed noise.
> >
> >
> > ---
> > References
> >
> > [1] Yushun Zhang. Adam Can Converge Without Any Modification On Update Rules

---

### Official Review · Reviewer_k2is · 2024-11-08

**Soundness:** 3
**Presentation:** 2
**Contribution:** 3
**Rating:** 5
**Confidence:** 2

**Summary:**

The authors provide examples to show that the high-probability complexities of Adam/AdaGrad (with momentum) and their delayed versions don’t exist poly logarithmic dependence on the confidence level generally when the gradient noise is heavy-tailed. The authors show that the high-probability complexities of Clip-Adam/AdaGrad and their delayed versions have polylogarithmic dependence on the confidence level under smooth convex and smooth nonconvex assumptions. The authors conducted numerical experiments for synthetic and real-world problems.

**Strengths:**

1. The authors provide high probability convergence complexity instead of the conventional in-expectation convergence complexity that most previous literature has focused on and such high probability convergence bounds can more accurately reflect the methods’ behavior than in-expectation ones.

2. The author emphasizes the importance of gradient clipping for adaptive algorithms (Adam \ AdaGrad) to deal with heavy- tailed noise through strict high probability convergence complexity analysis.

**Weaknesses:**

1. The author's statement on the probability convergence results corresponding to different methods is not clear enough, even though these results are similar.
2. The main theoretical results of this paper are based on the assumption of local smoothness of the optimization objective, even in convex cases, which is too strong.

**Questions:**

1. In the introduction section, you cited some viewpoints from previous literatures to illustrate that Adam and Clip-SGD have similar clipping effects for stochastic gradients. so, "it is natural to conjecture that clipping is not needed in Adam/AdaGrad" . Your theorem 1 emphasizes that Adam/AdaGrad without clipping do not have a high probability convergence complexity with polylogarithmic dependence on \delta even when the variance is bounded, rather than the divergence of Adam when the noise is heavy-tailed?

2. In the discussion section of Theorem 1, you stated that “We also conjecture that for \alpha<2 one can show even worse dependence on ε and δ forAdam/AdaGrad…”. Have similar conjectures been mentioned in previous literatures, or can an informal analysis be provided?

---

> ### Author Response · Authors · 2024-11-22
> **Response to the Reviewer k2is**
>
> We are grateful for your thorough review and comments. Thank you for your remarks about strengths of our paper.
>
> **Weaknesses**
>
> >**The author's statement on the probability convergence results corresponding to different methods is not clear enough, even though these results are similar.**
>
> Thank you for this comment. If we understood the question correctly, we are talking about Theorem 1. Due to space constraints, we decided to present our results in the main part as a unified theorem (Theorem 1) that combines Theorems 5-8. If you are talking about Theorems 2-4, please clarify what is unclear.
>
> >**The main theoretical results of this paper are based on the assumption of local smoothness of the optimization objective, even in convex cases, which is too strong.**
>
> We graciously disagree with this point. Indeed, one almost always makes the assumption of $L$-smoothness of the target function to provide convergence guarantees, e.g., see  Nemirovski et al. (2009) [2.11], Ghadimi & Lan (2012)[1.2], and  Li & Orabona (2020)[A] ,for, which is, in general, a stronger assumption than ours, because we only assume $L$-smoothness only in some set $Q$ instead of $\mathbb{R}^d$. However, we realize that you are most likely referring to assumptions such as $(L_0, L_1)$-smoothness, e.g., Zhang et al., 2019: https://arxiv.org/abs/1905.1188. We provide discussion about smoothness assumption on lines 133-136. However, please note that the main idea of our work is not to extend the class of problems under consideration in terms of smoothness, but to understand the behavior of adaptive state-of-the-art methods in the case of stochasticity with heavy tails.
>
> **Questions**
>
> >**In the introduction section, you cited some viewpoints from previous literature to illustrate that Adam and Clip-SGD have similar clipping effects for stochastic gradients. so, "it is natural to conjecture that clipping is not needed in Adam/AdaGrad" . Your theorem 1 emphasizes that Adam/AdaGrad without clipping do not have a high probability convergence complexity with polylogarithmic dependence on \delta even when the variance is bounded, rather than the divergence of Adam when the noise is heavy-tailed?**
>
> If we understood the question correctly, the answer to your question is yes. Theorem 1 shows that if Adam/AdaGrad methods converge with high probability, then there will be a factor of $\frac{1}{\delta}$ instead of $\log\left(\frac{1}{\delta}\right)$ in the bound on the number of iterations required for convergence. And in fact, this is enough to say that the clipped versions of Adam/AdaGrad do handle heavy-tailed noise in terms of convergence with high probability (since the $\log\left(\frac{1}{\delta}\right)$ factor occurs for the clipped versions) compared to Adam/AdaGrad without clipping.
>
> >**In the discussion section of Theorem 1, you stated that “We also conjecture that for $\alpha<2$ one can show even worse dependence on ε and δ forAdam/AdaGrad…”. Have similar conjectures been mentioned in previous literatures, or can an informal analysis be provided?**
>
> Similar analysis arose, for example, in [1], except only that convergence is demonstrated by mathematical expectation. Moreover, to obtain similar results as in Theorems 5-8 for $\alpha < 2$, one can most likely use the same proof schemes but with modified noise - for example, unbounded discrete random variable with existing $\alpha$-moment. But, it is worth noting that this is **not necessary** if even for $\alpha =2$ there is, as we show, already a negative result.
>
> **References**
>
> [1]  Zhang J, Karimireddy S P, Veit A, et al. Why are adaptive methods good for attention models?

---

### Author Response · Authors · 2024-11-22
**General comment for all Reviewers**

We thank the reviewers for their feedback and time. We appreciate that the reviewers acknowledged the multiple strengths of our work. To be more precise,
1) Reviewers QDsp, h1EE and Sb6r emphasize the importance of the negative result for convergence of AdaGrad/Adam with high probability.
2) All Reviewers note the novelty and importance of the high-probability analysis for AdaGrad/Adam with clipping
3) Reviewers h1EE and Sb6r indicated the presence of numerical experiments, which demonstrate the validation of theoretical results.

We will be happy to answer any questions that reviewers have. Also, in case there are no more questions left, we would be grateful if you would reconsider the scores according to our answers.

Moreover, a little later, we will add a file of the modified work (e.g., Reviewer Sb6r indicated that it would be better to add a table to compare previous results with ours).

---

### Meta-Review · Area_Chair_roWS · 2024-12-20

**Metareview:**

This paper investigates the convergence behavior of adaptive optimizers such as AdaGrad and Adam under the influence of heavy-tailed noise, a scenario relevant in both theoretical and practical contexts. The authors demonstrate that without gradient clipping, these methods can fail to converge in heavy-tailed noise settings. To address this, they propose an analysis of AdaGrad and Adam with gradient clipping, deriving high-probability convergence bounds that exhibit polylogarithmic dependence on the confidence level.

The paper has several strengths. The provided example illustrating the divergence of AdaGrad/Adam in heavy-tailed noise scenarios is both insightful and valuable for understanding the challenges in these settings. Additionally, the theoretical analysis is thorough and considers a wide range of cases, making it a contribution to understanding the behavior of adaptive optimizers.

However, the paper has notable weaknesses. There is a gap between the title and the specific algorithms analyzed. For instance, Algorithm 1 modifies the standard AdaGrad and Adam formulations by introducing a scalar-valued $b_t$, which deviates from the original algorithms. This discrepancy raises questions about the consistency between the theoretical analysis and practical implementations. The experimental validation is relatively weak, and there is ongoing debate among reviewers about the choice of $\beta_2$, particularly regarding its dependence on $K$. While $\beta_2 = 1 - 1/K$ aligns with prior theoretical analyses, it is not constant, and the implications of this choice remain a gap from classical Adam.

Although the choice of $\beta_2$ is not the main focus of the paper, the inconsistency between the analysis of modified AdaGrad/Adam and their standard forms, as well as the lack of experimental evidence to validate certain conjectures, limit the paper’s overall contribution. Given these concerns, I cannot recommend acceptance in its current version.

**Additional Comments On Reviewer Discussion:**

During the rebuttal period, most reviewers maintained a negative stance on the paper, with even the reviewers offering positive opinions expressing low confidence. The primary concerns revolved around gaps between the paper’s theoretical results and its claims, such as the treatment of $b_t$ as a vector rather than a scalar, the selection of  $\beta_2$, and inconsistencies with prior theoretical results. The authors' rebuttal did not sufficiently address these concerns in a convincing manner, leading to minimal improvement in reviewer scores. As a result, the overall consensus remained in favor of rejecting the paper.

---

### Decision · Program_Chairs · 2025-01-22

Reject